# 🎲 Can LVLMs Describe Videos like Humans? A Five-in-One Video Annotations Benchmark for Better Human-Machine Comparison

## Abstract

Large vision-language models (LVLMs) have made significant strides in addressing complex video tasks, sparking researchers' interest in their human-like multimodal understanding capabilities. Video description serves as a fundamental task for evaluating video comprehension, necessitating a deep understanding of spatial and temporal dynamics, which presents challenges for both humans and machines. Thus, investigating *whether LVLMs can describe videos as comprehensively as humans*—through reasonable human-machine comparisons using video captioning as a proxy task—will enhance our understanding and application of these models. However, current benchmarks for video comprehension have notable limitations, including short video durations, brief annotations, and reliance on a single annotator's perspective. These factors hinder a comprehensive assessment of LVLMs' ability to understand complex, lengthy videos and prevent the establishment of a robust human baseline that accurately reflects human video comprehension capabilities. To address these issues, we propose a novel benchmark, **FIOVA** (**F**ive **I**n **O**ne **V**ideo **A**nnotations), designed to evaluate the differences between LVLMs and human understanding more comprehensively. FIOVA includes 3,002 long video sequences (averaging 33.6 seconds) that cover diverse scenarios with complex spatiotemporal relationships. Each video is annotated by five distinct annotators, capturing a wide range of perspectives and resulting in captions that are $4 \sim 15$ times longer than most existing benchmarks, thereby establishing a robust baseline that represents human understanding comprehensively for the first time in video description tasks. Using the FIOVA benchmark, we conducted an in-depth evaluation of six state-of-the-art (SOTA) LVLMs, comparing their performance with humans. To enhance this evaluation, we proposed **FIOVA-DQ**, a novel event-based metric that incorporates weighted event importance derived from human annotations. Results show that while current LVLMs demonstrate some perception and reasoning capabilities, they still struggle with information omission and descriptive depth. Moreover, we found significant discrepancies between LVLMs and humans in complex videos, particularly where human annotators exhibited substantial disagreement, whereas LVLMs tended to rely on uniform strategies for challenging content. These findings underscore the limitations of using a single human annotator as the groundtruth for evaluation and highlight the need for new evaluation perspectives. We believe this work offers valuable insights into the differences between LVLMs and humans, ultimately guiding future advancements toward human-level video comprehension.

## 1 Introduction

Large Language Models (LLMs) have made significant strides in Natural Language Processing (NLP), excelling in tasks such as text generation (Li et al. (2024a;b); Mahapatra & Garain (2024)) and question answering (Zhuang et al. (2023); Saito et al. (2024)). Building on these advancements, large vision-language models (LVLMs), including GPT-4V (Achiam et al. (2023)) and LLaVA (Liu et al. (2024)), extend LLM capabilities into multimodal domains. LVLMs excel in integrating text, images, and videos, demonstrating remarkable progress in applications such as text-to-video gener-

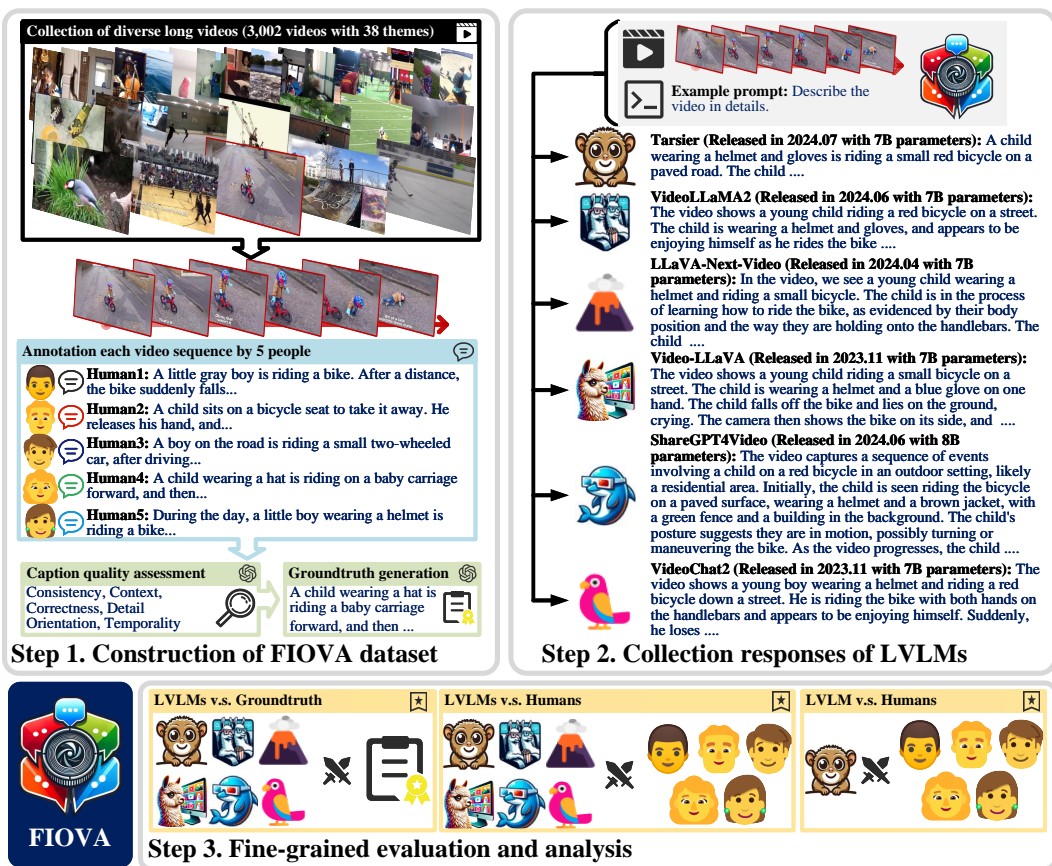

Figure 1: An overview of FIOVA. The overall workflow is divided into three steps (*i.e.*, construction of FIOVA dataset (see Section 2), collection responses of LVLMs (see Section 3), and fine-grained evaluation and analysis (see Section 4)), culminating in a benchmark that comprehensively compares the video understanding capabilities of humans and LVLMs.

ation (Huang et al. (2024b)) and video captioning (Huang et al. (2024a)). However, evaluating the true capabilities of LVLMs remains challenging, as traditional evaluation methods—typically based on text matching or embedding distances—often fail to capture the nuanced understanding required for human-like video comprehension (Hu et al. (2024b;a; 2022)).

This leads to the fundamental question: "*Can video-based LVLMs describe videos as comprehensively as humans?*" Video captioning (Aafaq et al. (2019); Ramanishka et al. (2016)) serves as a key task to assess a model's ability to perceive, comprehend, and generate meaningful video descriptions. Unlike structured tasks like object recognition (Logothetis & Sheinberg (1996)) or question answering (Antol et al. (2015)), video captioning demands an in-depth understanding of both spatial and temporal dynamics, presenting significant challenges for both machines and humans. Thus, investigating this question through reasonable human-machine comparisons using video captioning as a proxy task will enhance our understanding and application of these LVLMs.

However, current benchmarks (Miech et al. (2019); Lee et al. (2021); Chen & Dolan (2011); Caba Heilbron et al. (2015); Xu et al. (2016); Chen et al. (2024b); Zhou et al. (2018)) exhibit several major limitations: they typically feature simple scenarios (videos lasting about 10 seconds), provide brief annotations (averaging 15 words), and rely on single annotators (see Tab. 1). These constraints limit the insight into LVLMs' understanding of complex, long-duration videos and prevent the establishment of a robust human baseline that accurately reflects human comprehension capabilities.

To address these challenges, we propose a novel benchmark, **FIOVA** (**F**ive **I**n **O**ne **V**ideo **A**nnotations), designed to provide a comprehensive evaluation of the differences between LVLMs and human understanding. As shown in Fig. 1, FIOVA encompasses three key contributions: (1) **Comprehensive dataset construction:** We curated a dataset of 3,002 long video sequences (aver-

Table 1: Comparison of FIOVA and other video caption datasets. We split the datasets into two groups: automatic caption by ASR (Automatic Speech Recognition) (Miech et al. (2019); Lee et al. (2021); Zellers et al. (2021); Xue et al. (2022); Chen et al. (2024b)) or LVLM, and manual caption (Chen & Dolan (2011); Xu et al. (2016); Zhou et al. (2018); Caba Heilbron et al. (2015); Anne Hendricks et al. (2017); Rohrbach et al. (2015); Wang et al. (2019a; 2024a)). It is worth noting that FIOVA is the only dataset that provides multiple annotations for each video.

| Dataset | Text | Domain | #Videos | Avg/Total Video Len | | Avg Text Len |
|---------|------|--------|---------|------|------|--------------|
| HowTo100M | Automatic caption (by ASR) | Open | 136M | 3.6s | 134.5Kh | 4.0 words |
| ACAV | Automatic caption (by ASR) | Open | 100M | 10.0s | 277.7Kh | - |
| YT-Temporal-180M | Automatic caption (by ASR) | Open | 180M | - | - | - |
| HD-VILA-100M | Automatic caption (by ASR) | Open | 103M | 13.4s | 371.5Kh | 32.5 words |
| Panda-70M | Automatic caption (by LVLM) | Open | 70.8M | 8.5s | 166.8Kh | 13.2 words |
| MSVD | Manual caption (1 person) | Open | 1,970 | 9.7s | 5.3h | 8.7 words |
| LSMDC | Manual caption (1 person) | Movie | 118K | 4.8s | 158h | 7.0 words |
| MSR-VTT | Manual caption (1 person) | Open | 10K | 15.0s | 40h | 9.3 words |
| DiDeMo | Manual caption (1 person) | Flickr | 27K | 6.9s | 87h | 8.0 words |
| ActivityNet | Manual caption (1 person) | Action | 100K | 36.0s | 849h | 13.5 words |
| YouCook2 | Manual caption (1 person) | Cooking | 14K | 19.6s | 176h | 8.8 words |
| VATEX | Manual caption (1 person) | Open | 41K | $\sim$10s | $\sim$115h | 15.2 words |
| DREAM-1K | Manual caption (1 person) | Open | 1K | 8.9s | 2.5h | 59.3 words |
| **FIOVA (Ours)** | Manual caption (5 people) | Open | 3K | 33.6s | 28.3h | 63.28 words |

aging 33.6 seconds) that cover diverse scenarios with complex spatiotemporal relationships. Each video is annotated by five distinct annotators, capturing a wide range of human perspectives and resulting in captions that are 4 to 15 times longer than most existing benchmarks, establishing a robust baseline that comprehensively represents human understanding in video description tasks (see Section 2). (2) **Evaluation of state-of-the-art LVLMs:** We conducted an in-depth evaluation of six representative open-source LVLMs (VideoLLaMA2, LLaVA-NEXT-Video, Video-LLaVA, VideoChat2, Tarsier, and ShareGPT4Video), ensuring our evaluation reflects the latest advancements in the field. Additionally, we applied diverse processing techniques to model outputs, enabling a more comprehensive assessment of their capabilities and limitations (see Section 3). (3) **Fine-grained human-machine comparative analysis**: Leveraging the FIOVA benchmark, we performed detailed experiments to analyze the differences between LVLMs and human annotations across various aspects of video comprehension. To further enhance this analysis, we proposed **FIOVA-DQ**, an optimized event-based evaluation metric that incorporates human annotators' perspectives through weighted event importance, enabling a more fine-grained comparison of semantic understanding, fluency, and content relevance (see Section 4).

By providing a benchmark with multiple human annotations, FIOVA aims to bridge the gap between LVLM and human video understanding, offering insights into the current state of LVLMs and guiding the development of future AI systems for video comprehension tasks.

## 2 CONSTRUCTION OF FIOVA DATASET

Fig. 1 illustrates an overview of our work. In this section, we will introduce the first step in detail. Initially, we gathered FIOVA dataset $D = \{(V_1, C_1), \ldots, (V_n, C_n)\}$, in which $C_i = \{c_{i1}, c_{i2}, c_{i3}.c_{i4}, c_{i5}\}$ represents the set of human annotations for video $V_i$ (see Section 2.1). On this basis, we also combined $C_i$ to form a groundtruth $g_i$ as a comprehensive baseline for human understanding of video $V_i$ (see Section 2.3). Totally, FIOVA contains 3,002 $(V_i, C_i, g_i)$ pairs (*i.e.*, 3,002 videos, 15,010 human original descriptions, and 3,002 groundtruth descriptions).

### 2.1 VIDEO COLLECTION AND ANNOTATION

We curated a dataset consisting of 3,002 videos and 15,010 descriptions, specifically designed to evaluate the video comprehension capabilities of LVLMs. It spans 38 diverse themes, encompassing a wide range of real-world scenarios and interactions (see Appendix B.1).

To ensure high-quality annotations, each video was annotated by five individuals, focusing solely on the visual content, excluding audio or subtitles, except for naturally occurring text within the scene. This process emphasizes observable video elements, enhancing the dataset's relevance for video comprehension tasks. Annotators followed standardized guidelines to ensure consistency (see

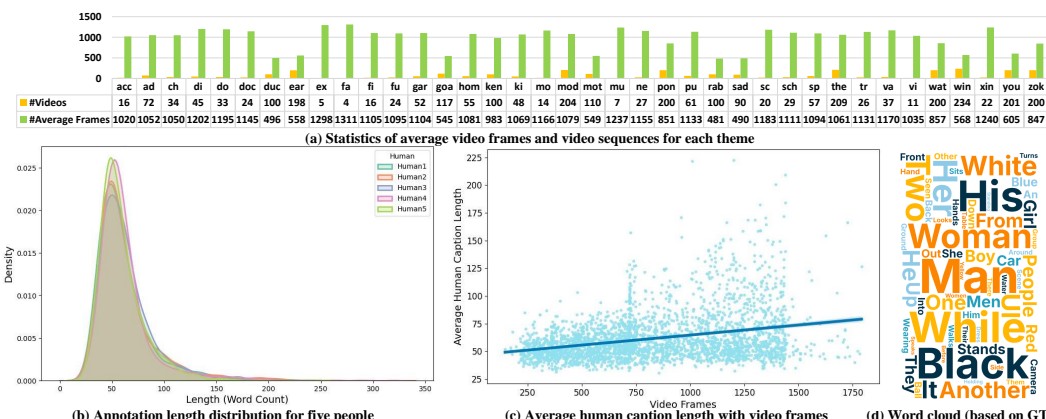

Figure 2: Statistical analysis of key aspects in FIOVA. (a) Statistics of average video frames and video sequences for each theme, see Tab. A1 for details of each theme. (b) Annotation length distribution for five people. The distribution of description lengths across human annotators remains highly consistent. (c) Average human caption length with video frames. The length of human descriptions increases with the length of the video, but the increase is not large and no redundant descriptions occur. (d) The word cloud of human descriptions (based on the groundtruth).

Appendix B.2), which included details like time of day, location, and prominent objects or actions, while avoiding literary or emotionally charged language. Public figures were described generically, and descriptions strictly adhered to the chronological order of events. These guidelines ensured neutrality, clarity, and factual accuracy, providing a reliable foundation for evaluation.

FIOVA presents additional challenges that distinguish it from existing datasets, making it more demanding for video understanding tasks. As shown in Fig. A1, FIOVA includes videos with varying resolutions and aspect ratios, requiring models to adapt to different visual formats. Frequent camera switches and diverse main subjects add complexity, challenging models to accurately follow transitions and identify critical elements. Moreover, FIOVA features footage with lens distortions, such as those from fisheye lenses, further complicating the interpretation of spatial relationships. These challenges are intended to stress-test LVLMs, pushing them to achieve higher adaptability and robustness in video comprehension.

Each video sequence is paired with five distinct English descriptions written by human annotators as coherent paragraphs of multiple declarative sentences. The number of sentences varied depending on the video's complexity, allowing for detailed accounts of events and transitions. With an average video length of 33.6 seconds, the dataset captures complex actions and interactions, making it ideal for tasks that require deep video understanding. Tab. 1 compares FIOVA with other existing datasets, and Fig. 2 presents statistical dimensions of FIOVA. Compared to others, FIOVA is annotated by multiple annotators and features more detailed and precise descriptions.

## 2.2 CAPTION QUALITY ASSESSMENT

In Section 2.1, we provided descriptions from five different annotators for each video, capturing diverse human perspectives to establish a robust human baseline. In addition to this diversity, a consolidated human description was generated as the final groundtruth, serving as a refined summary for video captioning evaluation. To create the groundtruth, we used GPT-3.5-turbo to evaluate descriptions across five key dimensions, following methods similar to those in Video-ChatGPT (Maaz et al. (2023)) and Tarsier (Wang et al. (2024a)). Following VideoLLaMA2 (Cheng et al. (2024)), these dimensions are: (1) **Consistency:** Whether the description is logically coherent and aligned with the video content. (2) **Context:** Whether the description accurately captures scene changes and relationships between events. (3) **Correctness:** Whether the information is accurate and free from misleading content. (4) **Detail Orientation:** Whether the description captures critical details, such as people, objects, scenes, and events. (5) **Temporality:** Whether the description follows the chronological order of events without skipping or over-summarizing. GPT-3.5-turbo assigned scores ranging from 1 to 10 for each caption across five dimensions (see Appendix D.1.1). This scoring allowed us to comprehensively analyze the quality of each annotator's description and identify those with the highest consistency and accuracy.

To better visualize the evaluation results, we plotted the score distribution of human annotators across all videos and all five dimensions. As shown in Fig. 3 (a-e), the score distributions are relatively consistent across different dimensions, indicating that the annotations are representative and reflect an average human understanding with reasonable cognitive abilities. Notably, the distribution for Detail Orientation differs slightly from other dimensions, suggesting that human captions generally provide above-average coverage of content and details, capturing most of the critical points in the videos. However, there are still deficiencies in specific details or comprehensiveness.

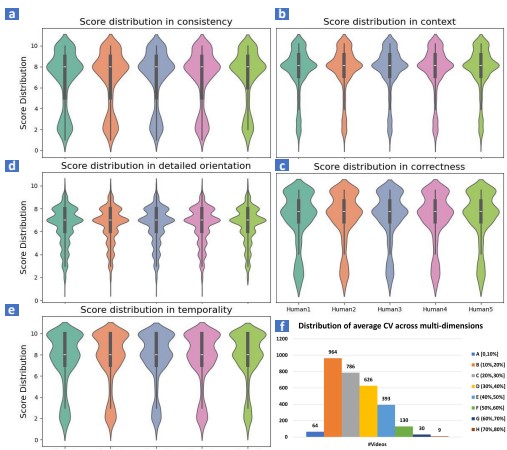

Building on this, we further examined the variability among annotators. To quantify this variability, we calculated the coefficient of variation (CV) based on the standard deviation and mean of the scores. A higher CV for a particular video indicates greater annotation variability, suggesting divergent interpretations among annotators. We refer to this variability as *disagreement*, reflecting differences in understanding among annotators. To perform a more detailed analysis of these disagreements, we added a sixth dimension—Annotation Length (see Fig. 2 (b))—to the existing five evaluation dimensions. By calculating the average CV for each video across all six dimensions (see Algorithm A1), we divided the dataset into eight distinct sub-groups based on the CV values (see Fig. 3 (f) and Appendix B.4). Videos with lower CVs (Group A) indicate high similarity in annotators' descriptions across multiple dimensions, while higher CVs (Group H) signify greater discrepancies. This classification not only provides insight into the variability in human annotations but also lays a foundation for subsequent algorithm evaluation, allowing us to compare different LVLMs to human groups in terms of video comprehension.

Figure 3: Distribution of scores from human annotators across multi-dimensions. (a-e) The distribution of human annotation scores as evaluated by GPT-3.5-turbo, focusing on the dimensions of consistency, context, correctness, detail orientation, and temporality. (f) The distribution of disagreement in video descriptions, measured by the average CV (coefficient of variation) among human annotators across multi-dimensions.

### 2.3 GROUNDTRUTH GENERATION

We used the GPT-3.5-turbo model to synthesize the five human-provided descriptions into a single, comprehensive video description that serves as the final groundtruth (see Appendix D.1.2). During this synthesis, the model integrates key elements from each of the five descriptions, balancing the diversity of perspectives with consistency and coherence. This ensures that the final groundtruth captures the most salient and informative aspects of the video while maintaining logical flow and completeness across all relevant dimensions, as illustrated in Fig. 4.

Using GPT-3.5-turbo for synthesis provides a systematic way to combine multiple viewpoints, reducing subjective bias and ensuring that no crucial detail is omitted. Each synthesized groundtruth represents a consolidated understanding of the video, balancing detail orientation, contextual relevance, and temporal accuracy. By combining the strengths of multiple human annotations, the generated groundtruth not only supplements individual descriptions but also sets a higher standard of quality, serving as a more stringent and standardized benchmark for evaluating model performance.

## 3 LVLMs RESPONSE COLLECTION

As illustrated in step 2 of Fig. 1, in this section, each video $V_i$ is processed by several LVLMs to form a benchmark of video & description & response pairs, denoted as $B = \{(V_i, C_i, R_i) \mid (V_i, C_i) \in D\}$, in which $R_i = \{r_{i1}, r_{i2}, \ldots, r_{in}\}$ represents the set of LVLMs' response for video $V_i$.

### 3.1 BASELINE MODELS SELECTION

We utilized six SOTA open-source LVLMs for our study: VideoLLaMA2 (Cheng et al. (2024)), Video-LLaVA (Lin et al. (2023)), LLaVA-NEXT-Video (Zhang et al. (2024)), Tarsier (Wang et al. (2024a)), VideoChat2 (Li et al. (2023)), and ShareGPT4Video (Chen et al. (2024a)). More detailed

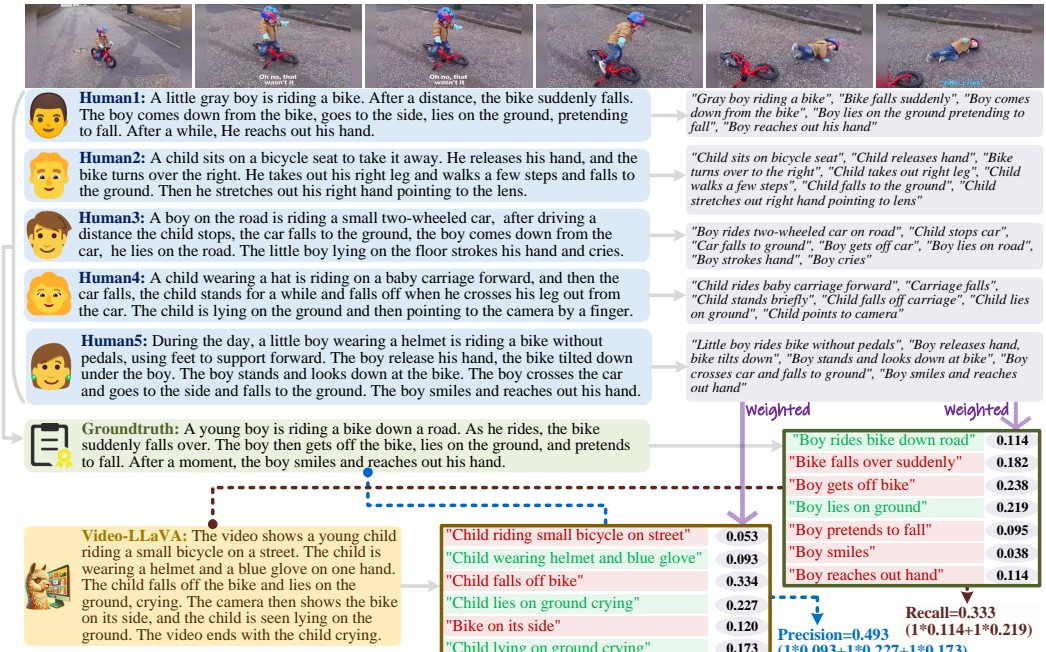

Figure 4: An example of FIOVA (see Fig. A7 for more details) and the calculation process of FIOVA-DQ.

introductions for these LVLMs can refer to Appendix A.1. These models were prompted with video description tasks, generating 18,012 responses (see Appendix D.2). The distribution of response lengths for each LVLM is shown in Fig. 5, which provides insight into the variability of model outputs. VideoLLaMA2 used default settings with a temperature of 0.2 and a maximum token limit of 1,024. VideoChat2 and ShareGPT4Video were configured with default settings, a temperature of 1.0, top_p of 0.9, and a maximum token limit of 1,024. Video-LLaVA had a temperature of 0.1 and the same token limit. Tarsier and LLaVA-NEXT-Video were set with a temperature of 0, top_p of 1, and a maximum token limit of 1,024. All models processed 8 frames using four RTX 3090 GPUs.

## 3.2 EVENT GENERATION

The video descriptions generated by the LVLMs in the previous section are suitable for evaluation using traditional metrics. However, the recently proposed AutoDQ (Wang et al. (2024a)) provides a novel event-based evaluation approach by extracting events from both reference and model-generated captions, enabling fine-grained assessments based on event matching. While AutoDQ has demonstrated its effectiveness in aligning model-generated descriptions with human annotations, it does not account for the cognitive importance of differ-

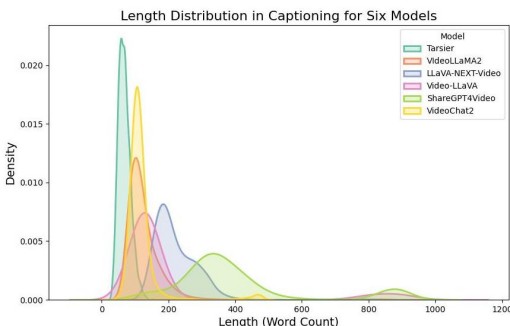

Figure 5: The distribution of response length.

ent events as perceived by human annotators. To address this limitation, we propose FIOVA-DQ, an extended evaluation metric that incorporates human cognitive weights into the event-based evaluation process. By assigning weights to events based on their importance across multiple annotators, FIOVA-DQ offers a more human-aligned assessment framework (see Section 4.1).

To support a broader range of evaluation metrics and achieve a comprehensive analysis, we used GPT-3.5-turbo to perform event extraction on both the groundtruth $g_i$ and the $j$-th LVLM's generated output $r_{ij}$ (see Appendix D.1.3). This ensures consistency and accuracy in event extraction. From this process, event collections $E^{gt}i$ for $g_i$ and $E^r ij$ for $r_{ij}$ are generated to support subsequent analysis. For FIOVA-DQ, each event in $E_i^{gt}$ is assigned a weight based on its average importance across the five annotators. These weights, normalized to sum to one, reflect the cognitive emphasis placed on different events by human annotators (see Fig. 4). This weighting mechanism enables

FIOVA-DQ to evaluate not only the alignment between model outputs and human annotations but also the relative importance of matched events, offering a more nuanced perspective.

# 4 FINE-GRAINED EVALUATION AND ANALYSIS

As shown in step 3 of Fig. 1, based on the FIOVA benchmark $D$, we compare LVLMs with both the representative human baseline (groundtruth) and the human interval (annotations by five individuals) across multiple dimensions. This allows for an in-depth analysis of the similarities and differences in video understanding between humans and LVLMs.

## 4.1 EVALUATION METHODS

Traditional metrics like BLEU (Papineni et al. (2002)) have limitations in evaluating detailed and longer video descriptions, often failing to capture the semantic nuances and contextual accuracy required for a comprehensive assessment. Recent studies have attempted to use models such as ChatGPT for content rating (Maaz et al. (2023); Achiam et al. (2023)), but the lack of interpretability in score assignment remains a challenge (see Appendix A.3). Therefore, we adopted AutoDQ (Wang et al. (2024a)), which extends traditional metrics like BLEU, GLEU, and METEOR by integrating text and semantic similarity, providing a more holistic evaluation of the alignment between LVLM-generated captions and human annotations.

To further enhance the evaluation process, we propose FIOVA-DQ, which builds upon AutoDQ by incorporating cognitive weights derived from human annotators. At first, events are extracted from both the groundtruth caption ($E^{gt}i$) and the LVLM-generated caption ($E^{r}ij$), as described in Section 3.2. For AutoDQ, two ratios are computed: (1) the ratio of events in $E^{gt}i$ that are also present in $E^{r}ij$ (*i.e.*, recall), and (2) the ratio of events in $E^{r}ij$ that are also present in $E^{gt}i$ (*i.e.*, precision). For FIOVA-DQ, these ratios are adjusted using weights assigned to each event in $E_i^{gt}$ based on their cognitive importance as perceived by annotators. Then, the harmonic mean of weighted precision and recall (*i.e.*, weighted F1 score) is calculated to provide a balanced measure of model performance. This adjustment ensures that critical events are given more emphasis, aligning the evaluation process more closely with human judgment.

Finally, we employed a combination of traditional metrics (BLEU, GLEU, and METEOR), AutoDQ-based metrics (F1, Precision, and Recall), and the newly proposed FIOVA-DQ metrics (weighted F1, weighted Precision, and weighted Recall) for evaluation. These metrics collectively enable two main evaluation tasks: (1) **Overall evaluation:** Assigns quality scores to each generated caption, assessing whether LVLMs can describe videos at a level comparable to humans using all metrics. (2) **Batch evaluation:** Evaluates the relative performance of multiple model outputs, providing a nuanced understanding of the models' ability to produce human-like descriptions.

## 4.2 OVERALL EVALUATION FOR LVLMS

**Traditional metrics.** According to the results in Tab. 2, Tarsier demonstrates outstanding performance across most traditional metrics, while ShareGPT4Video ranks the lowest, with scores significantly below those of other models.

Tarsier's success can be attributed to its high lexical overlap with the groundtruth, as its generated captions frequently match the vocabulary used in the reference descriptions. However, its lower METEOR score compared to BLEU and GLEU reveals limitations in capturing synonym usage and morphological variations. This indicates that while Tarsier excels in aligning with the vocabulary of the groundtruth, it lacks linguistic diversity and expressive flexibility. In contrast, ShareGPT4Video faces significant challenges on FIOVA despite its demonstrated ability to generate detailed captions using sliding window-based methods and segment integration, which have been successful in other video understanding benchmarks. A closer analysis reveals that its captions often contain substantial redundancy, which adversely affects its performance on traditional metrics like BLEU, GLEU, and METEOR. These metrics prioritize lexical similarity and penalize repetitive or redundant content, highlighting ShareGPT4Video's struggles in maintaining conciseness and relevance.

These results underscore the importance of balancing lexical similarity with linguistic diversity and reducing redundancy to achieve comprehensive and high-quality video descriptions. This highlights the need for models that combine precise lexical alignment with expressive richness and efficiency.

Table 2: Comparison of LVLMs via different metrics. The background color represents the performance of the metric. The darker the green, the better the performance.

| LVLMs | Traditional Metrics | | | AutoCQ-based Metrics | | | FIOVA-DQ-based Metrics | | |
|---|---|---|---|---|---|---|---|---|---|
| | BLEU | METEOR | GLEU | F1 | Recall | Precision | F1 | Recall | Precision |
| Tarsier | 0.043 | 0.265 | 0.119 | 0.351 | 0.283 | 0.628 | 0.320 | 0.584 | 0.584 |
| VIdeoLLaMA2 | 0.030 | 0.268 | 0.088 | 0.325 | 0.245 | 0.680 | 0.304 | 0.250 | 0.645 |
| LLaVA-NEXT-Video | 0.020 | 0.270 | 0.060 | 0.301 | 0.221 | 0.674 | 0.286 | 0.229 | 0.644 |
| Video-LLaVA | 0.027 | 0.257 | 0.077 | 0.285 | 0.208 | 0.709 | 0.269 | 0.216 | 0.680 |
| ShareGPT4Video | 0.010 | 0.218 | 0.034 | 0.281 | 0.201 | 0.731 | 0.263 | 0.203 | 0.714 |
| VideoChat2 | 0.037 | 0.281 | 0.098 | 0.309 | 0.237 | 0.656 | 0.287 | 0.243 | 0.621 |

**AutoDQ-based metrics.** To evaluate the performance of LVLMs in video captioning, we utilized AutoDQ for fine-grained event-based segmentation and comparison between model-generated captions and groundtruth annotations (see Tab. 2). This approach assesses the models' understanding of video content in terms of completeness and granularity.

Tarsier achieved the highest scores in both F1 and Recall, indicating that its captions comprehensively cover the events in the groundtruth. This highlights Tarsier's strength in content completeness. However, its low Precision score reveals challenges with descriptive accuracy, as its captions often include irrelevant or inaccurate information. While Tarsier demonstrates a solid understanding of overall video content, its lack of precision suggests a tendency to overgenerate. In contrast, ShareGPT4Video recorded the highest Precision but the lowest Recall. The high Precision reflects its ability to generate accurate and error-free descriptions, focusing on key events. However, the low Recall underscores its conservative approach, as it omits significant portions of the video content. This trade-off results in captions that are concise yet fail to capture the full scope of the video.

Other LVLMs demonstrated intermediate performance, striking a balance between Recall and Precision with moderate scores across both metrics. These results reveal the varying strategies employed by different models—some prioritize content completeness, while others focus on accuracy. The evaluation highlights the need for future models to achieve a balance, combining comprehensive content coverage with high descriptive precision to enhance video captioning quality.

**FIOVA-DQ-based metrics.** We incorporate human-weighted event importance into AutoDQ, resulting in FIOVA-DQ, which more effectively captures human intuitive judgments of description quality. This approach proves particularly suitable for evaluating the consistency and fluency of model-generated descriptions in multi-event long videos. Compared to AutoDQ, FIOVA-DQ reveals significant discrepancies between Recall and Precision metrics, offering a more granular understanding of model performance and better reflecting human preferences.

As with AutoDQ, Tarsier achieves the highest F1 and Recall scores. Notably, its Recall metric shows substantial improvement, indicating that Tarsier effectively captures most events, including key information emphasized by human annotators. However, its Precision metric decreases further, exposing deficiencies in event description accuracy under human-weighted evaluation—an aspect overlooked by previous metrics. For other LVLMs, the FIOVA-DQ metrics exhibit less pronounced changes compared to AutoDQ but follow a similar trend. The inclusion of human weighting enhances the metrics' sensitivity to human preferences, amplifying both the strengths and weaknesses of the models as evaluated on the FIOVA dataset.

## 4.3 BATCH EVALUATION FOR LVLMS

**Batch score evaluation for LVLMs.** In addition to evaluating the overall score, we conduct batch score evaluations across eight sub-groups (see Fig. 6). AutoDQ and FIOVA-DQ's performance trends are consistent with the overall evaluation, with Tarsier continuing to excel in Recall metrics. However, we observe a general decline in performance for most LVLMs in Group H. Group H consists of nine videos featuring multiple camera switches and frequent scene changes, with a coefficient of variation (CV) among human annotators exceeding 70%. These videos represent some of the most challenging content in the FIOVA dataset, making them particularly difficult to describe accurately. As expected, most LVLMs struggled to maintain descriptive completeness for Group H, resulting in notable omissions despite relatively accurate content. Interestingly, Tarsier performed better than other models in this group, likely due to its superior ability to capture temporal changes. This indicates that Tarsier is more capable of maintaining coherence amid rapid scene transitions, a critical factor for generating high-quality descriptions of complex sequences.

In terms of Precision, LVLMs demonstrated relatively consistent performance across different sub-groups, indicating their ability to accurately capture key details regardless of video complexity. Like overall evaluation, Tarsier's BLEU score is optimal in Group H, and its GLEU score remains stable across all sub-groups. GLEU allows for greater variation and emphasizes the fluency and overall quality of generated content, while BLEU focuses more on literal precision in word matching. Thus, when the generated text is semantically similar to the reference but differs in phrasing or word order, GLEU tends to assign a higher score, while BLEU is less favorable.

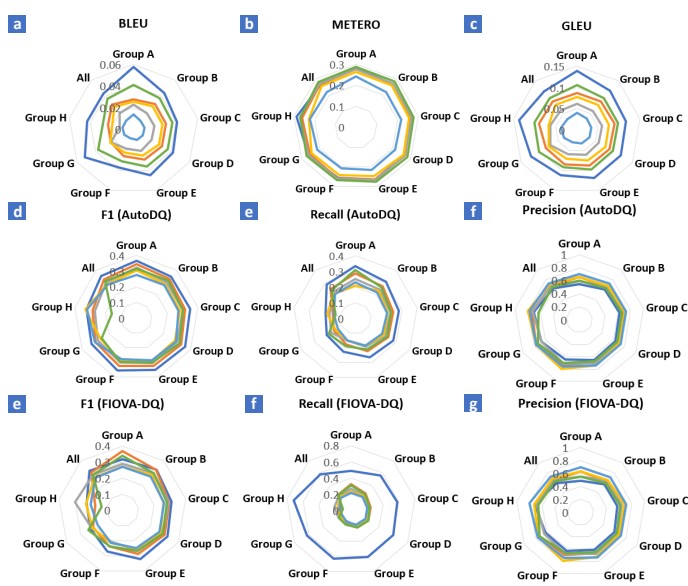

Figure 6: Radar plot of LVLMs on FIOVA and 8 sub-groups. See Appendix E.2 for details.

These findings underscore the limitations of traditional metrics in evaluating open-ended video captioning tasks. Metrics relying solely on lexical matching often fail to account for semantic coherence and fluency, both of which are critical for generating high-quality descriptions, particularly in complex videos with frequent scene transitions.

**Batch ranking for LVLMs.** Batch ranking serves as a key component to quantify the differences in consistency between LVLMs and human annotators when describing videos of varying difficulty levels. The procedure involves three main steps: (1) evaluating human annotators' consistency using six dimensions (Sec. 2.2), (2) assessing LVLM consistency across traditional metrics, AutoDQ, and FIOVA-DQ (use Algorithm A2), and (3) comparing the rankings of consistency scores between human and LVLM groups (use Algorithm A3). This approach combines multi-dimensional consistency evaluation with ranking difference analysis, providing a novel perspective for understanding the descriptive capabilities of LVLMs. A detailed process is shown in Fig. A8.

As shown in Fig. 7 (a), the CV of model performance decreases progressively from Group A to Group H. This trend suggests that models exhibit greater variability in performance for simpler videos (*e.g.*, Group A), whereas their outputs become more consistent for more complex videos (*e.g.*, Group H).

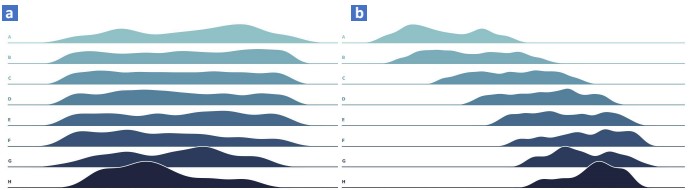

Figure 7: Comparison between humans and LVLMs based on the ranking of CV (coefficient of variation). (a) Ranking of CV for six LVLMs. (b) Difference between the ranking of CV for six LVLMs and humans.

The higher CV values in Groups A and B indicate that models employ diverse strategies for straightforward content, resulting in a broader range of descriptive quality. Conversely, as video complexity increases in Groups E to H, CV values decline, reflecting more stable outputs. This shift may be attributed to the increased difficulty of complex videos (*e.g.*, Group H), which imposes stricter requirements on descriptive capabilities, leading models to adopt more uniform approaches. These findings show the importance of evaluating models on complex and diverse content, as it reveals their ability to generalize and maintain stability under challenging conditions, providing deeper insights into their robustness.

**Batch ranking for LVLMs and humans.** Fig. 7 (b) shows that as the difficulty of accurately describing videos increases for humans (from Group A to Group H), the negative regions (such as Groups A and B) indicate that for easily describable videos, human annotators demonstrate more consistent performance, whereas models exhibit significant variations (see Fig. A19 in Appendix

E.4). This suggests that the models' descriptive capabilities are inadequate for simpler video content, failing to achieve the consistency demonstrated by humans.

Conversely, the positive regions (such as Group H) indicate that, for more challenging videos, human annotators exhibit greater variability in their descriptions, while the models display more consistent performance (see Fig. A21 in Appendix E.4). This consistency in models could be due to the similar strategies or shared limitations they employ when describing complex scenarios, leading to more uniform outputs. Most intermediate groups (such as C, D, and E) are close to zero, suggesting that for these videos, the coefficient of variation is relatively similar between models and humans, with no clear advantage for either (see Fig. A20 in Appendix E.4).

These observations align closely with the Overall and Batch Score Evaluations. In the Overall Score, LVLMs demonstrate a Precision exceeding 0.6, significantly surpassing Recall. This highlights the models' ability to produce accurate descriptions while revealing their limitations in comprehensiveness, as critical details are often omitted. In Group H, a marked decline in Recall scores is observed, with Precision remaining stable, consistent with Batch Ranking results. This pattern suggests that while LVLMs can generate accurate and consistent descriptions for complex videos, their descriptive coverage remains insufficient, particularly for multi-event scenarios. Overall, these findings show the inherent trade-off between accuracy and comprehensiveness in LVLMs' descriptive capabilities. Enhancing these models to balance high precision with comprehensive content coverage is essential, especially in complex video contexts where human annotations often exhibit significant variability.

### 4.4 SUMMARY

Based on the above results, we conclude that existing LVLMs exhibit notable perception and reasoning capabilities, enabling reasonably accurate video descriptions. However, most models face challenges with information omissions, limiting their ability to generate semantically comprehensive captions. Among the six evaluated models, Tarsier achieved the best overall performance, effectively leveraging temporal relationships to handle complex videos. Nevertheless, it requires improvements in descriptive precision and minimizing irrelevant content.

Compared to human-generated captions, LVLMs show significant discrepancies in simpler videos, often missing subtle nuances that human annotators readily capture. In contrast, for complex videos, LVLMs demonstrate greater consistency and stability, likely due to uniform strategies adopted under challenging scenarios. For videos of moderate complexity, LVLMs perform comparably to humans, balancing accuracy and completeness. However, issues such as hallucinations and redundancy remain prominent in some models, as illustrated in Fig.A23, Fig.A22, and Fig. A24. While all six models perform well in simple scenarios, such as Brazilian Jiu-Jitsu practice, their performance declines significantly when handling spatiotemporal inconsistencies or frequent scene transitions. These findings highlight the need for substantial improvements in processing complex video scenes with intricate temporal dynamics.

The experiments also reveal the limitations of traditional metrics in assessing open-ended video descriptions. These metrics rely on lexical matching, making them inadequate for capturing the semantic richness, fluency, and contextual relevance of captions, particularly for tasks involving diverse content and nuanced understanding. To address these limitations, new evaluation metrics are urgently needed. Future metrics should emphasize semantic alignment, linguistic fluency, and content relevance to provide a more comprehensive and accurate evaluation of LVLMs' capabilities.

## 5 CONCLUSIONS

This paper proposes FIOVA, a new benchmark designed to evaluate the judgment capabilities of LVLMs in video captioning across different evaluation settings and to assess their consistency with human judgments. Our findings indicate that while Tarsier performs well in terms of precision and temporal utilization, it often generates brief captions that lack detail, limiting comprehensiveness. In contrast, ShareGPT4Video, although comparable to GPT-4V in its claimed understanding, suffers from hallucinations and redundancy in its outputs. The FIOVA benchmark provides a complex environment for comparing LVLMs to human assessments, offering insights into their respective strengths and limitations across diverse video scenarios. Our results also emphasize the need for improved LVLMs that can effectively balance accuracy, comprehensiveness, and content relevance, particularly in complex settings. We hope that FIOVA will support further research in advancing video description and understanding.

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

APPENDIX

# A COMPREHENSIVE RELATED WORKS

## A.1 LVLMS FOR VIDEO CAPTION

In recent years, research on Large Vision-Language Models (LVLMs) has seen a notable surge, with some models even claiming to achieve performance on par with GPT-4V (Achiam et al. (2023)) in handling general video tasks such as visual question answering and video description. These advanced models aim to bridge the gap between visual and linguistic understanding, allowing for more sophisticated interactions with video content.

One of the standout models in this domain is Tarsier (Wang et al. (2024a)), which employs CLIP-ViT to encode individual video frames and leverages a Large Language Model (LLM) to model the temporal relationships between these frames. Through a carefully crafted two-stage training process, Tarsier demonstrates superior capabilities in generating video descriptions compared to existing open-source models, making it a leading player in this rapidly evolving space.

Building on earlier innovations, VideoLLaMA2 (Cheng et al. (2024)) advances video captioning by improving on its predecessor, VideoLLaMA (Zhang et al. (2023)). It introduces a custom-designed Spatio-Temporal Convolution (STC) connector that effectively captures the complex interplay between spatial and temporal elements in video data. This enhancement enables the model to generate more accurate and context-aware video descriptions and address broader video understanding tasks.

Another notable development comes from ShareGPT4Video (Chen et al. (2024a)), which advances video understanding in LVLMs and video generation in text-to-video models (T2VM) to new levels. By generating dense, detailed, and precise captions, ShareGPT4Video achieves state-of-the-art (SOTA) performance across three advanced video benchmarks, significantly enhancing the quality of video descriptions and the overall understanding of complex video content.

Video-LLaVA (Lin et al. (2023)) further pushes the boundaries of foundational LLMs by aligning visual representations with the language feature space, working towards a more unified LVLM architecture. This alignment is critical in enhancing the model's ability to understand and generate coherent, contextually appropriate captions that seamlessly integrate both visual and linguistic elements.

VideoChat2 (Li et al. (2023)) stands out for its impressive capabilities in spatio-temporal reasoning, event localization, and causal reasoning. By integrating a video backbone with a large language model via a learnable neural interface, VideoChat2 excels in tasks that require a deeper understanding of temporal sequences and the causal relationships between events in video data. This makes it particularly effective in scenarios that demand detailed analysis and interaction with dynamic video content.

The emergence of these models has prompted researchers to ask a fundamental question: "*Can video-based LVLMs describe videos like humans and exhibit human-level understanding?*" This question forms the basis of our work. We selected these state-of-the-art models as evaluation subjects and conducted a comprehensive comparison of human and machine video understanding using the FIOVA benchmark.

## A.2 VIDEO CAPTION DATASET

As the field of video understanding continues to evolve, researchers have introduced a growing number of video description datasets that cater to various levels of complexity and diversity in video content. These datasets play a crucial role in advancing video captioning models by providing training and evaluation materials that reflect real-world challenges.

One of the well-known datasets in this field is YouCook-II (Zhou et al. (2018)), which comprises 2,000 cooking videos evenly distributed across 89 distinct recipes. These videos, sourced from YouTube, encompass a wide range of cooking techniques and present various challenges typical of open-domain videos. The dataset features variations in camera angles, camera movement, lighting conditions, and background changes, making it an excellent resource for testing models on dynamic and complex scenarios.

The Microsoft Video Description (MSVD) (Chen & Dolan (2011)) dataset offers another foundational benchmark for video captioning tasks. It includes 1,970 short video clips from YouTube, each paired with human-annotated sentences that provide natural language descriptions of the video content. This dataset is widely used for training and evaluating models, given its open-domain nature and the diversity of content it covers.

Further expanding the scope, the MSR-Video to Text (MSR-VTT) (Xu et al. (2016)) dataset offers a larger and more diverse collection of open-domain videos for captioning tasks. It consists of 7,180 videos subdivided into 10,000 clips, organized into 20 distinct categories that encompass a broad range of scenarios, from sports to news events, and more. The MSR-VTT dataset serves as a benchmark for evaluating a model's capability to handle diverse, real-world video content, making it an important resource for researchers seeking to enhance the generalization abilities of their models.

Currently the largest dataset in the field, Panda-70M (Chen et al. (2024b)), features an astounding 70 million videos paired with high-quality text captions. This extensive dataset has significantly accelerated the development of video understanding by providing a vast array of training examples that capture a wide spectrum of real-world video content. Its scale and diversity allow researchers to train more robust models capable of handling complex, open-world scenarios.

Notably, FIOVA stands out as the only dataset that provides multiple annotations for each video, offering richer insights into how different viewers perceive and describe the same content. Additionally, the length of the video descriptions in FIOVA is considerably longer than in other datasets, providing more detailed and nuanced explanations of the video content. This makes FIOVA an exceptional resource for testing the ability of models to generate comprehensive, contextually rich descriptions, pushing the boundaries of what video captioning systems can achieve.

### A.3 VIDEO CAPTION EVALUATION

In the early stages of video description research, the primary focus was on pretraining video-language models, followed by fine-tuning on specific datasets for video captioning tasks. The performance of these models was typically assessed using well-established metrics such as BLEU (Papineni et al. (2002)), GLEU (Mutton et al. (2007)), METEOR (Banerjee & Lavie (2005)), and CIDEr (Vedantam et al. (2015)). These metrics, while useful for measuring the quality of generated descriptions based on syntactic and semantic alignment, often led to models that could achieve impressive results on specific datasets. However, a significant limitation was that these models frequently struggled to generalize well beyond their training data, especially when confronted with more diverse or open-world videos (Wang et al. (2024a)).

To address this challenge, recent research efforts have shifted towards developing models capable of zero-shot video description (Tewel et al. (2022); Wang et al. (2019b); Zhou et al. (2024)). These models aim to generate accurate captions for unseen videos without requiring fine-tuning on task-specific datasets. Although promising, the simplicity of many standard video description benchmarks limits their ability to fully evaluate these models' capabilities. These benchmarks often focus on straightforward, short videos with basic actions, which fails to stress-test models on more complex, nuanced content.

As the complexity of videos increases—whether in terms of length, visual diversity, or intricate narrative structure—traditional evaluation metrics struggle to reflect the true quality and relevance of the generated captions. This mismatch highlights the need for more sophisticated evaluation methods. In response, researchers have recently proposed using advanced language models, such as ChatGPT, for automatic evaluation (Sottana et al. (2023)), which has gained popularity for tasks like open-ended question answering. While this approach offers more flexibility in evaluating the nuances of video descriptions, directly assigning a numerical score to an entire video description often lacks interpretability, with the meaning of each score level being ambiguous and inconsistent (Maaz et al. (2023)).

To overcome the limitations of traditional evaluation metrics, we adopted AutoDQ (Wang et al. (2024a)), a recently proposed approach for automatic scoring. AutoDQ offers significant advantages over traditional methods, as it combines both text similarity and semantic similarity to evaluate the alignment between the LVLMs' video captions and human-generated captions. This approach

enables a more comprehensive evaluation of both the lexical accuracy and the semantic integrity of the descriptions, making it better suited for assessing the quality of detailed, nuanced video captions.

The AutoDQ evaluation process involves two main stages. First, events are extracted from both the groundtruth and the LVLM-generated captions. In the next stage, these events are compared to calculate two key metrics: recall, which measures how much of the groundtruth's events are captured by the model-generated caption, and precision, which evaluates how accurately the generated content aligns with the events present in the groundtruth. Finally, the F1 score—a balanced measure of precision and recall—is used to provide an overall assessment of the model's performance. This method allows for a more nuanced understanding of how effectively a model captures the content of a video, considering both completeness and accuracy.

In our evaluation of LVLMs using the FIOVA benchmark, we employed both traditional metrics (such as BLEU, GLEU, and METEOR) and the advanced AutoDQ approach. By combining these evaluation methods, we aim to provide a more comprehensive analysis of model performance, capturing both the lexical alignment and the deeper semantic relationships that are crucial for effective video comprehension. This combined approach ensures a scientifically rigorous comparison between LVLMs and human-generated video captions, particularly in complex video scenarios.

# B  Detailed Information of FIOVA Dataset

## B.1  Theme Abbreviations and Corresponding Meanings

Table A1: The video theme of the FIOVA dataset.

| Prefix | Video Theme & Description |
|--------|---------------------------|
| acc | Accident: This category encompasses records of sudden events such as traffic accidents and unexpected collisions. |
| ad | Advertisement: This category includes video content of commercial advertisements and product promotions for marketing communication. |
| ch | Children: This category captures scenes of children's daily activities, play, and interactions. |
| di | Dialogue: This category includes video content featuring conversations, discussions, and communicative interactions. |
| do | Daily Observations: This category records observations and events from everyday life. |
| doc | Documentary: This category encompasses documentaries with educational, informational, or historical content. |
| duc | Daily Unique Content: This category showcases videos of unique or unusual events in daily life. |
| ear | Event Action Record: This category records actions and behaviors during specific activities or events. |
| ex | Examination: This category involves records of exams, tests, or other assessment activities. |
| fa | Family Activities: This category captures scenes of family activities, parent-child interactions, and family life. |
| fi | Film Industry: This category includes video content related to film production, actor performances, and behind-the-scenes of movies. |
| fu | Fun: This category includes videos with entertaining, fun, or humorous content. |
| gar | Gathering Activities Recordings: This category records videos of social activities, gatherings, and collective events. |
| goa | Games of Action: This category includes videos of action games, sports competitions, and outdoor activities. |
| hom | Home: This category captures scenes of home environments, domestic life, and family relationships. |
| ken | Kinetic Engaging Narratives: This category includes videos with dynamic participation, physical activities, and interactive narratives. |
| ki | Kids Interaction: This category records interactions and social activities among children. |
| mo | Motion: This category involves videos of physical movement, action displays, and dynamic expressions. |
| mod | Movement Onsite Display: This category showcases videos of on-site activities, movements, and mobility. |
| mot | Motor: This category includes videos of mechanical motion, vehicle operation, and engine functionality. |
| mu | Music: This category records videos of music performances, music creation, and musical activities. |
| ne | News Event: This category includes videos of news reports, news events, and news interviews. |
| pon | People's Ordinary Narratives: This category records videos of ordinary people's daily lives and personal stories. |
| pu | Public Utility: This category showcases videos of public services, public utilities, and municipal engineering. |
| rab | Recreational Activities and Behavior: This category includes videos of recreational activities, leisure behaviors, and entertainment venues. |
| sad | Sports and Daily Activities: This category records videos of sports activities, daily exercises, and outdoor activities. |
| sc | Scholarly Contexts: This category includes videos of scholarly research, educational contexts, and academic discussions. |
| sch | Social and Cultural Happenings: This category records videos of social events, cultural activities, and community life. |
| sp | Sports and Physical activities: This category includes videos of sports, physical exercises, and competitive activities. |
| the | Typical Human Experiences: This category records videos of typical human experiences, universal emotions, and everyday challenges. |
| tr | Thematic Representation: This category includes videos of thematic presentations, topic discussions, and thematic events. |
| va | Vacation and Activities: This category records videos of vacation activities, leisure travel, and holiday experiences. |
| vi | Various Interactions: This category includes videos of various interactions, social activities, and interpersonal relationships. |
| wat | Wildlife and Adventure Themes: This category records videos of wildlife, adventure activities, and nature exploration. |
| win | Warm Interactive Narratives: This category includes videos of warm interactions, touching stories, and positive communications. |
| xin | Experiences Interactions Narratives: This category records videos of experiential interactions, event narratives, and personal experiences. |
| you | Youthful Unison Observed: This category records videos of collective activities among young people, teamwork, and youthful vitality. |
| zok | Zoom Occurrences Kinetics: This category includes videos of fast-paced actions, dynamic events, and high-energy activities. |

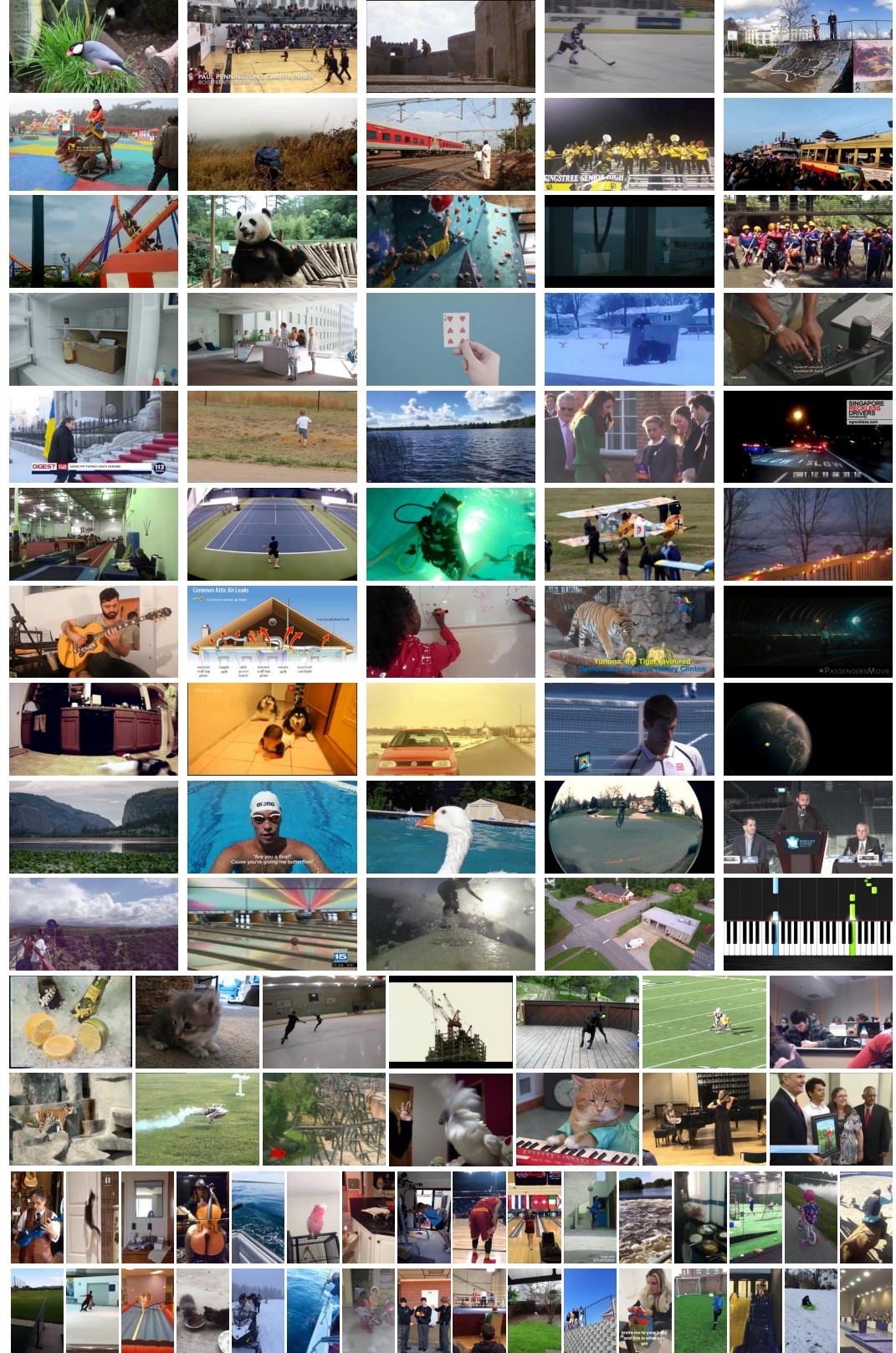

Figure A1: The representative data of FIOVA. Each video is strictly selected based on themes.

To ensure the legality, diversity, and high quality of the FIOVA dataset, we implemented a systematic approach to video sourcing and selection, as described below:

**Legitimacy of Video Sources.** All videos in the FIOVA dataset were sourced from legal and publicly accessible copyright-compliant platforms. The acquisition process adhered to the following principles:

- **Public Copyright Resources:** Videos were selected from platforms with explicit public copyright permissions, such as YouTube. These videos are explicitly allowed for non-commercial research purposes according to the terms of their source platforms.
- **Compliance Statement:** We strictly followed the terms of use of these platforms, ensuring that all selected videos comply with applicable copyright regulations. By choosing videos permitted for non-commercial research, we ensured the dataset's compliance.

**Diversity in Video Selection.** To construct a dataset capable of evaluating LVLMs across diverse scenarios, we prioritized diversity during the video selection process in the following aspects:

- **Coverage of Themes and Scenes:** The FIOVA dataset spans a wide range of themes, including daily activities, sports events, and natural landscapes. This diversity ensures that LVLMs can be evaluated across a variety of real-world scenarios.
- **Rich Dynamic Complexity:** Videos were carefully selected to represent complex dynamic characteristics, such as intricate spatiotemporal relationships, multi-agent interactions, and mixed short- and long-term sequences. These features reflect the actual challenges of semantic understanding tasks faced by LVLMs.

**Video Screening and Quality Control.** To ensure the quality of the dataset, we designed and executed a rigorous video screening and quality control process, comprising the following steps:

- **Initial Screening:** During the initial phase, videos meeting public copyright criteria were selected, with a focus on diversity in content.
- **Manual Review:** Each video underwent manual review to ensure clarity, narrative consistency, and suitability for video understanding tasks.
- **Multidimensional Processing:** At the processing stage, videos were grouped and balanced to ensure an appropriate distribution of length, content, and event complexity within the dataset, providing a reliable foundation for comprehensive LVLM evaluation.

By adhering to these strategies, the FIOVA dataset ensures legality, diversity, and high quality, serving as a representative framework for the evaluation and optimization of LVLMs.

B.2 HUMAN ANNOTATION RULES

---

**Annotation Scheme and Standards.**

1. **Annotators should label based solely on the visual content of the video, without referring to the audio content or any subtitles in the video**, except for the text that appears naturally in the scene (such as store signs, road signs, *etc.*). Annotators can choose to use this information or not, based on their judgment.
- **Example:** If a news image appears with the title "Earthquake Report," this text can be referenced. However, if text appears in the form of movie subtitles at the bottom of the video, it should not be used.

2. **Annotators should describe each video using a few simple declarative sentences to form a paragraph.** The number of sentences depends on the changes in events and scenes in the video, and the content can be appropriately enriched.

3. **Introduce simple and observable scene information**, such as time (morning, noon, evening, late night), location (*e.g.*, on a basketball court, beside a highway, in a bar), and the main objects and their positions in the scene (*e.g.*, a truck overturned in the middle of the road, spectators filling the stands around the stadium). Avoid using overly literary descriptions.

4. **Do not include the names of public figures** in the video, such as Obama, Clinton, Sun Yang, Yao Ming, Yang Mi, *etc.*Use third-person references such as "a man," "a woman," "a boy," "a girl," "he," "she," *etc.*, instead.

5. **Optionally include observable details of characters**, such as clothing, hairstyle and color, age, *etc.*, *e.g.*, "A basketball player wearing a white jersey dribbled past another player wearing a black jersey."

6. **Describe the behaviors and actions of individual characters as well as interactions between them.** For interactions between multiple people, use references such as "this person, that person," "one person, another person," "the one on the left, in the middle, on the right," or "this group, that group" to refer to different entities. There are no strict requirements for the specific language used, but the relationships and actions must be clearly and concisely described.

7. **Do not use emotionally biased words** (mostly adjectives or adverbs), such as "pitiful," "disgusting," "joyfully," *etc.*

8. **Do not use idioms** (*e.g.*, "a dime a dozen"), **proverbs** (*e.g.*, "No pain no gain"), or **internet slang** (*e.g.*, "imho (in my humble opinion)").

9. **Do not use overly literary descriptions** or speculate on the psychological state of characters. For example: "As she thought of her youth slipping away, a faint sorrow appeared on her face."

10. **Do not use subjective inference terms**, such as "obviously." Sentences should be concise; use shorter phrases where possible, *e.g.*, replace "at the same time" with "meantime."

11. **Do not use unnecessary conjunctions** if there is no causal relationship between events in the video.

12. **Avoid redundant or conversational language.** For example: instead of "Just after Andy rode his bike home, he immediately ran out again," simplify to "After a boy rode home, he ran out again" or "A boy rode home and then ran out again."

13. **The events in the video must be described in the order in which they occur**, without skipping ahead or using summarizing language.

---

To ensure the quality and robustness of the annotations in the FIOVA dataset, a carefully designed annotator arrangement strategy was implemented. Below, we describe the approach taken and its contributions to the diversity and representativeness of the dataset.

**Annotator Assignment.** Unlike some datasets annotated by a fixed group of individuals, the annotation of FIOVA involved multiple groups of annotators. Specifically:

- **Dynamic Annotator Groups:** Each video was independently annotated by five annotators; however, the annotators assigned to different videos varied.
- **Training and Standardization:** All annotators were required to undergo rigorous training to ensure a thorough understanding of the annotation guidelines and the ability to deliver consistent, high-quality annotations.

**Diversity in Annotations.** The use of multiple annotator groups was a deliberate choice aimed at enhancing the diversity, coverage, and adaptability of the GT. The key benefits of this approach include:

- **Diverse Descriptive Perspectives:** Allowing different annotators to work on the dataset brought varied linguistic styles and perspectives, minimizing bias that might arise from relying on a fixed annotator group.
- **Comprehensive Semantic Coverage:** The involvement of diverse annotators improved the coverage of video details, capturing nuanced aspects of the scenes and events depicted.
- **Enhanced Robustness:** The diversity in annotators' perspectives enabled the GT to better generalize and adapt to various evaluation scenarios, ensuring that the dataset remains applicable across diverse use cases.

**Quality Control Measures.** While annotator diversity introduces variability in descriptive styles, robust quality control measures were implemented to ensure the reliability and consistency of the annotations. These measures include:

- **Standardized Guidelines:** A unified set of annotation instructions was provided to all annotators, ensuring consistency across annotations.
- **Post-Annotation Review:** All annotations underwent a quality review process to verify their alignment with video content and eliminate errors.
- **Semantic Integration:** Using GPT-3.5-turbo, the annotations from five annotators were integrated into a single, cohesive description, balancing consistency with the retention of diverse perspectives.

Through these measures, the FIOVA dataset provides a robust, diverse, and high-quality GT that supports the evaluation of LVLMs in long-video description tasks.

B.3 WORD CLOUD FOR FIVE PEOPLE ANNOTATION AND LVLMs CAPTION

Figure A2: Word cloud for five people annotation and LVLMs caption.

### B.4 DISTRIBUTION OF DISAGREEMENT AMONG HUMAN ANNOTATORS (BASED ON MULTIPLE DIMENSIONS)

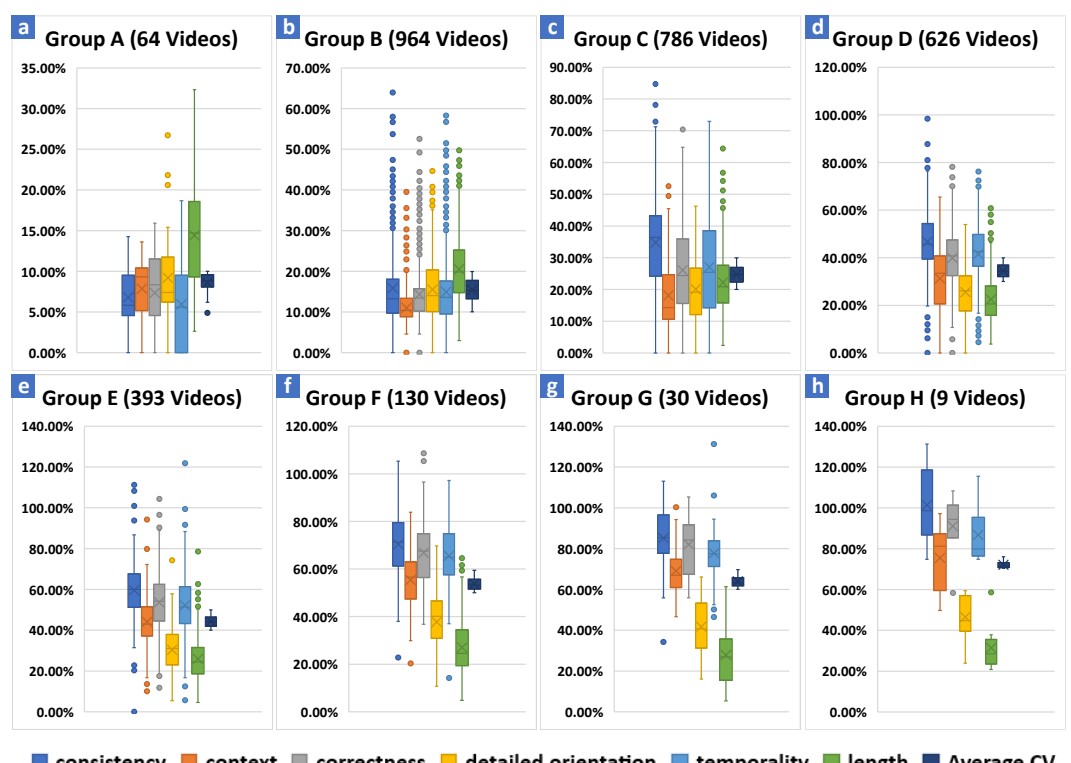

Figure A3: The distribution of the multi-dimensional coefficient of variation for 8 groups. Please refer to Section 2.2 for more details. The dataset is divided based on the coefficient of variation (CV) of human annotators across multiple dimensions, resulting in 8 groups. Each group represents a different degree of disagreement among the 5 annotators, ranging from the smallest (Group A) to the largest (Group H).

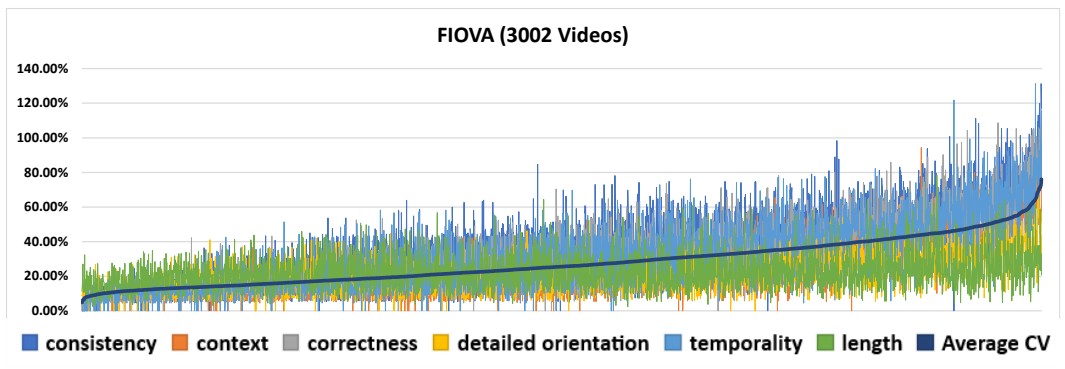

Figure A4: Distribution of multiple dimensions coefficient of variation (CV) and their mean value on FIOVA.

### B.5 EXAMPLES FOR VARIOUS GROUPS

To demonstrate the evaluation capability of GPT-3.5-turbo more intuitively, we provide the following cases to show the characteristics of different groups.

#### B.5.1 HIGH CONSISTENCY GROUP SAMPLE (GROUP A)

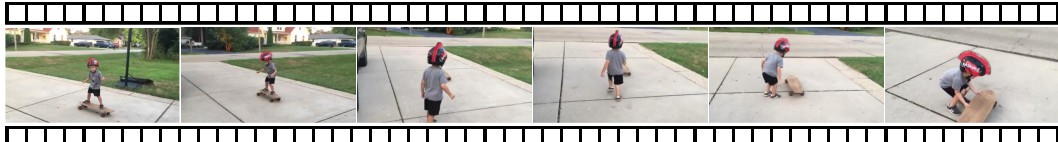

Figure A5: An example of videos in Group A.

The descriptions are highly consistent in style, semantically comprehensive, and linguistically fluent. The following are the annotators' descriptions:

- **Human 1:** *A little boy wearing a red hat stands on the scooter and slips forward. He waves his hand. He steps off the scooter. He looks at the scooter sliding forward and runs forward. The boy presses the scooter, then the scooter stops. The boy grabs the scooter with both hands.*

- **Human 2:** *A little boy wearing a helmet is playing skateboard in the open air. He jumps down and pauses. He looks at the skateboard slipping away. He runs following the skateboard and squats down, and stops the skateboard with his hand. He picks up the skateboard and holds it in the arms.*

- **Human 3:** *A child wearing a hat stands on a skateboard. There is a car next to the skateboard. The skateboard slides forward. The child goes down from the skateboard, and the skateboard continues sliding forward. The child catches up the skateboard and stops the skateboard by his hands. The child picks up the skateboard.*

- **Human 4:** *A boy wears a skateboard helmet. The skateboard slides forward. The boy waves hands to the camera len. The boy goes down from the skateboard. The skateboard still moves forward. The boy chases the skateboard and stops it. The boy picks up the skateboard.*

- **Human 5:** *A boy standing on a skateboard is doing skateboarding. The boy lands on one foot, while the other foot also takes off from the skateboard. The skateboard continues moving forward. The boy catches up the skateboard and stops it.*

The evaluation results by GPT-3.5-turbo indicate that the descriptions exhibit minimal differences in contextual consistency (CV: 0.00%) and context (CV: 0.00%), while showing small variations in correctness (CV: 4.56%) and temporality (CV: 4.56%). The detail orientation has a slightly higher variation (CV: 8.84%), and the length of descriptions displays the largest variation (CV: 11.40%). Overall, the average CV across all dimensions is 4.89%. These findings demonstrate highly concentrated semantic distributions across annotators, indicating strong agreement in their descriptions despite minor differences in specific dimensions.

#### B.5.2 HIGH VARIABILITY GROUP SAMPLE (GROUP H)

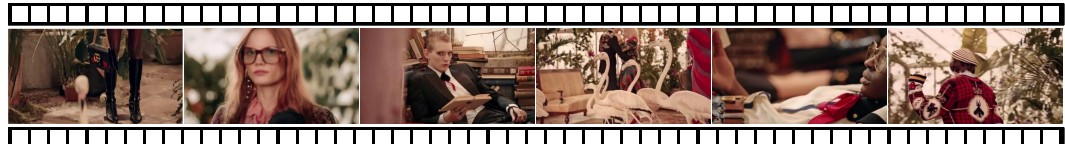

Figure A6: An example of videos in Group H.

The descriptions differ significantly in content, detail, and linguistic style:

- **Human 1:** *A woman wearing a small glasses is reading books. A woman wearing a big glasses is looking forward. A man sitting beside a lot of books and holding a book looks at the front. The woman wearing big glasses lies on the ground. A group of cranes walk by, a man and a woman dancing behind. A woman in pink walks, a man and a woman dancing behind. A black woman lies down and reads, a red dress woman sitting in a chair looks at the right. The woman with big glasses waves around the crane. A man wearing glasses is reading. The pink dress woman is walking through, the man wearing glasses is reading, the black woman is lying on a black and white shirt and reading. A man wearing a hat dances and walks through the black man upside down. A woman is lying next to a group of cranes. A woman steps on the book and walks. The woman in pink is dancing and walking through, a crane also comes.*

- **Human 2:** *The lens sweeps a lady from top to bottom, and then there appears a woman with curly hair. A man is wearing a suit, the man lying down is looking at her. Lens switch, the lady is lying on the floor, a group of white flamingos walk by, someone next to them is dancing. A man and a woman push around, the first lady appears lying down and reading, the man in suit also wears glasses reading, the curly hair women and flamingos are dancing, someone next to them stretches his leg doing exercise.*

- **Human 3:** *In a yard, a black-skinned woman is carrying a bag in the hands and reading a book, another long-haired woman is staring at the camera. A woman wearing a suit is lying on the stool, holding A book and looks at the lens, the long hair woman is lying on the carpet. A group of birds walk through the hall, a red dress man pushes a blonde woman away, the black skin woman next to him sitting to the side reads, another woman with black skin is lying down and reading. A woman wearing a red hat is sitting to the side, the long hair woman shakes hands, a woman in suit wears glasses, another woman wearing a striped shirt lies next to the carpet. The man in red keeps beating, A woman lying on the table raises her legs, the long hair woman is lying on the carpet, a pink dress woman is shaking the body and walking through.*

- **Human 4:** *A woman standing next to some leaves. A woman is lying on the ground. Some geese are walking. A man and a woman are talking. A man is reading a book. A woman is sitting in a chair. A woman is waving her hands. A man is wearing glasses. Several people are lying on the ground. A man is leaning up and a man is walking by his side.*

- **Human 5:** *A woman carrying a bag is standing and reading. A woman wearing glasses looks at the camera. A person holding a book looks at the woman. The woman wearing glasses is lying on the ground. Several people are dancing, a person is lying down and reading, a person is sitting on a chair. A man is waving his hands. The reading people wears the glasses. A man jumps forward and looks at another person who stands on the stool. The women with glasses is lying on the ground. A person steps on the book. Everyone does their own thing.*

The evaluation results by GPT-3.5-turbo highlight significant variability across annotators' semantic coverage and linguistic styles. Consistency exhibits the highest variability with a CV of 98.54%, followed by correctness (CV: 105.34%), temporality (CV: 76.70%), and context (CV: 49.79%). Descriptions also show notable differences in detail orientation (CV: 53.93%) and length (CV: 37.87%). Overall, the average CV across all dimensions is 70.36%, reflecting substantial semantic inconsistency. These findings underline the diversity in annotators' understanding and descriptions of the video, capturing a wide range of perspectives and interpretative styles.

## B.6 EXAMPLE FOR CORRESPONDENCE BETWEEN VIDEOS, HUMAN DESCRIPTIONS, AND GENERATED GROUNDTRUTH

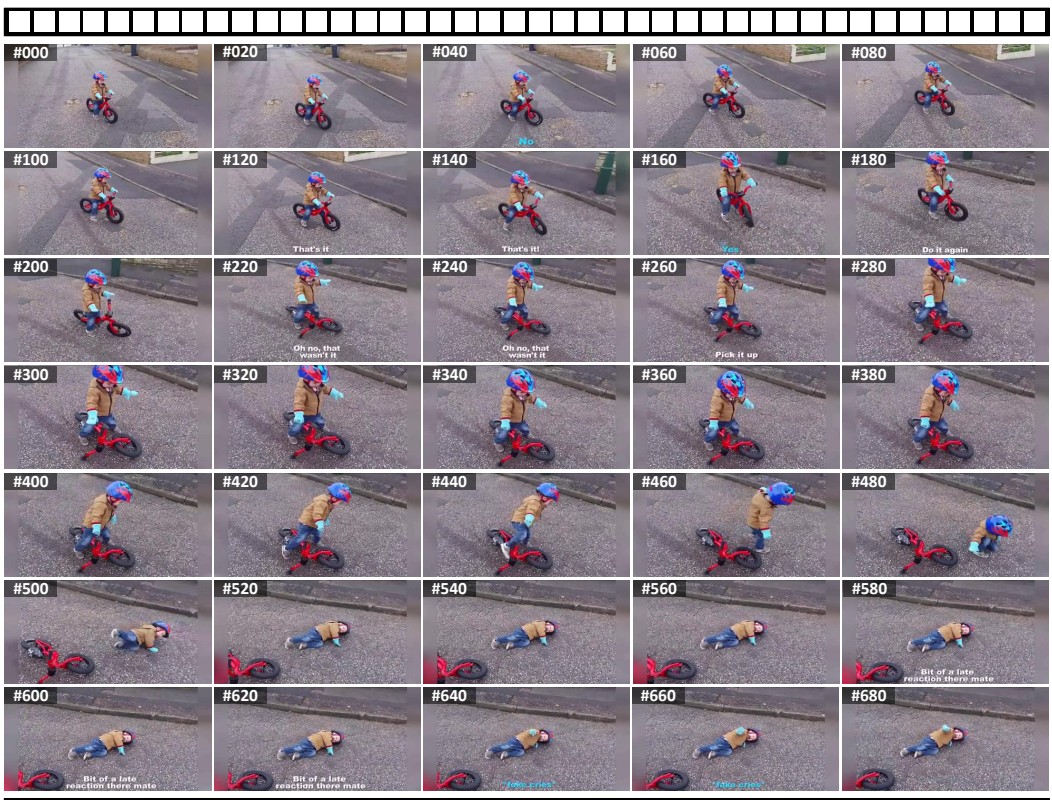

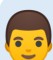 **Human1:** (#000) A little gray boy is riding a bike. (#220) After a distance, the bike suddenly falls. (#440) The boy comes down from the bike, goes to the side, (#500) lies on the ground, pretending to fall. (#640) After a while, He reachs out his hand.

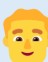 **Human2:** (#000) A child sits on a bicycle seat to take it away. (#200) He releases his hand, and the bike turns over the right. (#440) He takes out his right leg and walks a few steps and (#500) falls to the ground. (#640) Then he stretches out his right hand pointing to the lens.

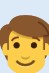 **Human3:** (#000) A boy on the road is riding a small two-wheeled car, (#220) after driving a distance the child stops, the car falls to the ground, (#440) the boy comes down from the car, (#500) he lies on the road. (#600) The little boy lying on the floor strokes his hand and cries.

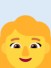 **Human4:** (#000) A child wearing a hat is riding on a baby carriage forward, (#220) and then the car falls, the child stands for a while and (#440) falls off when he crosses his leg out from the car. (#500) The child is lying on the ground and then (#640) pointing to the camera by a finger.

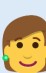 **Human5:** (#000) During the day, a little boy wearing a helmet is riding a bike without pedals, using feet to support forward. (#200) The boy release his hand, the bike tilted down under the boy. (#240) The boy stands and looks down at the bike. (#440) The boy crosses the car and goes to the side and (#500) falls to the ground. (#600) The boy smiles and (#640) reaches out his hand.

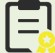 **Groundtruth:** (#000) A young boy is riding a bike down a road. (#220) As he rides, the bike suddenly falls over. (#440) The boy then gets off the bike, (#500) lies on the ground, and pretends to fall. (#640) After a moment, the boy smiles and reaches out his hand.

Figure A7: A detailed example for correspondence between videos, human descriptions, and generated groundtruth.

Fig. A7 illustrates the detailed annotation process for a selected video from the FIOVA dataset, accompanied by annotations from five human annotators and the synthesized groundtruth generated by GPT-3.5-turbo. The upper panel presents sampled frames extracted at 20-frame intervals, capturing key events in the video sequence. The lower panel provides individual descriptions from the five annotators (Human1-Human5), highlighting their observations, followed by the synthesized GT created by integrating these annotations.

The video depicts a young boy riding a bicycle down a road. The boy encounters multiple events, including stopping the bike, falling off, and pretending to fall intentionally. Finally, the boy lies on the ground and points toward the camera. Each human annotator provides a unique perspective while describing the same sequence of events. A detailed comparison of their annotations reveals:

- **Core Event Agreement:** All annotators capture the core sequence of events: riding the bike (#000), stopping (#200), falling off the bike (#440), lying on the ground (#500), and gesturing toward the camera (#640). These observations form the backbone of the GT synthesis process.

- **Diversity in Detail and Focus:** Annotators vary in their descriptions of finer details, such as:
    - **Human1:** Focuses on the boy's playful intent, explicitly mentioning the "pretending to fall" action at #500.
    - **Human3:** Interprets the boy's actions differently, describing him as "stroking his hand and crying" at #600, which contrasts with other annotations.
    - **Human5:** Highlights additional context by describing the boy's method of riding "without pedals" and his subsequent smile and pointing gesture.

This diversity reflects the richness of multi-perspective annotations in capturing both objective events and subjective interpretations.

The groundtruth generated by GPT-3.5-turbo combines the perspectives of the five annotators into a cohesive narrative that captures key events while addressing conflicts in the descriptions:

- **Resolution of Annotation Conflicts:**
    - **"Pretending to Fall":** Human1's explicit mention of "pretending" is corroborated by other annotations, leading to its inclusion in the groundtruth.
    - **"Crying" vs. "Smiling":** Human3 describes the boy as "crying," while Human5 interprets the action as "smiling." Upon integrating contextual information—such as the playful nature of the fall mentioned by Human1 and Human5—the groundtruth concludes that the boy smiles after the fall, aligning with the majority perspective.
- **Maintaining Core Event Coverage:** The groundtruth ensures complete coverage of events, including the boy riding, stopping, falling, lying on the ground, and pointing to the camera.

This case exemplifies the strength of multi-perspective annotation combined with LLM-based synthesis for generating high-quality groundtruth. This process not only captures the complexity of human interpretations but also ensures a unified and accurate representation of video content. The approach highlights the unique advantages of FIOVA in evaluating LVLMs' ability to describe complex, multi-event videos with human-like precision.

# C CALCULATION PROCESS OF COEFFICIENT OF VARIATION (CV)

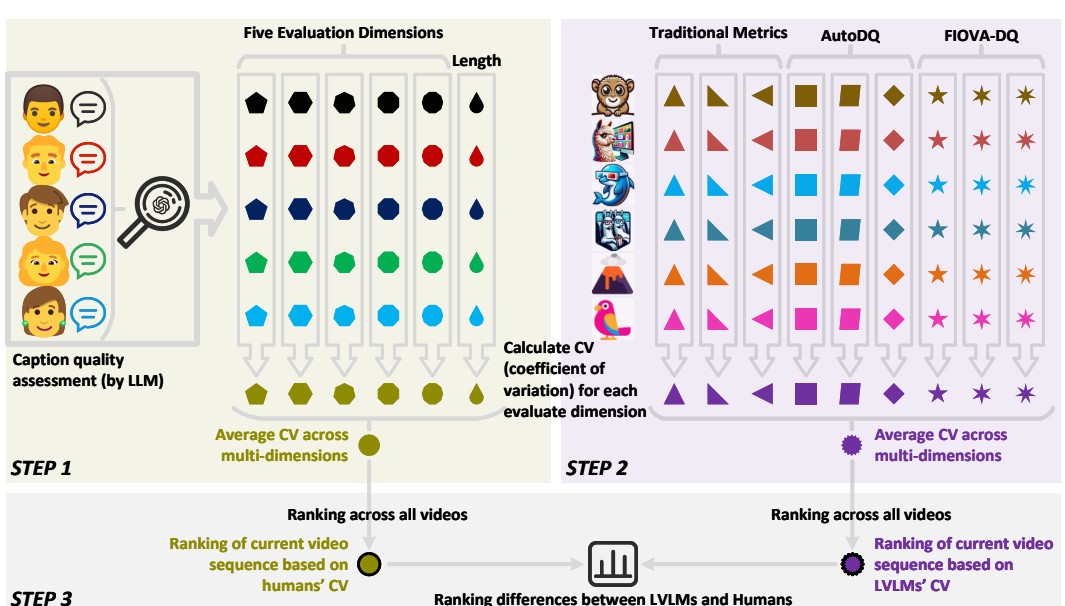

Figure A8: Batch ranking for multi-dimensional consistency and human-machine comparison.

Fig. A8 illustrates the Batch Ranking process used in FIOVA to evaluate video descriptions by comparing human and machine consistency. The process consists of three main steps:

- **Step 1. Human Caption Consistency Evaluation (see Algorithm A1):** The quality of five human-provided captions is assessed across six evaluation dimensions (*i.e.*, Consistency, Context, Correctness, Detail Orientation, Temporality, and Length) using an LLM. The coefficient of variation (CV) is calculated for each dimension to measure the diversity among human descriptions. The average CV across all dimensions determines the overall consistency score for the video, which is used to group videos into different categories (A-H).

- **Step 2. LVLM Consistency Evaluation (see Algorithm A2):** Captions generated by six representative LVLMs are assessed across traditional metrics (*e.g*.., BLEU, GLEU, METEOR), event-level semantic consistency metrics (AutoDQ), and the newly proposed FIOVA-DQ metric. The CV is calculated for each metric across the six models to evaluate their consistency. The average CV provides the overall consistency score for the LVLM group on each video.

- **Step 3. Human-Machine Comparison (see Algorithm A3):** The videos are ranked based on their consistency scores for humans and LVLMs separately. The ranking difference between human annotations and LVLMs provides a quantitative measure of the alignment and divergence in descriptive strategies between humans and machines.

This framework allows for a fine-grained analysis of model performance compared to human benchmarks, revealing the strengths and weaknesses of LVLMs in long video description tasks.

---

**Algorithm A1** Framework for CV calculation between humans

---

**Input:** $D = \{(V_1, C_1), \ldots, (V_n, C_n)\}$: FIOVA dataset;
$C_i = \{c_{i1}, c_{i2}, c_{i3}, c_{i4}, c_{i5}\}$: human annotations for video $V_i$;
$E = \{\text{Consistency}, \text{Context}, \text{Correctness}, \text{Detail Orientation}, \text{Temporality}, \text{Length}\}$: evaluation dimensions;
**Output:** $CV_{dimension}^{human}$: Dictionary of coefficient of variation between humans for each evaluation dimension;
         dimension;
$CV_{video}^{human}$: Dictionary of mean coefficient of variation between humans for each video;
$Intervals$: Dictionary of intervals dividing $CV_{video}^{human}$

    /* Step 1: Calculate CV for each dimension                                   */
**1** Initialize $CV_{dimension}^{human} \leftarrow \{\}$               // Dictionary to store CV for each dimension
**2** **for** $d \leftarrow 1$ **to** $|E|$ **do**
**3**     Initialize $CV_{E[d]} \leftarrow \{\}$        // Dictionary to store CV for each video in dimension $E[d]$
**4**     **for** $i \leftarrow 1$ **to** $|D|$ **do**
**5**        Initialize scores list $S_i \leftarrow []$
         **for** $j \leftarrow 1$ **to** $|C_i|$ **do**
**6**             $s_{ij} \leftarrow$ score of $c_{ij}$ in $E[d]$
             Append $s_{ij}$ to $S_i$
**7**        Calculate mean $\mu_i$ of $S_i$
         Calculate standard deviation $\sigma_i$ of $S_i$
         Calculate coefficient of variation $cv_i \leftarrow \frac{\sigma_i}{\mu_i}$
         $CV_{E[d]}[i] \leftarrow cv_i$                           // Store CV for video $V_i$
**8**     $CV_{dimension}^{human}[E[d]] \leftarrow CV_{E[d]}$

    /* Step 2: Calculate mean CV for each video                               */
**9** Initialize $CV_{video}^{human} \leftarrow \{\}$             // Dictionary to store mean CV for each video
**10** **for** $i \leftarrow 1$ **to** $|D|$ **do**
**11**     Initialize sum of CVs $sum_{CV} \leftarrow 0$
       **for** $d \leftarrow 1$ **to** $|E|$ **do**
**12**        $sum_{CV} \leftarrow sum_{CV} + CV_{dimension}^{human}[E[d]][i]$
**13**     Calculate mean $mean_{CV} \leftarrow \frac{sum_{CV}}{|E|}$
       $CV_{video}^{human}[i] \leftarrow mean_{CV}$                  // Store mean CV for video $V_i$

    /* Step 3: Divide $CV_{video}^{human}$ into intervals based on the maximum value         */
**14** Sort $CV_{video}^{human}$ in ascending order by value and store sorted keys as $sorted\_keys$
    Calculate $max\_CV \leftarrow \max(CV_{video}^{human}.values())$
    Calculate number of intervals $N \leftarrow \lceil max\_CV \times 10 \rceil$         // Each interval represents 10%
**15** Initialize $Intervals \leftarrow \{\}$         // Dictionary to store interval information for each video
**16** **for** $i \leftarrow 1$ **to** $|sorted\_keys|$ **do**
**17**     $video\_id \leftarrow sorted\_keys[i]$
       $cv \leftarrow CV_{video}^{human}[video\_id]$
       Calculate interval index $index \leftarrow \lfloor cv \times 10 \rfloor$
       **if** $index \geq N$ **then**
**18**        $index \leftarrow N - 1$
**19**     $Intervals[video\_id] \leftarrow index$                  // Store interval for video $V_i$
**20** **return** $CV_{dimension}^{human}, CV_{video}^{human}, Intervals$

---

1566
1567
1568
1569
1570
1571
1572
1573
1574
1575
1576
1577
1578
1579

---

**Algorithm A2** Framework for CV calculation between LVLMs

---

**Input:** $D = \{(V_1, R_1), \ldots, (V_m, R_m)\}$: FIOVA dataset;
$R_i = \{r_{i1}, r_{i2}, \ldots, r_{in}\}$: LVLMs' responses for video $V_i$;
$E = \{\text{F1}, \text{Recall}, \text{Precision}, \text{BLEU}, \text{METEOR}, \text{GLEU}\}$: evaluation dimensions;
**Output:** $CV^{lvlm}_{dimension}$: Dictionary of coefficient of variation for each evaluation dimension;
$CV^{lvlm}_{video}$: Dictionary of mean coefficient of variation between LVLMs for each video;

```
/* Step 1:  Calculate CV for each dimension                                    */
```
21 Initialize $CV^{lvlm}_{dimension} \leftarrow \{\}$                          `// Dictionary to store CV for each dimension`
22 **for** $d \leftarrow 1$ **to** $|E|$ **do**
23     Initialize $CV_{E[d]} \leftarrow \{\}$                 `// Dictionary to store CV for each video in dimension E[d]`
24     **for** $i \leftarrow 1$ **to** $|D|$ **do**
25        Initialize scores list $S_i \leftarrow []$
       **for** $j \leftarrow 1$ **to** $|R_i|$ **do**
26           $s_{ij} \leftarrow$ score of $r_{ij}$ in $E[d]$
          Append $s_{ij}$ to $S_i$
27        Calculate mean $\mu_i$ of $S_i$
       Calculate standard deviation $\sigma_i$ of $S_i$
       Calculate coefficient of variation $cv_i \leftarrow \frac{\sigma_i}{\mu_i}$
       $CV_{E[d]}[i] \leftarrow cv_i$                         `// Store CV for video Vi`
28     $CV^{lvlm}_{dimension}[E[d]] \leftarrow CV_{E[d]}$
```
/* Step 2:  Calculate mean CV for each video                                   */
```
29 Initialize $CV^{lvlm}_{video} \leftarrow \{\}$                              `// Dictionary to store mean CV for each video`
30 **for** $i \leftarrow 1$ **to** $|D|$ **do**
31     Initialize sum of CVs $sum_{CV} \leftarrow 0$
    **for** $d \leftarrow 1$ **to** $|E|$ **do**
32        $sum_{CV} \leftarrow sum_{CV} + CV^{lvlm}_{dimension}[E[d]][i]$
33     Calculate mean $mean_{CV} \leftarrow \frac{sum_{CV}}{|E|}$
    $CV^{lvlm}_{video}[i] \leftarrow mean_{CV}$                      `// Store mean CV for video Vi`
34 **return** $CV^{lvlm}_{dimension}, CV^{lvlm}_{video}$

---

1608
1609
1610
1611
1612
1613
1614
1615
1616
1617
1618
1619

---

**Algorithm A3** Calculate ranking differences

---

**Input:** $CV_{video}^{lvlm}$: Dictionary of mean coefficient of variation between LVLMs for each video;
$CV_{video}^{human}$: Dictionary of mean coefficient of variation between humans for each video;
**Output:** $Rankings^{human}$: Dictionary of rankings based on humans' CV;
$Rankings^{lvlm}$: Dictionary of rankings based on LVLMs' CV;
$Rankings^{diff}$: Dictionary of difference between $Rankings^{human}$ and $Rankings^{lvlm}$;

/* Step 1:  Rank videos based on $CV_{video}^{human}$ and $CV_{video}^{lvlm}$                                                      */

35  Sort $CV_{video}^{human}$ by value in ascending order and store the sorted video IDs as $sorted\_ids^{human}$

    // Ranking by CV values from smallest to largest

36  Sort $CV_{video}^{lvlm}$ by value in ascending order and store the sorted video IDs as $sorted\_ids^{lvlm}$

    // Ranking by CV values from smallest to largest

37  Initialize $Rankings^{human} \leftarrow \{\}$                                              // Dictionary to store human rankings

38  Initialize $Rankings^{lvlm} \leftarrow \{\}$                                                 // Dictionary to store LVLM rankings

39  **for** $rank \leftarrow 1$ **to** $|sorted\_ids^{human}|$ **do**

40     $video\_id \leftarrow sorted\_ids^{human}[rank]$

     $Rankings^{human}[video\_id] \leftarrow rank$

41  **for** $rank \leftarrow 1$ **to** $|sorted\_ids^{lvlm}|$ **do**

42     $video\_id \leftarrow sorted\_ids^{lvlm}[rank]$

     $Rankings^{lvlm}[video\_id] \leftarrow rank$

/* Step 2:  Calculate difference between rankings                                                      */

43  Initialize $Rankings^{diff} \leftarrow \{\}$                                        // Dictionary to store ranking differences

44  **foreach** $video\_id \in CV_{video}^{human}.keys()$ **do**

45     $Rankings^{diff}[video\_id] \leftarrow |Rankings^{human}[video\_id] - Rankings^{lvlm}[video\_id]|$

46  **return** $Rankings^{human}, Rankings^{lvlm}, Rankings^{diff}$

---

# D PROMPTS

## D.1 GPT-AIDED EVALUATION PROMPTS

### D.1.1 PROMPT FOR EVALUATION OF HUMAN ANNOTATIONS

---

**The Prompt for Consistency of Annotation (by GPT).**

**Prompt**

You are an intelligent chatbot designed for evaluating the factual accuracy of generative outputs for video-based caption. Your task is to compare the provided text and determine if they are factually consistent. Here's how you can accomplish the task:
——

##INSTRUCTIONS:
- Focus on the consistency of the text with the expected content or background. The text should correspond to the correct information and should not contain any contradictions or significant differences.
- The text must be consistent in the information it provides about the content.
- Consider synonyms or paraphrases as valid matches, but only if they maintain the consistency in the conveyed information.
- Evaluate the consistency of the text.
- DO NOT PROVIDE ANY OTHER OUTPUT TEXT OR EXPLANATION. Only provide a single evaluation score from 1 to 10. For example, your response should look like this: {"score": [score]}.

- - - - - - - - - - - - - - - - - - - - - - - - - - - - - - - - - - - - - - - - -

**User:**
Please evaluate the following video caption:
Provided caption: **"{Caption}"**
DO NOT PROVIDE ANY OTHER OUTPUT TEXT OR EXPLANATION. Only provide a single evaluation score from 1 to 10. For example, your response should look like this: {"score": [score]}.

---

**The Prompt for Context of Annotation (by GPT).**

**Prompt**

You are an intelligent chatbot designed for evaluating the factual accuracy of generative outputs for video-based caption. Your task is to compare the provided text and determine if they are factually consistent. Here's how you can accomplish the task:
——

##INSTRUCTIONS:
- Evaluate whether the text aligns with the overall context of the expected content or background. It should not provide information that is out of context or misaligned.
- The text must capture the main themes and sentiments relevant to the content.
- Consider synonyms or paraphrases as valid matches.
- Provide your evaluation of the contextual understanding of the text.
DO NOT PROVIDE ANY OTHER OUTPUT TEXT OR EXPLANATION. Only provide a single evaluation score from 1 to 10. For example, your response should look like this: {"score": [score]}.

- - - - - - - - - - - - - - - - - - - - - - - - - - - - - - - - - - - - - - - - -

**User:**
Please evaluate the following video caption:
Provided caption: **"{Caption}"**
DO NOT PROVIDE ANY OTHER OUTPUT TEXT OR EXPLANATION. Only provide a single evaluation score from 1 to 10. For example, your response should look like this: {"score": [score]}.

---

**The Prompt for Correctness of Annotation (by GPT).**

**Prompt**
You are an intelligent chatbot designed for evaluating the factual accuracy of generative outputs for video-based caption. Your task is to compare the provided text and determine if they are factually consistent. Here's how you can accomplish the task:
——
##INSTRUCTIONS:
- Focus on the factual correctness of the text. The text should not contain any misinterpretations or misinformation.
- The text must be factually accurate and align with the expected content or context.
- Consider synonyms or paraphrases as valid matches.
- Evaluate the factual accuracy of the text.
DO NOT PROVIDE ANY OTHER OUTPUT TEXT OR EXPLANATION. Only provide a single evaluation score from 1 to 10. For example, your response should look like this: {"score": [score]}.
- - - - - - - - - - - - - - - - - - - - - - - - - - - - - - - - - - - - - - - - - - - - - -
**User:**
Please evaluate the following video caption:
Provided caption: **"{Caption}"**
DO NOT PROVIDE ANY OTHER OUTPUT TEXT OR EXPLANATION. Only provide a single evaluation score from 1 to 10. For example, your response should look like this: {"score": [score]}.

**The Prompt for Detailed Orientation of Annotation (by GPT).**

**Prompt**
You are an intelligent chatbot designed for evaluating the factual accuracy of generative outputs for video-based caption. Your task is to compare the provided text and determine if they are factually consistent. Here's how you can accomplish the task:
——
##INSTRUCTIONS:
- Check if the text covers all major points relevant to the content. The text should not leave out any key aspects.
- Evaluate whether the text includes specific details rather than just generic points. It should provide comprehensive information that is tied to specific elements of the content.
- Consider synonyms or paraphrases as valid matches.
- Provide a single evaluation score that reflects the level of detail orientation of the text, considering both completeness and specificity.
DO NOT PROVIDE ANY OTHER OUTPUT TEXT OR EXPLANATION. Only provide a single evaluation score from 1 to 10. For example, your response should look like this: {"score": [score]}.
- - - - - - - - - - - - - - - - - - - - - - - - - - - - - - - - - - - - - - - - - - - - - -
**User:**
Please evaluate the following video caption:
Provided caption: **"{Caption}"**
DO NOT PROVIDE ANY OTHER OUTPUT TEXT OR EXPLANATION. Only provide a single evaluation score from 1 to 10. For example, your response should look like this: {"score": [score]}.

**The Prompt for Temporality of Annotation (by GPT).**

**Prompt**
You are an intelligent chatbot designed for evaluating the factual accuracy of generative outputs for video-based caption. Your task is to compare the provided text and determine if they are factually consistent. Here's how you can accomplish the task:

‾‾
##INSTRUCTIONS:
- Focus on the temporal consistency of the text. It should correctly reflect the sequence of events or details as they are presented.
- Consider synonyms or paraphrases as valid matches, but only if the temporal order is maintained.
- Evaluate the temporal accuracy of the text.
DO NOT PROVIDE ANY OTHER OUTPUT TEXT OR EXPLANATION. Only provide a single evaluation score from 1 to 10. For example, your response should look like this: {"score": [score]}.

- - - - - - - - - - - - - - - - - - - - - - - - - - - - - - - - - - - - - - - - - - - - - - - - - -

**User:**
Please evaluate the following video caption:
Provided caption: **"{Caption}"**
DO NOT PROVIDE ANY OTHER OUTPUT TEXT OR EXPLANATION. Only provide a single evaluation score from 1 to 10. For example, your response should look like this: {"score": [score]}.

---

An Example of Evaluation Consistency of Annotation (by GPT).

**Prompt:** You are an intelligent chatbot designed for evaluating the factual accuracy of generative outputs for video-based caption. Your task is to compare the provided text and determine if they are factually consistent. Here's how you can accomplish the task:
‾‾
##INSTRUCTIONS:
- Focus on the consistency of the text with the expected content or background. The text should correspond to the correct information and should not contain any contradictions or significant differences.
- The text must be consistent in the information it provides about the content.
- Consider synonyms or paraphrases as valid matches, but only if they maintain the consistency in the conveyed information.
- Evaluate the consistency of the text.
  - DO NOT PROVIDE ANY OTHER OUTPUT TEXT OR EXPLANATION. Only provide a single evaluation score from 1 to 10. For example, your response should look like this: {"score": [score]}.

- - - - - - - - - - - - - - - - - - - - - - - - - - - - - - - - - - - - - - - - - - - - - - - - - -

**User:**
Please evaluate the following video caption:
Provided caption: **"A silver car is running slowly on the road. A red motorcycle running beside crashed on a red car. The motorcycle is overturned. The following vehicles began to slow down. A man in a blue coat riding a motorcycle parked next to the fallen motorcycle. From the red car two people came out. The slow motion of the vedio re-played."**
DO NOT PROVIDE ANY OTHER OUTPUT TEXT OR EXPLANATION. Only provide a single evaluation score from 1 to 10. For example, your response should look like this: {"score": [score]}.
**GPT:** {"score": 3}

---

D.1.2 PROMPT FOR GROUNDTRUTH GENERATION

The Prompt for Groundtruth Generation (by GPT).

**Prompt**
Given five video descriptions. Combine the five video descriptions into a single, coherent description that captures the essence of the video clip.

Please generate the response in the form of a Python dictionary string with keys "gt". The value of "gt" is a List(str), of which is groundtruth for this video description.

- - - - - - - - - - - - - - - - - - - - - - - - - - - - - - - - - - - - - - - - - - - - - - -

**User:**
Video description 1: **"{Caption1}"**
Video description 2: **"{Caption2}"**
Video description 3: **"{Caption3}"**
Video description 4: **"{Caption4}"**
Video description 5: **"{Caption5}"**
DO NOT PROVIDE ANY OTHER OUTPUT TEXT OR EXPLANATION. Only provide one Python dictionary string. For example, your response should look like this: {"gt": [gt]}

---

### An Example of Groundtruth Generation (by GPT).

**Prompt:** Given five video descriptions. Combine the five video descriptions into a single, coherent description that captures the essence of the video clip.
Please generate the response in the form of a Python dictionary string with keys "gt". The value of "gt" is a List(str), of which is groundtruth for this video description.

- - - - - - - - - - - - - - - - - - - - - - - - - - - - - - - - - - - - - - - - - - - - - - -

**User:**
Video description 1: **"A silver car is running slowly on the road. A red motorcycle running beside crashed on a red car. The motorcycle is overturned. The following vehicles began to slow down. A man in a blue coat riding a motorcycle parked next to the fallen motorcycle. From the red car two people came out. The slow motion of the vedio re-played."**
Video description 2: **"A silver business car on the side of the road is driving, followed by a motorcycle, and a red car in front of a commercial car changing lanes crashed on a motorcycle, and the man on the motorcycle fell to the ground. A white van in the following stopped. The motorcycle driver in red stood up and the following motorcycle stoped, the man in blue came down to help the blue men pick up the package. From the red car two men and one women waled to the man in red. The crash process is re played back in slow motion. "**
Video description 3: **"The vehicle on the road was running normally. A red car crashed into the motorcycle next to it when changing lanes. The motorcycle was knocked down and the two men on the motorcycle fell. A motorcycle that passed by stopped to inquire about the situation. The two men on the motorcycle stood up, and the red car stopped. From the car down two men and a woman came. The crash scene is replayed again. "**
Video description 4: **"On the highway cars were driving normal. A motorcycle when driving hit a car which drove from the side lane to middle lane. The motorcycle driver fell in from of a van and several motocycles passed by. A motorcycle driver helped the man stand on the ground. A few people from the crashed car came over to check, and then slow down the crash video. "**
Video description 5: **"In the daytime, a front silver hatchback in the right side of the road was running slowly. On the left side a motorcycle quickly passed, with two people. The motorcycle passed the silver car, which collided with a red car in front which was turning left, and the car was knocked to the ground and then climbed up. Rear vehicles stoped. From the rear three motorcycles came, one of which stopped next to the one who was knocked over. From the red car a woman and three men came out, and went to the people who was hit. Slow motion played back the accident moment. "**
DO NOT PROVIDE ANY OTHER OUTPUT TEXT OR EXPLANATION. Only provide one Python dictionary string. For example, your response should look like this: {"gt": [gt]}
**GPT:** { "gt": [ "A silver car and a red motorcycle are driving on the road. The red car in front of the silver car changes lanes and collides with the motorcycle, causing it to crash. The motorcycle driver falls to the ground. A white van and several motorcycles behind stop. A man in a blue coat riding a motorcycle stops to help the fallen motorcycle driver. Two men and a woman from the red car come out to check on the situation. The crash scene is then replayed in slow motion." ] }

### D.1.3 PROMPT FOR EVENT EXTRACTION

---

**The Prompt for Event Extraction (by GPT).**

**Prompt**

Given a video description. Extract at most 10 key events from the video description paragraph. Requirements:

- Every event is represented by a brief sentence within 10 words, with a subject, a predicate and optionally an object,avoid unnecessary appearance descriptions.
- Every event must be atomic, meaning that it can not be further split into multiple events.
- Scene cuts and camera motions are NOT events.
- Substitute pronouns by the nouns they refer to.

Please generate the response in the form of a Python dictionary string with keys "events". The value of "events" is a List(str), of which each item is an event.

- - - - - - - - - - - - - - - - - - - - - - - - - - - - - - - - - - - - - - - - - - - - - - - - - - - - - - - -

**User:**

Video description: **"{Caption}"**

DO NOT PROVIDE ANY OTHER OUTPUT TEXT OR EXPLANATION. Only provide the Python dictionary string. For example, your response should look like this: {"events": [event1, event2,...]}

---

**An Example of Event Extraction (by GPT).**

**Prompt:**

Given a video description. Extract at most 10 key events from the video description paragraph. Requirements:

- Every event is represented by a brief sentence within 10 words, with a subject, a predicate and optionally an object,avoid unnecessary appearance descriptions.
- Every event must be atomic, meaning that it can not be further split into multiple events.
- Scene cuts and camera motions are NOT events.
- Substitute pronouns by the nouns they refer to.

 Please generate the response in the form of a Python dictionary string with keys "events". The value of "events" is a List(str), of which each item is an event.

- - - - - - - - - - - - - - - - - - - - - - - - - - - - - - - - - - - - - - - - - - - - - - - - - - - - - - - -

**User:**

Video description: **A silver car and a red motorcycle are driving on the road. The red car in front of the silver car changes lanes and collides with the motorcycle, causing it to crash. The motorcycle driver falls to the ground. A white van and several motorcycles behind stop. A man in a blue coat riding a motorcycle stops to help the fallen motorcycle driver. Two men and a woman from the red car come out to check on the situation. The crash scene is then replayed in slow motion.**

DO NOT PROVIDE ANY OTHER OUTPUT TEXT OR EXPLANATION. Only provide the Python dictionary string. For example, your response should look like this: {"events": [event1, event2,...]}

**GPT:**

{ "events": [
"Silver car and red motorcycle drive on road",
"Red car changes lanes and collides with motorcycle",
"Motorcycle driver falls to ground",
"White van and motorcycles behind stop",
"Man in blue coat riding motorcycle helps driver",
"Men and woman from red car check situation",
"Crash scene replayed in slow motion" ]
}

### D.1.4 PROMPT FOR CROSS CHECKING

---

**The Prompt for Cross Checking (by GPT).**

**Prompt**

Given a video description and a list of events. For each event, classify the relationship between the video description and the event into three classes: entailment, neutral, contradiction.
- "entailment" means that the video description entails the event.
- "contradiction" means that some detail in the video description contradicts with the event.
- "neutral" means that the relationship is neither "entailment" or "contradiction".
Output a list in Json format: [ {"event": "copy an event here", "relationship": "put class name here" }, ... ].

- - - - - - - - - - - - - - - - - - - - - - - - - - - - - - - - - - - - - - - - - - - - - - - - - - - - - - -

**User:**
Video description: **"{Caption}"**
Events: **"{Event}"**
DO NOT PROVIDE ANY OTHER OUTPUT TEXT OR EXPLANATION. Only output the JSON. Output:

---

**An Example of Cross Checking (by GPT).**

**Prompt:**
Given a video description and a list of events. For each event, classify the relationship between the video description and the event into three classes: entailment, neutral, contradiction.
- "entailment" means that the video description entails the event.
- "contradiction" means that some detail in the video description contradicts with the event.
- "neutral" means that the relationship is neither "entailment" or "contradiction".
 Output a list in Json format: [ {"event": "copy an event here", "relationship": "put class name here" }, ... ].

- - - - - - - - - - - - - - - - - - - - - - - - - - - - - - - - - - - - - - - - - - - - - - - - - - - - - - -

**User:**
Video description: **The video clip shows a road scene in Singapore. It begins with a view of a wet road, possibly from rain. The scene then transitions to a road with moving vehicles, including cars and buses. A silver car is prominently featured in the center, moving slowly due to traffic. Motorcycles and other vehicles are seen passing by the car. On the right side of the road, there is a group of people who appear to be involved in an incident or accident. The background consists of trees and roadside barriers.**
Events:
**{ "events": [**
**"Silver car and red motorcycle drive on road",**
**"Red car changes lanes and collides with motorcycle",**
**"Motorcycle driver falls to ground",**
**"White van and motorcycles behind stop",**
**"Man in blue coat riding motorcycle helps driver",**
**"Men and woman from red car check situation",**
**"Crash scene replayed in slow motion" ]**
**}**
DO NOT PROVIDE ANY OTHER OUTPUT TEXT OR EXPLANATION. Only output the JSON. Output:
**GPT:**
{ "event": "Silver car and red motorcycle drive on road", "relationship": "entailment" },
{ "event": "Red car changes lanes and collides with motorcycle", "relationship": "neutral" },
{ "event": "Motorcycle driver falls to ground", "relationship": "entailment" },
{ "event": "White van and motorcycles behind stop", "relationship": "neutral" },
{ "event": "Man in blue coat riding motorcycle helps drive", "relationship": "contradiction" },

{ "event": "Men and woman from red car check situation", "relationship": "entailment" },
{ "event": "Crash scene replayed in slow motion", "relationship": "contradiction" }

## D.2 VIDEO CAPTION PROMPTS

We use the video description prompt provided by the official repository. If there is no official prompt, we will use "Describe the video in details." as a standard prompt.

Table A2: The URLs of official LVLMs repository in this work.

| LVLMs | Main Creator | #Parameters | Access |
|---|---|---|---|
| **VideoLLaMA2** (Cheng et al. (2024)) | Alibaba Group | 7B | Weights |
| **Video-LLaVA** (Lin et al. (2023)) | Peking University | 7B | Weights |
| **ShareGPT4Video** (Chen et al. (2024a)) | University of Science and Technology of China | 8B | Weights |
| **Tarsier** (Wang et al. (2024a)) | ByteDance | 7B | Weights |
| **VideoChat2** (Li et al. (2023)) | Shanghai AI Laboratory | 7B | Weights |
| **LLaVA-NEXT-Video** (Zhang et al. (2024)) | ByteDance | 7B | Weights |

> **The Prompt for VideoLLaMA2, Video-LLaVA, ShareGPT4Video, Tarsier, and VideoChat2.**
>
> Describe the video in details.

> **The Prompt for LLaVA-NEXT-Video.**
>
> Please provide a detailed description of the video, focusing on the main subjects, their actions, and the background scenes.

Along with the prompt, we opted to use 8 frames per video as the input data. This decision was made to balance evaluation efficiency and information capture, aligning with the standard experimental paradigms in the current field of video tasks. The details are as follows:

- **Consistency with Experimental Paradigm:** FIOVA is designed to provide an open and high-quality evaluation benchmark for long-video description tasks, enabling comparisons of LVLM performance and their differences from human annotators. To ensure reproducibility and scalability, our experimental setup (including frame selection) followed the widely adopted fixed-frame sampling strategy in the video understanding field. This choice facilitates horizontal comparisons with existing works and offers a reference framework for future research.

- **Methodological Generality:** The number of input frames is a critical factor in long-video tasks. Selecting 8 frames balances computational cost and semantic capture, enabling effective performance evaluation. This strategy has been validated in many related works, such as VideoGPT+ Maaz et al. (2024) and Emu-3 Wang et al. (2024b), which also adopt 8 frames as input. These examples highlight the representativeness of this setup for long-video understanding tasks. Additionally, current LVLMs typically face constraints on the number of input frames; too many frames could lead to resource limitations or performance degradation. The 8-frame setup is well-suited to the computational capabilities of mainstream LVLMs while avoiding information redundancy.

- **Fairness and Feasibility of the Evaluation Platform:** All experimental results in our study are based on the 8-frame setup. This configuration validates FIOVA's evaluation capability while ensuring fairness and feasibility. The selection of 8 frames strikes a balance among semantic capture, experimental efficiency, and model constraints, making it a reasonable setting aligned with the standard experimental paradigms in video tasks.

Although this study adopts the 8-frame setup, the FIOVA benchmark is designed with flexibility for expansion. Researchers can adjust the frame sampling strategy according to specific research needs, further exploring LVLMs' potential in complex long-video tasks. We also plan to open frame-setting options in future studies to support diversified experimental designs.

# E  DETAILED EXPERIMENTAL RESULTS

## E.1  LVLMs v.s. Humans on traditional metrics

Table A3: Comparison of LVLMs and Humans on FIOVA based on traditional metrics (BLEU, METEOR, and GLEU). The background color represents the performance of the metric. The darker the green, the better the performance.

| Metrics | LVLMs | Human1 | Human2 | Human3 | Human4 | Human5 | GT |
|---|---|---|---|---|---|---|---|
| **BLEU (↑)** | Tarsier | 0.025 | 0.025 | 0.024 | 0.025 | 0.024 | 0.043 |
| | VideoLLaMA2 | 0.018 | 0.019 | 0.018 | 0.018 | 0.018 | 0.030 |
| | LLaVA-NEXT-Video | 0.013 | 0.014 | 0.014 | 0.014 | 0.013 | 0.020 |
| | Video-LLaVA | 0.017 | 0.019 | 0.018 | 0.018 | 0.017 | 0.027 |
| | ShareGPT4Video | 0.006 | 0.007 | 0.006 | 0.006 | 0.006 | 0.010 |
| | VideoChat2 | 0.021 | 0.024 | 0.023 | 0.022 | 0.022 | 0.037 |
| **METEOR (↑)** | Tarsier | 0.232 | 0.232 | 0.229 | 0.230 | 0.231 | 0.265 |
| | VideoLLaMA2 | 0.245 | 0.248 | 0.246 | 0.247 | 0.247 | 0.268 |
| | LLaVA-NEXT-Video | 0.246 | 0.249 | 0.248 | 0.249 | 0.247 | 0.270 |
| | Video-LLaVA | 0.238 | 0.242 | 0.240 | 0.240 | 0.240 | 0.257 |
| | ShareGPT4Video | 0.194 | 0.196 | 0.197 | 0.195 | 0.192 | 0.218 |
| | VideoChat2 | 0.256 | 0.260 | 0.257 | 0.258 | 0.258 | 0.281 |
| **GLEU (↑)** | Tarsier | 0.091 | 0.092 | 0.090 | 0.091 | 0.090 | 0.119 |
| | VideoLLaMA2 | 0.068 | 0.071 | 0.070 | 0.069 | 0.068 | 0.088 |
| | LLaVA-NEXT-Video | 0.047 | 0.049 | 0.049 | 0.048 | 0.047 | 0.060 |
| | Video-LLaVA | 0.061 | 0.063 | 0.063 | 0.062 | 0.061 | 0.077 |
| | ShareGPT4Video | 0.027 | 0.028 | 0.027 | 0.027 | 0.026 | 0.034 |
| | VideoChat2 | 0.075 | 0.078 | 0.078 | 0.077 | 0.076 | 0.098 |

In Table A3, it is observed that comparing model outputs with GPT-summarized human captions (aggregated GT) results in higher metric scores than directly comparing model outputs with single human captions. Below, we provide an analysis and explanation for this phenomenon:

**Improved Information Coverage by GPT-Summarized Descriptions.** Each video in the FIOVA dataset is annotated by five independent annotators who watched the full video before providing detailed descriptions. Due to their differing focuses, each annotator's description may emphasize various aspects, such as:

- **Action Details:** Certain annotators might prioritize characters' actions and their sequences.
- **Contextual Information:** Others may focus on the environment, background, or secondary events.

GPT-3.5-turbo aggregates these descriptions, effectively integrating multi-perspective information from all five annotators into a comprehensive and diverse GT. By synthesizing multiple viewpoints, the aggregated GT captures a broader spectrum of video content, ensuring improved coverage compared to single human descriptions. For instance, as shown in Fig. A7, certain annotators emphasize the actions of a child, while others document background details. The aggregation process ensures that both types of information are represented in the GT, enhancing its overall comprehensiveness.

**Reasons for Higher Metric Scores.** The higher scores observed when comparing model outputs with aggregated GT can be attributed to two main factors:

- **Broader Alignment Possibility:** The aggregated GT encompasses richer and more diverse content, making it easier for model outputs to align with various aspects of the GT. Consequently:
  - Model outputs are more likely to match specific details captured by at least one annotator.
  - The inclusion of diverse content reduces the chance of missing critical information, resulting in improved BLEU and METEOR scores.

- **Limitations of Single Annotator Descriptions:** Single annotators may provide descriptions that focus on limited aspects of a video, potentially omitting significant details. When compared to model outputs, such descriptions may highlight perceived gaps in coverage, leading to relatively lower metric scores.

By integrating multi-perspective annotations, aggregated GT provides a richer, more comprehensive reference for evaluation, ensuring fairness and alignment with FIOVA's design principles. This strategy not only enhances the reliability of metric-based evaluations but also supports future research in thoroughly assessing model performance. Aggregated GT will continue to serve as a critical component of FIOVA for evaluating LVLMs in long-video understanding tasks.

## E.2 RESULTS ON DIFFERENT GROUPS

Table A4: Comparison of LVLMs on FIOVA based on traditional metrics (BLEU, METEOR, and GLEU), AutoDQ-based metrics, and FIOVA-DQ. The background color represents the performance of the metric. The darker the green, the better the performance.

| Metrics | LVLMs | Group | | | | | | | | All |
|---|---|---|---|---|---|---|---|---|---|---|
| | | A | B | C | D | E | F | G | H | |
| BLEU (↑) | Tarsier | 0.058 | 0.044 | 0.041 | 0.042 | 0.045 | 0.038 | 0.052 | 0.043 | 0.043 |
| | VIdeoLLaMA2 | 0.028 | 0.031 | 0.030 | 0.030 | 0.030 | 0.026 | 0.024 | 0.024 | 0.030 |
| | LLaVA-NEXT-Video | 0.023 | 0.020 | 0.020 | 0.020 | 0.021 | 0.019 | 0.024 | 0.013 | 0.020 |
| | Video-LLaVA | 0.026 | 0.028 | 0.027 | 0.028 | 0.026 | 0.022 | 0.024 | 0.020 | 0.027 |
| | ShareGPT4Video | 0.014 | 0.011 | 0.011 | 0.010 | 0.010 | 0.008 | 0.010 | 0.010 | 0.010 |
| | VideoChat2 | 0.041 | 0.037 | 0.036 | 0.037 | 0.036 | 0.031 | 0.037 | 0.028 | 0.037 |
| METEOR (↑) | Tarsier | 0.288 | 0.267 | 0.263 | 0.265 | 0.265 | 0.255 | 0.264 | 0.288 | 0.265 |
| | VIdeoLLaMA2 | 0.278 | 0.271 | 0.267 | 0.269 | 0.265 | 0.260 | 0.255 | 0.260 | 0.268 |
| | LLaVA-NEXT-Video | 0.277 | 0.272 | 0.271 | 0.268 | 0.267 | 0.263 | 0.264 | 0.274 | 0.270 |
| | Video-LLaVA | 0.265 | 0.262 | 0.255 | 0.260 | 0.249 | 0.241 | 0.246 | 0.229 | 0.257 |
| | ShareGPT4Video | 0.244 | 0.221 | 0.219 | 0.213 | 0.215 | 0.208 | 0.215 | 0.223 | 0.218 |
| | VideoChat2 | 0.289 | 0.286 | 0.279 | 0.281 | 0.277 | 0.267 | 0.272 | 0.269 | 0.281 |
| GLEU (↑) | Tarsier | 0.139 | 0.120 | 0.117 | 0.118 | 0.119 | 0.113 | 0.124 | 0.137 | 0.119 |
| | VIdeoLLaMA2 | 0.086 | 0.088 | 0.089 | 0.087 | 0.087 | 0.085 | 0.084 | 0.089 | 0.088 |
| | LLaVA-NEXT-Video | 0.062 | 0.059 | 0.060 | 0.059 | 0.062 | 0.060 | 0.071 | 0.063 | 0.060 |
| | Video-LLaVA | 0.078 | 0.077 | 0.076 | 0.078 | 0.076 | 0.072 | 0.076 | 0.066 | 0.077 |
| | ShareGPT4Video | 0.041 | 0.035 | 0.035 | 0.034 | 0.033 | 0.030 | 0.034 | 0.037 | 0.034 |
| | VideoChat2 | 0.106 | 0.098 | 0.098 | 0.098 | 0.098 | 0.093 | 0.103 | 0.101 | 0.098 |
| F1 (AutoDQ) (↑) | Tarsier | 0.366 | 0.346 | 0.350 | 0.359 | 0.350 | 0.355 | 0.329 | 0.324 | 0.351 |
| | VIdeoLLaMA2 | 0.346 | 0.328 | 0.316 | 0.332 | 0.325 | 0.324 | 0.304 | 0.285 | 0.325 |
| | LLaVA-NEXT-Video | 0.322 | 0.297 | 0.302 | 0.302 | 0.304 | 0.302 | 0.284 | 0.268 | 0.301 |
| | Video-LLaVA | 0.304 | 0.283 | 0.282 | 0.287 | 0.288 | 0.292 | 0.265 | 0.331 | 0.285 |
| | ShareGPT4Video | 0.277 | 0.276 | 0.274 | 0.295 | 0.285 | 0.279 | 0.306 | 0.320 | 0.281 |
| | VideoChat2 | 0.315 | 0.315 | 0.303 | 0.318 | 0.301 | 0.297 | 0.255 | 0.160 | 0.309 |
| Recall (AutoDQ) (↑) | Tarsier | 0.333 | 0.305 | 0.279 | 0.280 | 0.265 | 0.226 | 0.212 | 0.193 | 0.283 |
| | VIdeoLLaMA2 | 0.286 | 0.268 | 0.243 | 0.242 | 0.222 | 0.176 | 0.157 | 0.147 | 0.245 |
| | LLaVA-NEXT-Video | 0.252 | 0.241 | 0.227 | 0.215 | 0.193 | 0.151 | 0.179 | 0.168 | 0.221 |
| | Video-LLaVA | 0.211 | 0.229 | 0.207 | 0.207 | 0.183 | 0.150 | 0.148 | 0.183 | 0.208 |
| | ShareGPT4Video | 0.229 | 0.216 | 0.204 | 0.196 | 0.183 | 0.149 | 0.130 | 0.140 | 0.201 |
| | VideoChat2 | 0.309 | 0.257 | 0.231 | 0.235 | 0.211 | 0.186 | 0.195 | 0.128 | 0.237 |
| Precision (AutoDQ) (↑) | Tarsier | 0.548 | 0.609 | 0.626 | 0.642 | 0.659 | 0.645 | 0.667 | 0.711 | 0.628 |
| | VIdeoLLaMA2 | 0.659 | 0.662 | 0.681 | 0.682 | 0.698 | 0.727 | 0.769 | 0.741 | 0.680 |
| | LLaVA-NEXT-Video | 0.593 | 0.664 | 0.666 | 0.678 | 0.707 | 0.712 | 0.669 | 0.730 | 0.674 |
| | Video-LLaVA | 0.657 | 0.684 | 0.707 | 0.708 | 0.745 | 0.802 | 0.766 | 0.801 | 0.709 |
| | ShareGPT4Video | 0.698 | 0.720 | 0.730 | 0.735 | 0.743 | 0.758 | 0.761 | 0.779 | 0.731 |
| | VideoChat2 | 0.605 | 0.633 | 0.659 | 0.665 | 0.679 | 0.707 | 0.730 | 0.637 | 0.656 |
| F1 (FIOVA-DQ) (↑) | Tarsier | 0.318 | 0.331 | 0.312 | 0.324 | 0.324 | 0.271 | 0.231 | 0.226 | 0.320 |
| | VIdeoLLaMA2 | 0.367 | 0.328 | 0.295 | 0.305 | 0.286 | 0.238 | 0.211 | 0.174 | 0.304 |
| | LLaVA-NEXT-Video | 0.288 | 0.303 | 0.292 | 0.289 | 0.261 | 0.207 | 0.220 | 0.301 | 0.286 |
| | Video-LLaVA | 0.270 | 0.287 | 0.261 | 0.275 | 0.252 | 0.215 | 0.215 | 0.230 | 0.269 |
| | ShareGPT4Video | 0.272 | 0.275 | 0.264 | 0.266 | 0.251 | 0.215 | 0.178 | 0.203 | 0.263 |
| | VideoChat2 | 0.337 | 0.301 | 0.281 | 0.292 | 0.270 | 0.238 | 0.246 | 0.133 | 0.287 |
| Recall (FIOVA-DQ) (↑) | Tarsier | 0.485 | 0.567 | 0.575 | 0.599 | 0.613 | 0.633 | 0.623 | 0.716 | 0.584 |
| | VIdeoLLaMA2 | 0.321 | 0.277 | 0.243 | 0.249 | 0.227 | 0.170 | 0.149 | 0.114 | 0.250 |
| | LLaVA-NEXT-Video | 0.264 | 0.248 | 0.235 | 0.226 | 0.202 | 0.144 | 0.173 | 0.210 | 0.229 |
| | Video-LLaVA | 0.230 | 0.241 | 0.207 | 0.215 | 0.194 | 0.156 | 0.154 | 0.165 | 0.216 |
| | ShareGPT4Video | 0.219 | 0.218 | 0.202 | 0.205 | 0.185 | 0.156 | 0.124 | 0.145 | 0.203 |
| | VideoChat2 | 0.308 | 0.266 | 0.233 | 0.246 | 0.220 | 0.186 | 0.175 | 0.104 | 0.243 |
| Precision (FIOVA-DQ) (↑) | Tarsier | 0.485 | 0.567 | 0.575 | 0.599 | 0.613 | 0.633 | 0.623 | 0.716 | 0.584 |
| | VIdeoLLaMA2 | 0.627 | 0.628 | 0.640 | 0.647 | 0.671 | 0.702 | 0.733 | 0.674 | 0.645 |
| | LLaVA-NEXT-Video | 0.549 | 0.632 | 0.635 | 0.652 | 0.680 | 0.682 | 0.652 | 0.692 | 0.644 |
| | Video-LLaVA | 0.630 | 0.646 | 0.684 | 0.677 | 0.725 | 0.795 | 0.755 | 0.724 | 0.680 |
| | ShareGPT4Video | 0.694 | 0.706 | 0.707 | 0.713 | 0.734 | 0.750 | 0.766 | 0.803 | 0.714 |
| | VideoChat2 | 0.553 | 0.591 | 0.622 | 0.636 | 0.654 | 0.674 | 0.731 | 0.661 | 0.621 |

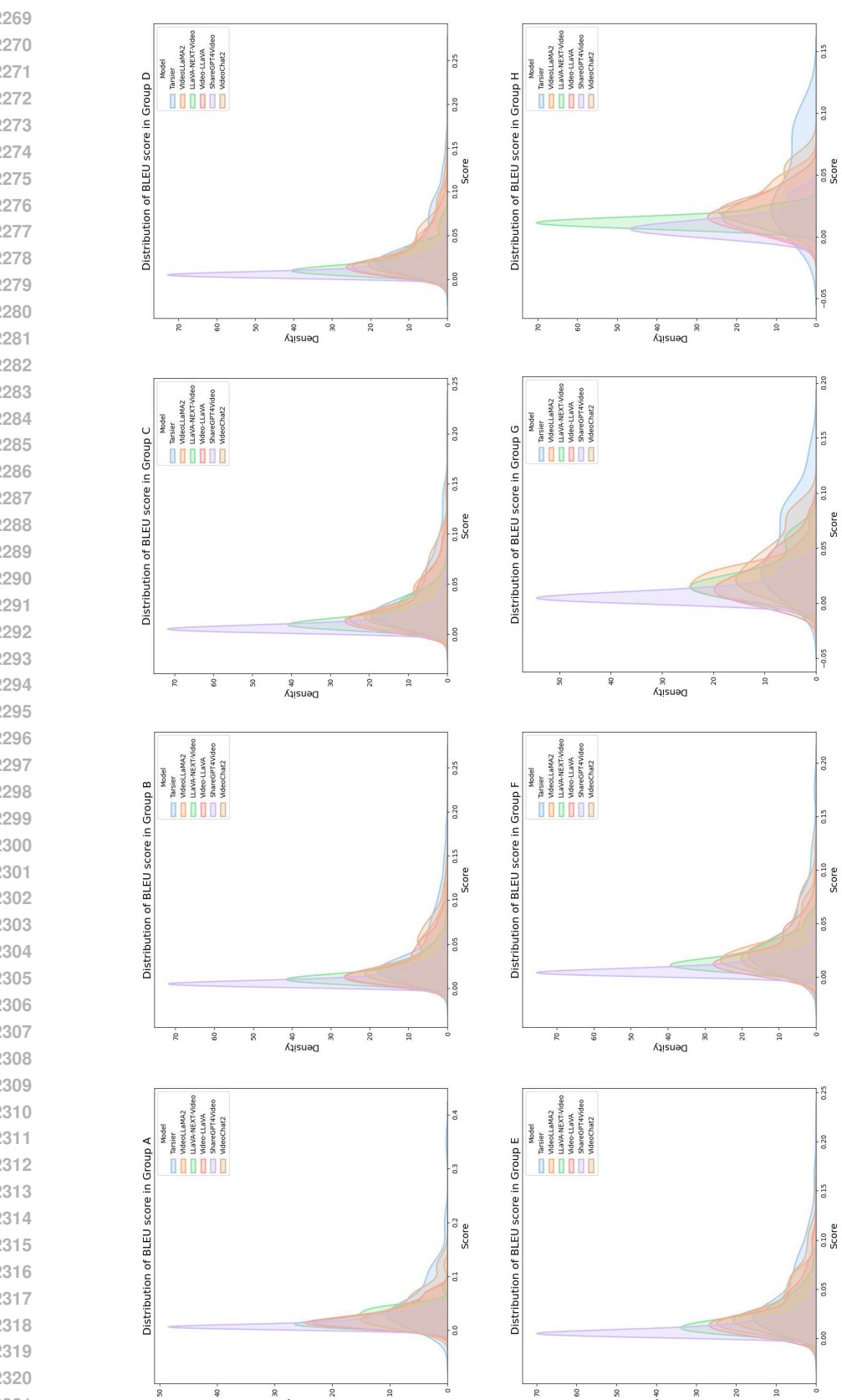

Figure A9: Distribution of LVLMs scores in different groups, based on BLEU metric.

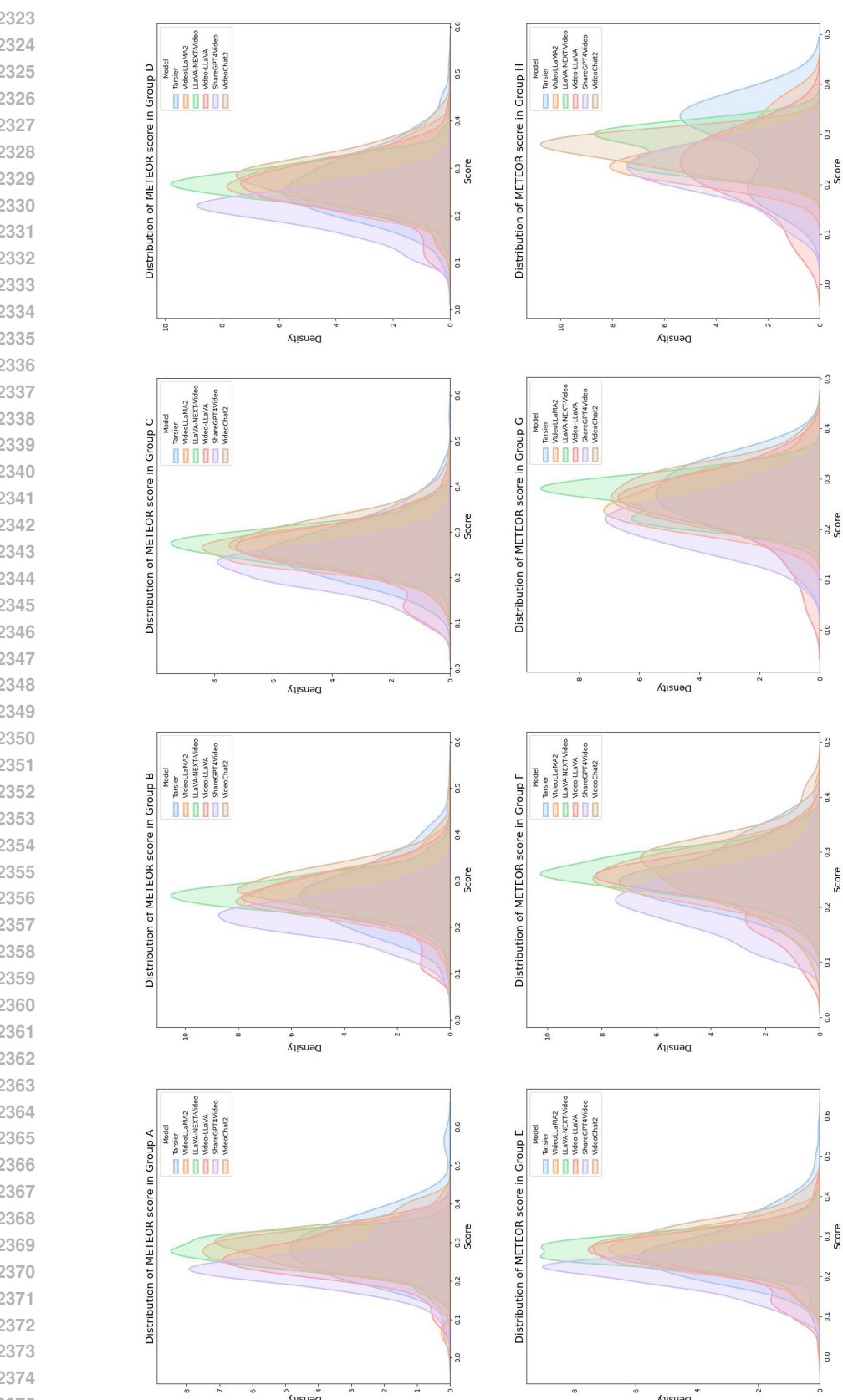

Figure A10: Distribution of LVLMs scores in different groups, based on METEOR metric.

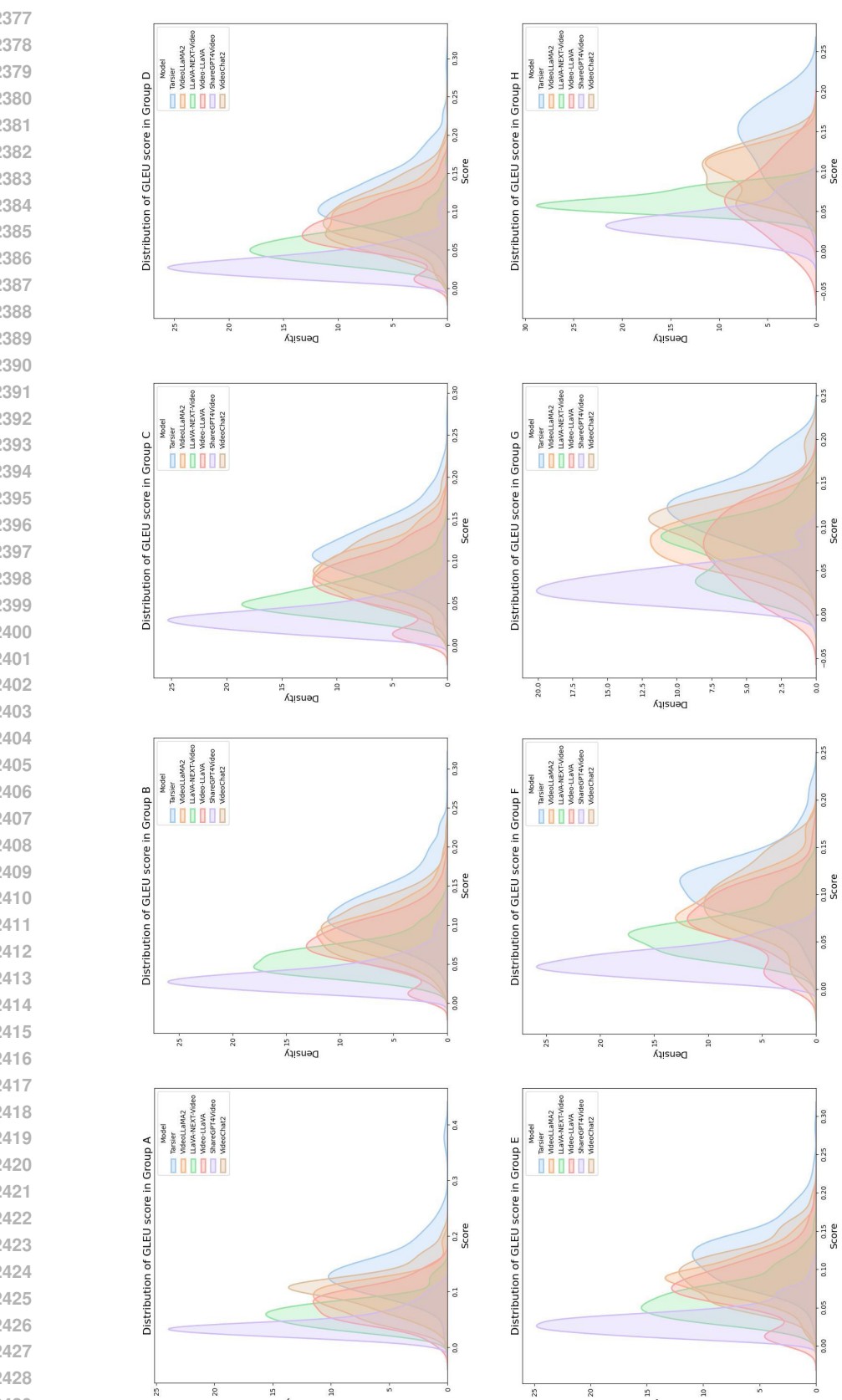

Figure A11: Distribution of LVLMs scores in different groups, based on GLEU metric.

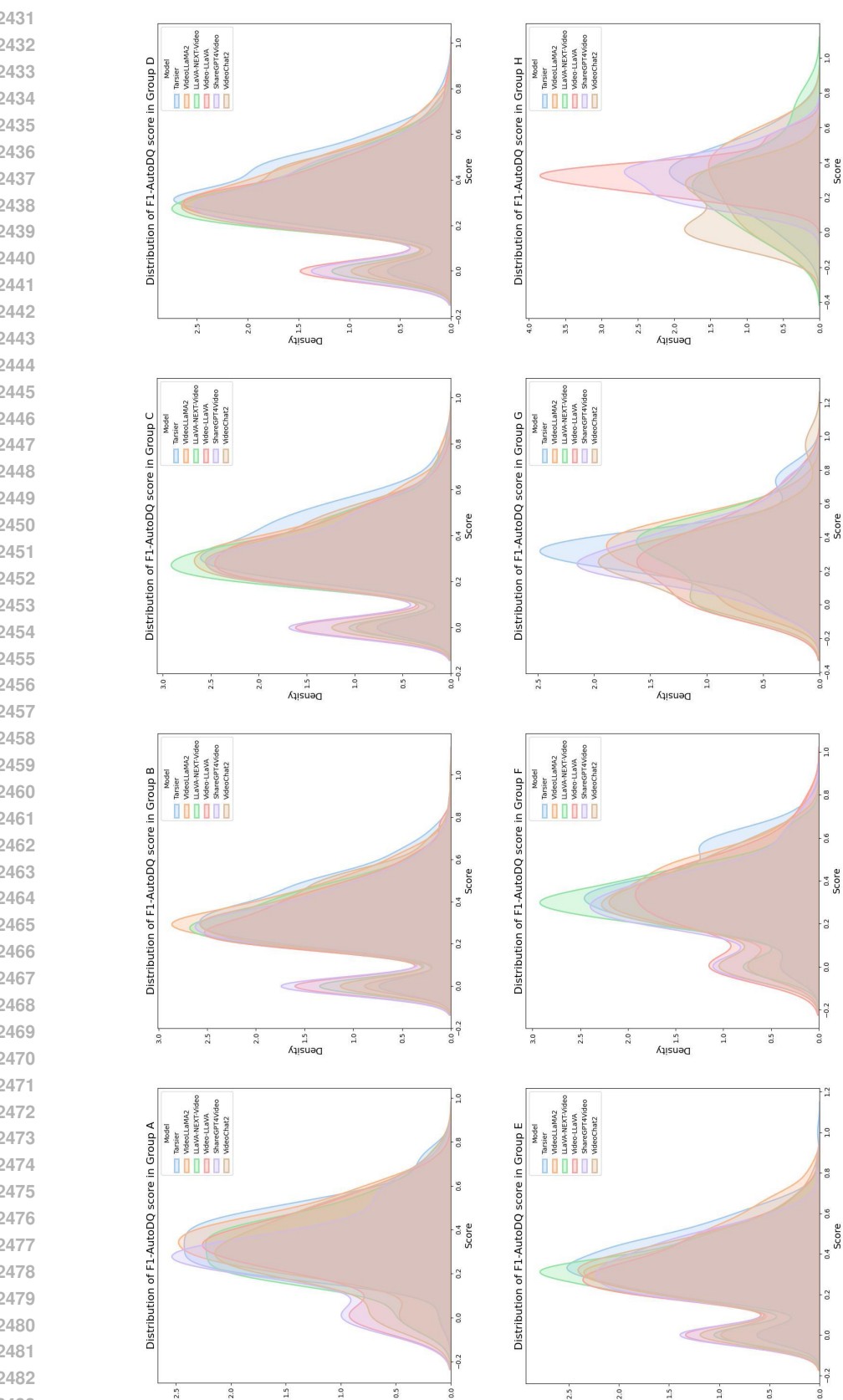

Figure A12: Distribution of LVLMs scores in different groups, based on F1 (AutoDQ) metric.

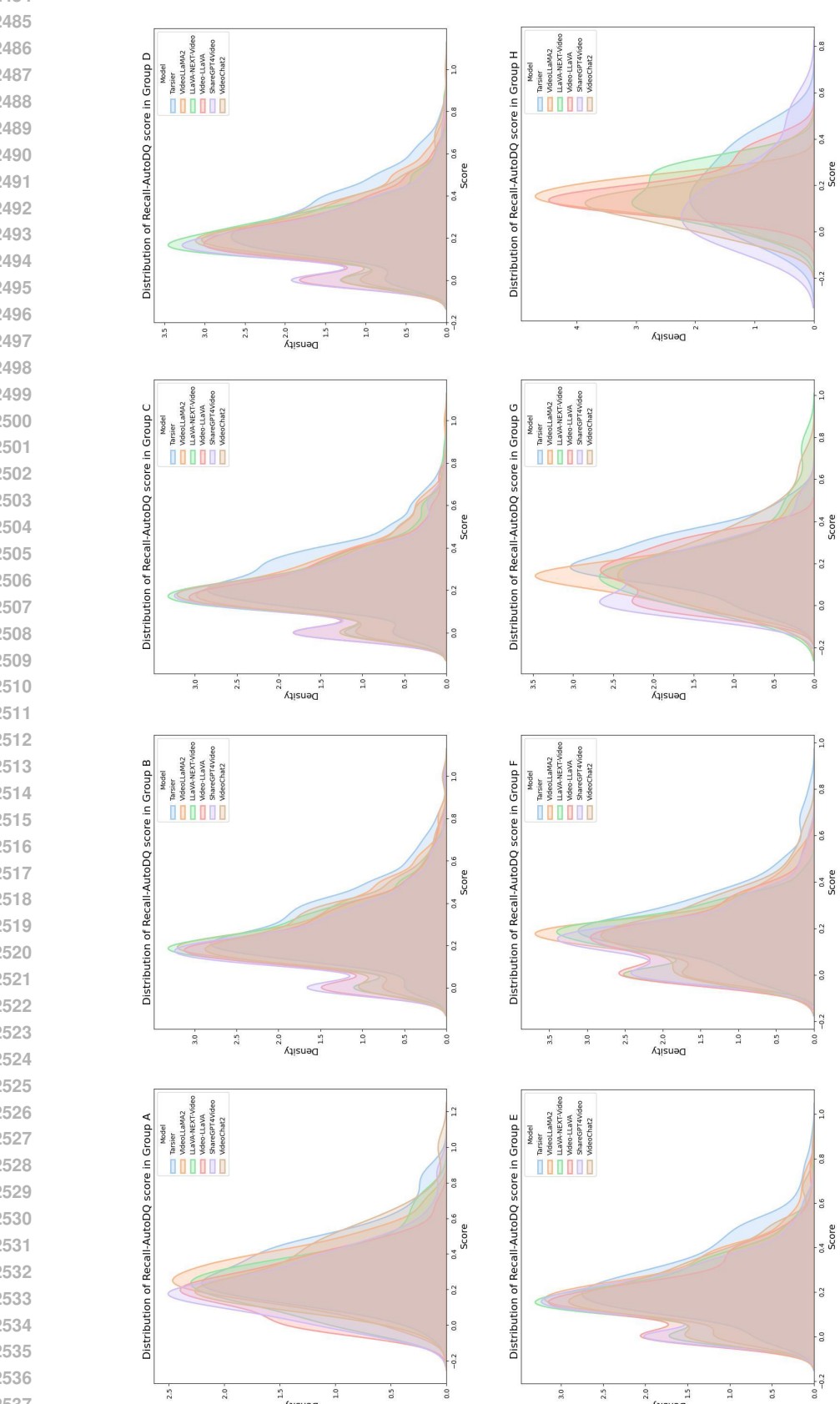

Figure A13: Distribution of LVLMs scores in different groups, based on Recall (AutoDQ) metric.

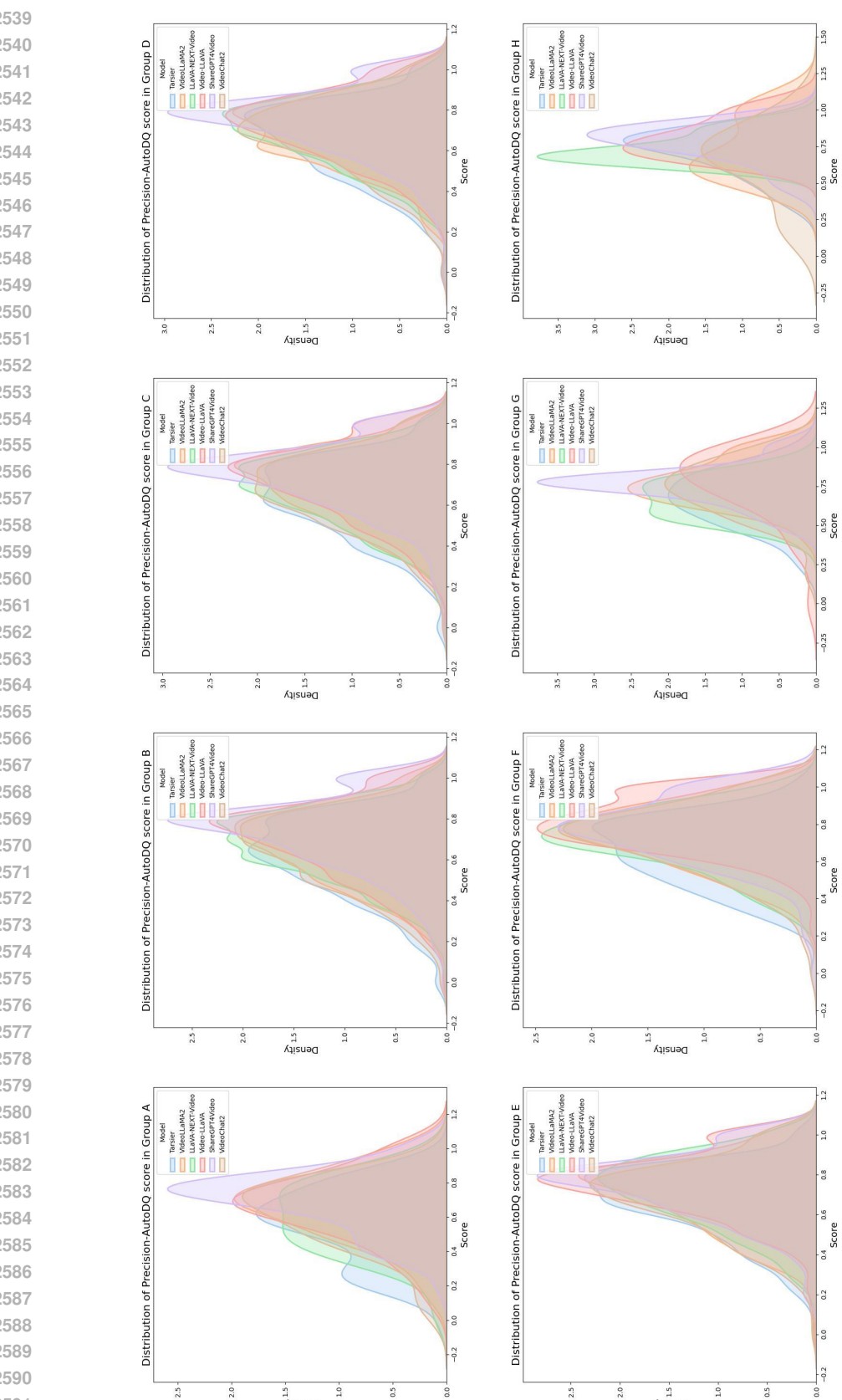

Figure A14: Distribution of LVLMs scores in different groups, based on Precision (AutoDQ) metric.

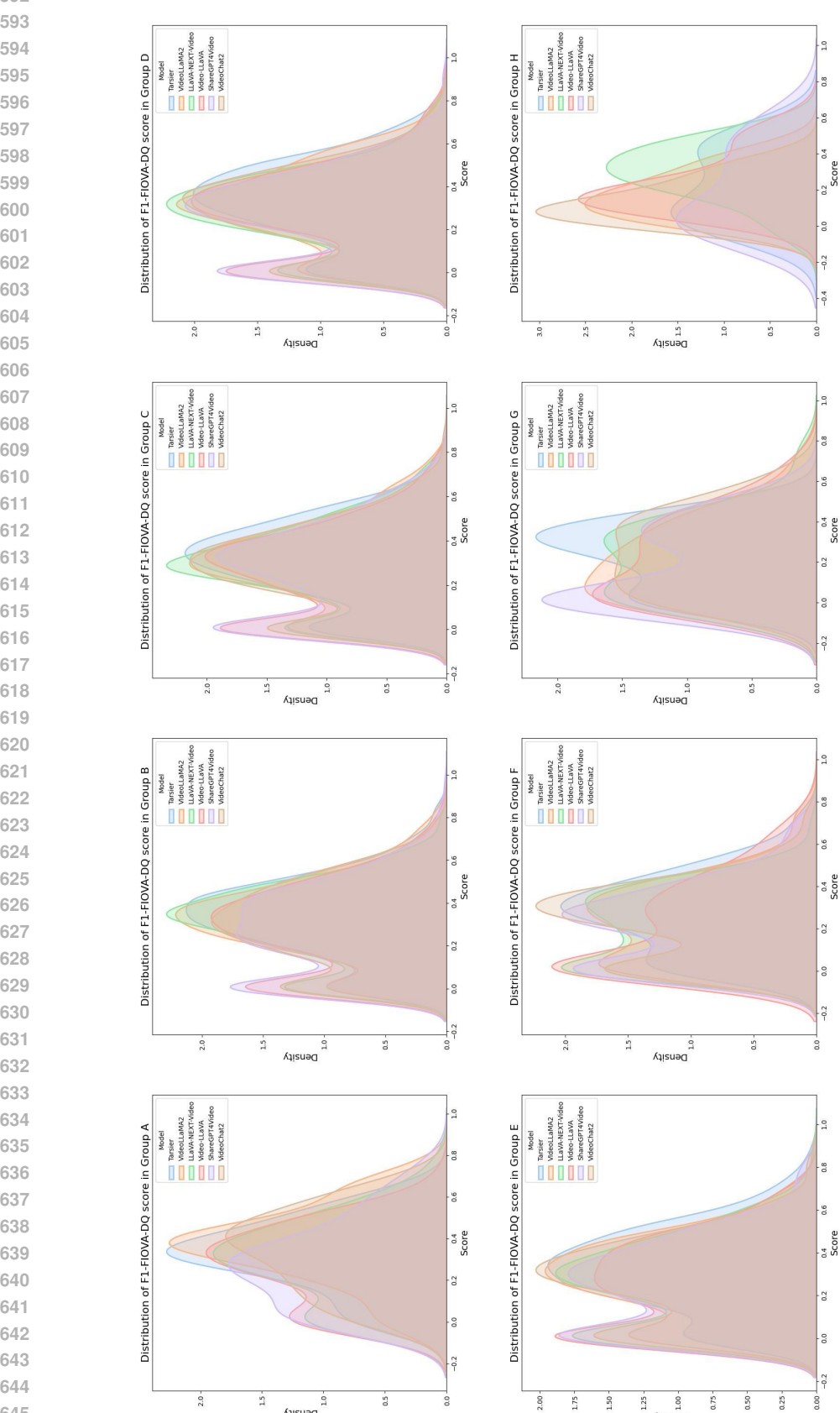

Figure A15: Distribution of LVLMs scores in different groups, based on F1 (FIOVA-DQ) metric.

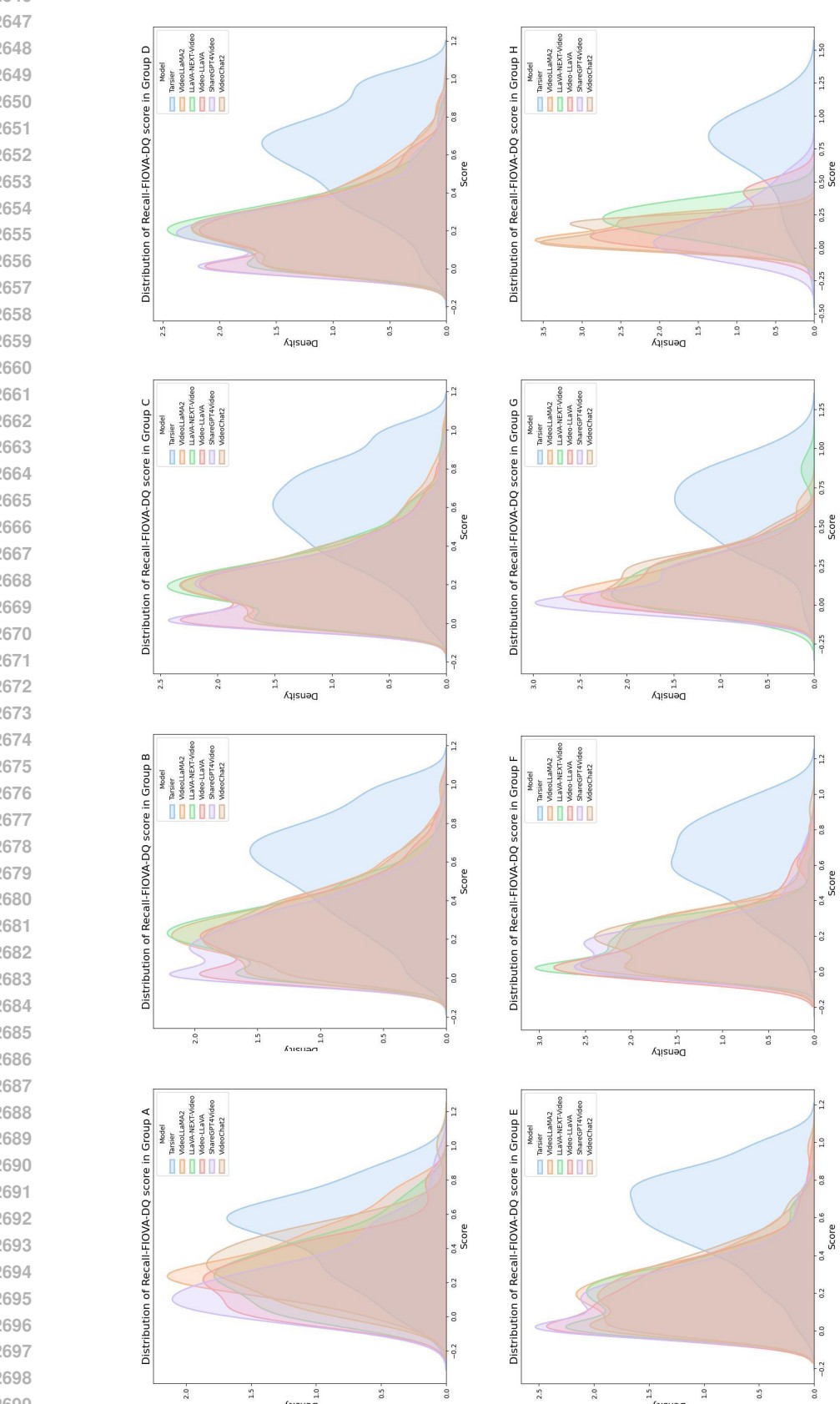

Figure A16: Distribution of LVLMs scores in different groups, based on Recall (FIOVA-DQ) metric.

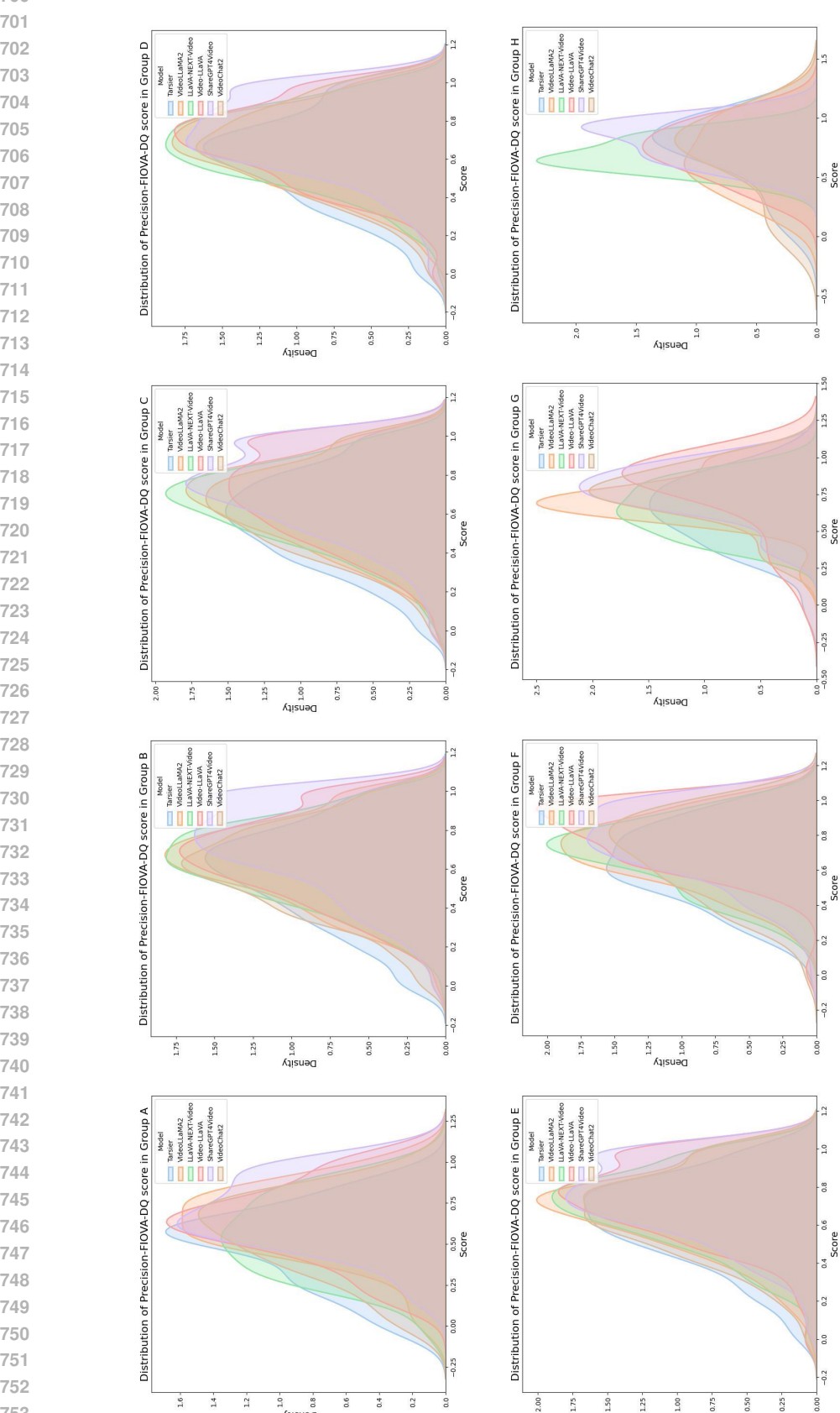

Figure A17: Distribution of LVLMs scores in different groups, based on Precision (FIOVA-DQ) metric.

## E.3 Comparison between Humans and LVLMs in Caption Length

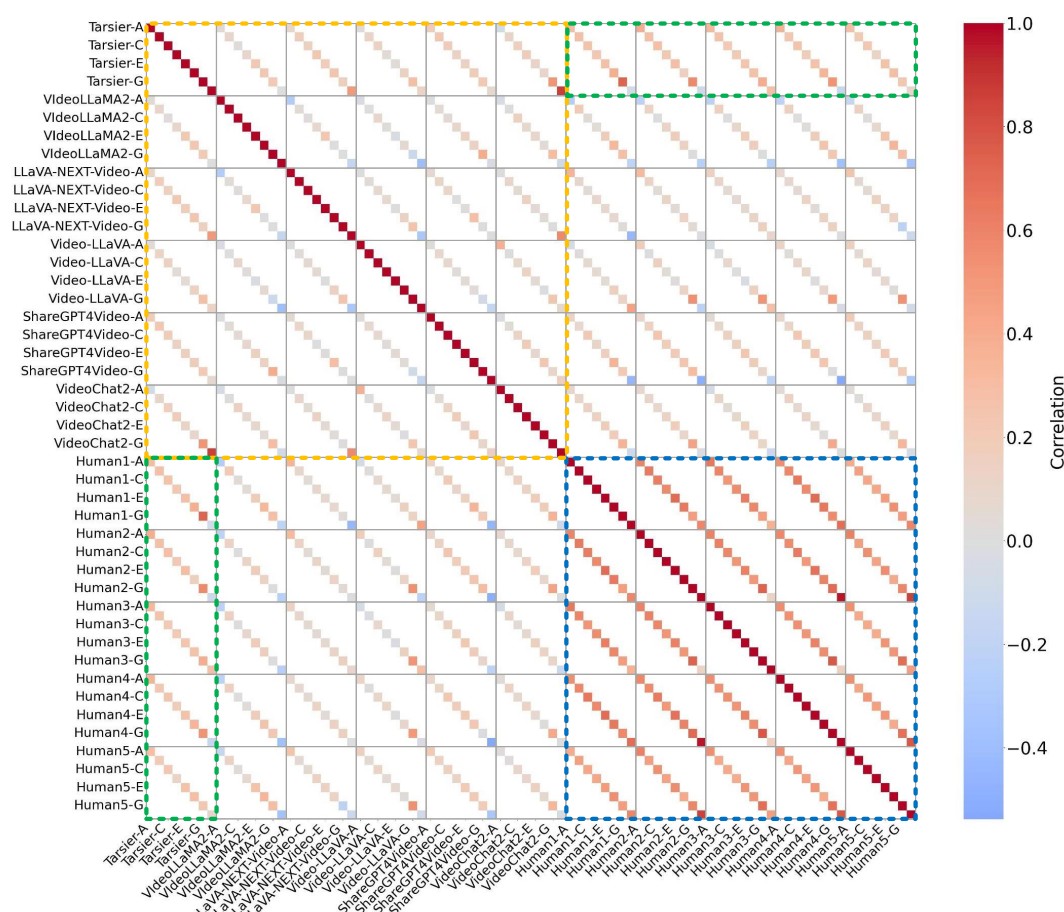

Figure A18: Correlation between LVLMs and humans in video description length (based on 8 subgroups). It can be seen that the blue dashed box represents the results of humans, and the description length is highly consistent between human annotators. The yellow dashed box shows the results of LVLMs. The description lengths between LVLMs vary greatly, especially for the descriptions of Group H, which have basically no correlation. The green dashed line is a comparison between Tarsier, the model with the best performance in multiple indicators, and humans. It can be seen that Tarsier has a higher correlation with human description length than other models.

## E.4 More Specific Examples

### E.4.1 Error Type Categorization

We annotate the examples (Fig. A19 to Fig. A24) with error categories and identified five common types of errors:

1. Omission: The model fails to describe critical events or objects in the video. While this cannot be directly marked in the model's output, we provide textual analyses of such omissions after the relevant examples.

2. Misrepresentation: The description contains information inconsistent with the video content. These errors are marked in **purple** in the model outputs.

3. Redundancy: The model repeats descriptions of the same event. These errors are marked in **yellow** in the outputs.

4. Excessive Redundancy: The model overextends or speculates excessively, introducing unnecessary content. These errors are marked in **green** in the outputs.

5. Hallucination Issues: The model includes content not present in the video. These errors are marked in **red** in the outputs.

**Human1:** A little gray boy is riding a bike. After a distance, the bike suddenly falls. The boy comes down from the bike, goes to the side, lies on the ground, pretending to fall. After a while, He reaches out his hand.

**Human2:** A child sits on a bicycle seat to take it away. He releases his hand, and the bike turns over the right. He takes out his right leg and walks a few steps and falls to the ground. Then he stretches out his right hand pointing to the lens.

**Human3:** A boy on the road is riding a small two-wheeled car, after driving a distance the child stops, the car falls to the ground, the boy comes down from the car, he lies on the road. The little boy lying on the floor strokes his hand and cries.

**Human4:** A child wearing a hat is riding on a baby carriage forward, and then the car falls, the child stands for a while and falls off when he crosses his leg out from the car. The child is lying on the ground and then pointing to the camera by a finger.

**Human5:** During the day, a little boy wearing a helmet is riding a bike without pedals,using feet to support forward. The boy release his hand, the bike tilted down under the boy. The boy stands and looks down at the bike. The boy crosses the car and goes to the side and falls to the ground. The boy smiles and reaches out his hand.

**Groundtruth:** A young boy is riding a bike down a road. As he rides, the bike suddenly falls over. The boy then gets off the bike, lies on the ground, and pretends to fall. After a moment, the boy smiles and reaches out his hand.

**Tarsier:** A child wearing a helmet and gloves is riding a small red bicycle on a paved road. The child loses balance and falls off the bicycle, landing on the ground. The scene ends with a text overlay stating 'Bit of a late reaction there mate' against a black background.

**VideoLLaMA2:** The video shows a young child riding a red bicycle on a street. The child is wearing a helmet and gloves, and appears to be enjoying himself as he rides the bike. At one point, the child falls off the bike and lies on the ground, but quickly gets back up and continues riding. **The video also shows the child playing with a red ball and a toy car.** Overall, the video is cute and entertaining, and shows a happy and playful child.

**LLaVA-Next-Video:** In the video, we see a young child wearing a helmet and riding a small bicycle. The child is in the process of learning how to ride the bike, as evidenced by their body position and the way they are holding onto the handlebars. The child appears to be in a safe and controlled environment, **possibly a driveway or a quiet street, as there are no other vehicles or pedestrians visible in the background.** The child is wearing a jacket, **suggesting it might be a cool day or they are dressed for protection.** The child's helmet is a bright color, which is a good safety measure for their age. The child's focus is on the task at hand, and they seem to be enjoying the experience of riding the bike. The background is relatively simple, with a clear view of the sky and a few trees, indicating that the setting is outdoors. The video captures a moment of childhood innocence and the joy of learning a new skill.

**Video-LLaVA:** The video shows a young child riding a small bicycle on a street. The child is wearing a helmet and **a blue glove on one hand.** The child falls off the bike and lies on the ground, crying. The camera then shows the bike on its side, and the child is seen lying on the ground. The video ends with the child crying. The video captures a heartwarming moment of a child learning to ride a bike and experiencing the joy of riding it. **The child's fall and crying add a touch of vulnerability and innocence to the scene. The video also highlights the importance of safety measures, such as wearing a helmet and gloves, while riding a bike. Overall, the video is a sweet reminder of the joys and challenges of childhood.**

**ShareGPT4Video:** The video captures a sequence of events involving a child on a red bicycle in an outdoor setting, likely a residential area. Initially, the child is seen riding the bicycle on a paved surface, wearing a helmet and a green fence and **a building** in the background. The child's posture suggests they are in motion, possibly turning or maneuvering the bike. As the video progresses, the child continues to ride, with the camera angle slightly changing to follow the movement, revealing more of the green fence and the **building's facade.**Subsequently, the child appears to have dismounted from the bicycle, standing beside it on the pavement, with the camera angle shifting to a more frontal view. The child's posture and the bicycle's position suggest a pause in activity. The environment remains consistent, with the green fence and **building** in the background.**The child then begins to move again, pushing the bicycle forward, indicating a resumption of activity. The camera follows the child's movement, maintaining a frontal perspective. The child's actions suggest they are either pushing the bike or starting to ride it again.In the following moments, the child is seen standing upright next to the bicycle, with the camera angle slightly changing to capture the child's upper body and the bicycle more prominently.** The child's posture and the bicycle's position suggest a pause in activity, possibly indicating a moment of rest or contemplation.The child then appears to be in motion again, pushing the bicycle forward, with the camera angle shifting to follow the child's movement. The child's posture and the bicycle's position suggest active engagement in pushing or riding the bike. Finally, the child is seen lying on the ground, with the bicycle lying on its side, indicating a fall. The camera angle shifts to a more overhead view, capturing the child's prone position and the bicycle's position on the ground. The environment remains consistent, with the green fence and **building** in the background, and the camera's perspective provides a clear view of the child's situation.

**VideoChat2:** The video shows a young boy wearing a helmet and riding a red bicycle down a street. He is riding the bike with both hands on the handlebars and appears to be enjoying himself. Suddenly, he loses his balance and falls off the bike, landing on the ground. The boy seems to be okay and gets up quickly, brushing himself off. The video ends with a caption that reads "on bit of reaction it takes to make it." **Overall, the video captures a typical moment of a child learning to ride a bike and the importance of being prepared for unexpected events.**

Figure A19: Human performance is relatively consistent, but there is significant variation among models, indicating that the models have poor descriptive ability in these scenarios. In some simple scenarios, humans are not only able to quickly capture key content in videos and describe it effectively, but also show a high degree of consistency. In contrast, LVLMs often struggle to grasp key details when handling such videos, leading to inadequate descriptive ability. This difficulty primarily stems from the models' limitations in understanding the overall context and interconnections within the video, particularly in integrating video events with background information. As a result, these models often fail to match human performance.

In LVLMs, LLaVA-NEXT-Video, Video-LLaVA, and VideoChat2 all exhibit varying degrees of redundancy, while ShareGPT4video shows significant hallucination and repetitive description phenomena. Tarsier does not exhibit obvious hallucination or repetitive descriptions, but there are omissions regarding the video content, such as failing to notice the actions after the little boy lies on the ground.

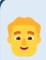

**Human1:** Three men are standing on the sidelines and watching the game. A white dress man throws the ball, and another gray dress man swings the bat. He does not hit the ball, and the bat flies out.Another gray dress man receives the bat on the sidelines. A green dress woman stands up and kisses the gray dress man. A gray dress man comes over to talk with the green dress woman. The video is repeatedly played from different angles.

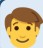

**Human2:** Several red dress men stand on the sidelines of the baseball field. A white dress athletes in the middle of the field pitches the ball,and another white dress player waves the bat. And the bat flies out. And he lifts his arms to look far away. A gray dress man stands in the auditorium.He smiles and holds a bat.The woman next to him stands up and kisses his cheek. A black dress man comes from the back row,and the woman turns back to talk with him. The lens replays the scene of bat flying out. The bat is caught by the gray dress man in the auditorium. The scene of catching the bat is replayed in slow motion. The whole process is replayed again.

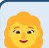

**Human3:** There are several men stand outside the ball park. There are several players in the ball park playing baseball. After a player has thrown the ball, the opposing player hits the baseball with bat and throws the bat away.Outside the ball park, a woman kisses the man smiling and standing with the bat in his hand.A man comes from behind and talks something to a woman. The lens replays the scene that the player throws out the bat and the man outside the pitch catches the bat.

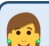

**Human4:** Three men wearing a red hat watches the ball on the sidelines. An athlete throws a ball on the ball part, the opposing players hits the ball and throws the bat to the audience. The bat is received by a man wearing short-sleeves. A woman next to him kisses the short-sleeved man. A man wearing a hat comes next to the woman and talks to her. And then the video just now is played in a slow motion.

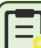

**Human5:** During the day, several men wearing red hats stand on the sidelines of the ball park. On the pitch, the pitcher throws the ball and the baseball player hits the ball with bat. The bat is threw out. The players watch the bat flying out. In the auditorium, a man holds a bat,and a woman next to him kisses his cheek. The people around applaud. The scene of the bat being threw out and the man catching the bat is replayed in a slow motion.

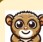

**Groundtruth:** Several men in red hats stand on the sidelines of a baseball game, watching as the pitcher throws the ball and the batter hits it, sending the bat flying. In the stands, a man catches the bat thrown from the field, while a woman kisses him on the cheek. Another man approaches the woman and engages in conversation. The video clip is replayed multiple times, showing the action from different angles and in slow motion.

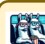

**Tarsier:** The video clip is from a baseball game. It starts with a group of players in red uniforms sitting on the bench. The scene then shifts to the pitcher preparing to throw the ball. The batter is seen swinging at the pitch, and the ball is hit into the crowd. The camera follows the ball as it goes into the stands, showing the audience reacting to the hit. The video ends with a subscription screen for MLB.TV.

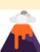

**VideoLLaMA2:** The video shows a baseball game in progress, with a player **named bryce harper** hitting a home run and **being congratulated by his teammates and fans in the stands.** The camera pans to different angles, showing the excitement and energy of the crowd. **The video also shows the player's reaction and celebration after hitting the home run. Overall, the video captures the excitement and joy of watching a baseball game and the thrill of a home run.**

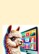

**LLaVA-Next-Video:** The video appears to be a scene from a baseball game, possibly a professional or semi-professional match, given the presence of players in uniforms and the crowd in the background. The main focus is on a player holding a baseball bat, **who is being congratulated by a woman. The player is smiling and appears to be in a celebratory mood, possibly after hitting a home run or making a significant play.** The woman is holding the bat, and they are both in the foreground, with the crowd in the background. The player is wearing a baseball uniform, and the woman is wearing a red shirt. The crowd is a mix of spectators, some of whom are holding umbrellas, suggesting it might be a sunny day or they are prepared for rain. **The atmosphere is lively, with people cheering and clapping, indicating a positive outcome in the game. The video captures a moment of celebration and camaraderie between the player and the woman, possibly a teammate or a fan, amidst the excitement of the ongoing game.**

**Video-LLaVA:** The video starts with a man in a red shirt and white pants standing in front of a crowd. He is holding a baseball bat and is ready to hit the ball. The camera then pans to a man in a green shirt and gray pants who is also holding a baseball bat. He is standing next to the man in the red shirt. The man in the red shirt then hits the ball, and the camera pans to the crowd. **The man in the green shirt is seen running to first base. The video then shows the man in the green shirt running to second base. The man in the red shirt is seen running to first base. The video then shows the man in the green shirt running to third base. The video ends with the man in the green shirt running back to second base. Overall, the video captures a baseball game where the man in the red shirt hits the ball, and the man in the green shirt runs to first base, second base, and third base.**

**ShareGPT4Video:** The video captures a series of moments from a baseball game, beginning with a scene of a player in a red uniform, likely a batter, preparing to swing at a pitch. The focus then shifts to a player in a white uniform, possibly a pitcher, in the midst of a pitch, with the batter in the red uniform in the background. **The camera then moves to a wider shot of the field, showing a player in a red uniform running towards first base, indicating a hit, with the pitcher in the background and the batter watching from the dugout.** The scene transitions to a close-up of a player in a red uniform, possibly a catcher, in a crouched position, with the pitcher in the background, suggesting a moment of anticipation or reaction to the play. *The camera then shifts to a wider shot of the field, showing a player in a red uniform running towards first base, with the pitcher in the background and the batter in the dugout, indicating a continuation of the game. The focus then moves to a player in a red uniform, likely a catcher, in a crouched position, with the pitcher in the background, suggesting a moment of anticipation or reaction to the play. The scene transitions to a close-up of a player in a red uniform, possibly a catcher, in a crouched position, with the pitcher in the background, suggesting a moment of anticipation or reaction to the play. The video continues with a close-up of a player in a red uniform, likely a catcher, in a crouched position, with the pitcher in the background, suggesting a moment of anticipation or reaction to the play. The scene then shifts to a wider shot of the field, showing a player in a red uniform running towards first base, with the pitcher in the background and the batter in the dugout, indicating a continuation of the game. The focus then moves to a player in a red uniform, likely a catcher, in a crouched position, with the pitcher in the background, suggesting a moment of anticipation or reaction to the play. The video concludes with a close-up of a player in a red uniform, likely a catcher, in a crouched position, with the pitcher in the background, suggesting a moment of anticipation or reaction to the play. Throughout the video, the camera captures the dynamic and engaging moments of the baseball game, from the anticipation of a pitch to the excitement of a hit and the strategic positioning of the players on the field.*

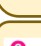

**VideoChat2:** The video shows a group of children playing in a park. They are seen running around and playing with each other. One of the children is seen doing a backflip, while another child is seen doing a cartwheel. The children are having a great time and seem to be enjoying themselves. The park is surrounded by trees and there are some benches and picnic tables nearby. The weather seems to be sunny and pleasant. Overall, the video captures the joy and energy of children playing in a park.

Figure A20: There is no significant difference in performance between the models and humans. When key content in a video is very obvious and easy to identify (such as someone playing baseball or a clear change of scenery), LVLMs can quickly capture these elements just like humans and generate corresponding descriptions. This type of video primarily relies on intuitive visual information rather than deep contextual or cultural background.

In this video, due to the camera switches and the complexity of the video content, each model has information omissions. In addition, ShareGPT4Video has a lot of repetitive and redundant descriptions. Compared to other models, VideoChat2 incorrectly identifies the entire video as children playing.

**Human1:** A woman wearing a small glasses is reading books. A woman wearing a big glasses is looking forward. A man sitting beside a lot of books and holding a book looks at the front. The woman wearing big glasses lies on the ground. A group of cranes walk by, a man and a woman dancing behind. A woman in pink walks, a man and a woman dancing behind. A black woman lies down and reads, a red dress woman sitting in a chair looks at the right. The woman with big glasses waves around the crane. A man wearing glasses is reading. The pink dress woman is walking through, the man wearing glasses is reading, the black woman is lying on a black and white shirt and reading. A man wearing a hat dances and walks through the black man upside down. A woman is lying next to a group of cranes. A woman steps on the book and walks. The woman in pink is dancing and walking through, a crane also comes.

**Human2:** The lens sweeps a lady from top to bottom, and then there appears a woman with curly hair. A man is wearing a suit, the man lying down is looking at her. Lens switch, the lady is lying on the floor, a group of white flamingos walk by, someone next to them is dancing. A man and a woman push around, the first lady appears lying down and reading, the man in suit also wears glasses reading, the curly hair women and flamingos are dancing, someone next to them stretches his leg doing exercise.

**Human3:** In a yard, a black-skinned woman is carrying a bag in the hands and reading a book, another long-haired woman is staring at the camera. A woman wearing a suit is lying on the stool, holding A book and looks at the lens, the long hair woman is lying down and reading. A group of birds walk through the hall, a red dress man pushes a blonde woman away, the black skin woman next to him sitting to the side reads, another woman with black skin is lying down and reading. A woman wearing a red hat is sitting to the side, the long hair woman shakes hands, a woman in suit wears glasses, another woman wearing a striped shirt lies next to the carpet. The man in red keeps beating, A woman lying on the table raises her legs, the long hair woman is lying on the carpet, a pink dress woman is shaking the body and walking through.

**Human4:** A woman standing next to some leaves. A woman is lying on the ground. Some geese are walking. A man and a woman are talking. A man is reading a book. A woman is sitting in a chair. A woman is waving her hands. A man is wearing glasses. Several people are lying on the ground. A man is leaning up and a man is walking by his side.

**Human5:** A woman carrying a bag is standing and reading. A woman wearing glasses looks at the camera. A person holding a book looks at the woman. The woman wearing glasses is lying on the ground. Several people are dancing, a person is lying down and reading, a person is sitting on a chair. A man is waving his hands. The reading people wears the glasses. A man jumps forward and looks at another person who stands on the stool. The women with glasses is lying on the ground. A person steps on the book. Everyone does their own thing.

**Groundtruth:** A black-skinned woman is carrying a bag in the hands and reading a book, while a woman in pink walks through the scene. A man is wearing a suit, the man lying down is looking at her. A woman wearing glasses, who at times waves her hands and lies on the ground, interacts with a group of cranes or flamingos that walk by. Another man wearing a hat dances and appears to walk through a black man who is upside down. A woman steps on a book as she walks, and a red dress woman sits in a chair looking to the right.

**Tarsier:** The video clip features a series of scenes with different individuals in various settings. It begins with a woman standing outdoors, **followed by a scene with a group of people and swans in what appears to be a garden or event space.** Next, a man is shown reading a book indoors. Another scene features a person in a hat and colorful outfit interacting with a flamingo in a lush, decorative environment. The video concludes with a person lying on the floor, surrounded by plants and other people in an indoor setting with a glass ceiling. The word 'GUCCI' appears on the screen at one point.

**VideoLLaMA2:** The video shows a fashion shoot with a model wearing a Gucci outfit. The model is posing in a greenhouse surrounded by plants and flowers. She is wearing a colorful skirt, a red blouse, and a pair of glasses. The model is also holding a purse and a book. The video also shows the model walking and dancing in the greenhouse. **The video is visually appealing and the model looks confident and stylish. The background music is upbeat and adds to the overall atmosphere of the video. The video is well-edited and the lighting is good. The video is a great example of fashion photography and it showcases the beauty and elegance of the Gucci brand.**

**LLaVA-Next-Video:** The video features a woman and a man in a room with a distinctive aesthetic. The woman is wearing a floral dress with a high neckline and a red belt, and she is holding a black purse. **She is seated on the floor,** with her legs crossed, and appears to be reading a book. **Her glasses are off, and she is looking directly at the camera.** The man is lying on the floor, also with his legs crossed, and is wearing a suit with a red tie. He is holding a book and appears to be reading as well. **The room they are in has a vintage feel, with a large window that lets in natural light, and there are potted plants and books scattered around, suggesting a cozy and intellectual atmosphere. The woman's dress and the man's attire suggest a formal or semi-formal occasion. The room is decorated with various pieces of furniture, including a chair and a couch, and there are more books and plants in the background, indicating a space that is used for reading or studying. The overall scene is calm and relaxed, with the subjects engaged in quiet activities.**

**Video-LLaVA:** The video starts with a man and a woman sitting on a bench, and the man is reading a book. The woman is wearing a red shirt and a black skirt. The man is wearing a black coat and a red tie. The scene then shifts to a man in a black coat and red tie walking through a park with a group of flamingos. **The man is holding a book in his hand.** The scene then shifts to a woman in a pink dress lying on the ground, and a man in a black coat and red tie is reading a book. The woman is wearing a pink dress and a black coat. The man is wearing a black coat and a red tie. **The scene then shifts to a man in a black coat and red tie sitting on a bench, and a woman in a pink dress is lying on the ground. The man is reading a book, and the woman is wearing a pink dress. The scene then shifts to a man in a black coat and red tie sitting on a bench, and a woman in a pink dress is lying on the ground. The man is reading a book, and the woman is wearing a pink dress. The scene then shifts to a man in a black coat and red tie sitting on a bench, and a woman in a pink dress is lying on the ground.**

**ShareGPT4Video:** The video presents a series of scenes set in a whimsical, indoor environment that resembles a greenhouse or conservatory, filled with lush greenery and a variety of objects that contribute to its unique atmosphere. Initially, the scene is set with a person lying on the ground, surrounded by a vibrant array of plants and a colorful, patterned blanket. **As the video progresses, the person begins to move, initially sitting up and then standing, indicating a transition from a state of rest to activity. The person's movements are subtle, with slight adjustments in posture and position, suggesting a gentle, exploratory interaction with the environment. Throughout the video, the camera's perspective shifts to highlight different aspects of the scene, from the person's movements to the surrounding objects and the overall ambiance of the setting. The person's actions are deliberate and measured, with movements that suggest a careful exploration or interaction with the environment. At one point, the person is seen lying on the ground again, possibly indicating a moment of rest or contemplation, before resuming movement. The camera's focus alternates between the person and the surrounding environment, capturing the person's interactions with the space and the objects within it. As the video continues, the person's movements become more pronounced, with actions that suggest a dynamic engagement with the environment. The person is seen standing, walking, and interacting with various objects, including a large, colorful blanket and a patterned rug. The camera's perspective shifts to emphasize these interactions, capturing the person's movements and the details of the objects they engage with. Towards the end of the video, the person's actions become more focused and deliberate, with movements that suggest a purposeful interaction with the objects around them. The camera's perspective shifts to highlight these interactions, capturing the person's movements and the details of the objects they engage with. The video concludes with the person standing and facing the camera, suggesting a moment of pause or reflection amidst the ongoing exploration and interaction with the whimsical, indoor environment.**

**VideoChat2:** The video starts with a woman standing in a room with a man. The woman is wearing a white shirt and the man is wearing a black shirt. **They are both standing in front of a mirror. The woman then walks towards a table and picks up a book. She starts reading the book while sitting on a chair.** The camera then shows a group of people sitting in a room with a large window. They are all dressed in formal attire and are engaged in a conversation. The video then shows a woman sitting on a couch with a book in her hand. She is wearing a white shirt and has a pair of glasses on her face. **The camera then shows a woman sitting on a chair with a book in her hand. She is wearing a white shirt and has a pair of glasses on her face. The video ends with a woman sitting on a couch with a book in her hand. She is wearing a white shirt and has a pair of glasses on her face.**

Figure A21: There is a significant variation in descriptions among humans, but the models perform more consistently.

Due to the strong artistic elements in this video, the content is quite complex, making it difficult for humans to reach a consistent descriptive conclusion. As a result, LVLMs struggle to focus on certain scene details, leading to hallucinations, repetition, and redundancy. This issue is particularly prominent in ShareGPT4Video.

Humans often vary in their descriptions of complex videos due to personal experiences, emotions, cultural backgrounds, and individual preferences, which can make their descriptions differ significantly. In contrast, LVLMs tend to be more consistent in their descriptions. These models are trained on vast datasets with the goal of learning a more universal, standardized way of describing. The training of these models typically focuses on identifying and describing visual elements that are widely recognized in most contexts, unaffected by individual traits. Thus, these models exhibit higher consistency and predictability in generating descriptions.

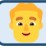 **Human1:** A woman is sitting, and several people are sitting together. The table is covered with bread. The other three are standing. The woman looks at the camera. At a party, the woman laughs.

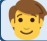 **Human2:** A woman holding a cup sits on the steps. Several people are sticking papers to the balloon. There are food on the table. A cake in one man's hand falls to the ground. At another party, the woman holds a windmill in her hand. There are food on the table. The children run around.

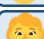 **Human3:** A woman dressed in white holding a cup sitting. She looks to somewhere else. There is a dining table next to her. She is holding a corn and eating. She gives some food to the girls then she smiles.

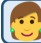 **Human4:** A black woman sits on the steps, bread is putted on the table, a black man throws the hamburger on the ground. Many people play together, there are corn and burger on the table, some little girls run to her and talk with her.

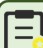 **Human5:** A woman is sitting in a seat with a glass of water. A man squeezed the tomato sauce on the cake and the cake falls to the ground. Woman is holding a windmill. A group of people are dining. There are a variety of foods on the table. A group of children run around on the lawn.

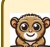 **Groundtruth:** A woman is sitting at a party, looking at the camera and laughing. Several people are sitting together at a table covered with bread while others are standing. Meanwhile, a cake falls to the ground as a man tries to stick papers to a balloon. The woman then holds a windmill and interacts with children running around. There are various foods on the table, including corn, burgers, and a tomato sauce squeezed on a cake.

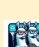 **Tarsier:** The video clip shows a series of scenes from a party event. It begins with a woman sitting on steps, holding a cup. The scene then shifts to children making decorations, followed by a close-up of ketchup being poured onto a hamburger. The next scene shows a variety of food items on a table, including corn on the cob and hamburgers. The focus then moves to a group of people standing and talking, with some holding drinks. The final scene features a woman interacting with others, with a graphic overlay promoting 'Party City' at the end.

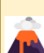 **VideoLLaMA2:** The video shows a group of people having a picnic in a park. The scene is set in a sunny day with people sitting on blankets and enjoying food and drinks. The table is filled with various items, including hamburgers, hot dogs, corn on the cob, and drinks. People are seen laughing and talking while enjoying their meal. The video also shows children playing with toys and a woman serving food to the guests. **The overall atmosphere of the video is joyful and relaxed. The video is shot in a close-up perspective, allowing viewers to see the details of the food and the people's expressions. The video is well-lit, with the sun casting a warm glow over the scene. The video is set to upbeat music, adding to the festive mood. The video is shot in high definition, providing clear and detailed visuals. The video is also edited with smooth transitions and a consistent frame rate, ensuring a seamless viewing experience. Overall, the video captures the essence of a summer picnic, highlighting the joy and camaraderie of spending time with friends and family.**

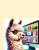 **LLaVA-Next-Video:** The video features a group of people gathered around a table, enjoying a meal together. The main subjects are a woman **and a man, both of whom are seated at the table. The woman is holding a plate of food, possibly a hot dog, and appears to be in the process of eating it. The man is also seated at the table, but it's not clear what he is doing at the moment.** The table is laden with various food items, including hot dogs and corn on the cob, suggesting a casual outdoor dining setting. In the background, there are other people engaged in conversation and enjoying their meals, contributing to the relaxed and social atmosphere of the scene. **The setting seems to be a park or an outdoor event, as indicated by the presence of other individuals and the casual attire of the people. The overall mood of the video is cheerful and convivial, with everyone enjoying their time together.**

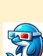 **Video-LLaVA:** The video starts with a woman sitting on a bench, and a man is seen pouring ketchup on a hot dog. The woman then stands up and walks away. The scene then shifts to a group of people sitting around a table, enjoying a meal. **The camera then pans to a woman who is talking to the camera.** The video then shows a man making a sandwich, and the camera pans to a woman who is eating a hot dog. **The video ends with the woman sitting on the bench again. Overall, the video seems to be a casual meal scene with people enjoying food and each other's company. The video captures the simple pleasures of life, such as enjoying a hot dog or a sandwich. The video's focus is on the people and their interactions with each other, rather than the food itself. The video's tone is light-hearted and relaxed, making it a perfect watch for anyone looking for a break from their busy lives.**

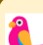 **ShareGPT4Video:** The video begins with a scene of a woman in a white top and dark pants, seated on a bench in a public space, possibly a park or plaza, with a cityscape in the background. **She appears to be in a contemplative or relaxed state, with her gaze directed downwards.** The scene then transitions to a different setting, where a group of people is gathered around a table laden with food items, including sandwiches, chips, and drinks, suggesting a social gathering or event. The focus shifts to a close-up of a person's hand holding a white plate with a pink design, indicating a moment of selection or presentation of food. The narrative progresses to show a plate with a red substance, possibly food, on a wooden surface, **hinting at a meal or snack being prepared or consumed.** The perspective then widens to reveal a person in a white top and dark pants, seated at a dining table with a plate of food, suggesting a mealtime scenario. The scene shifts to a close-up of a person's face, **showing a moment of distress or discomfort,** possibly related to the food on the plate. The narrative continues with a person in a white top and dark pants, now standing and holding a plate with a red substance, indicating a continuation of the meal or snack. **The focus then moves to a person in a white top and dark pants, seated at a dining table with a plate of food, suggesting a mealtime scenario.**

**VideoChat2:** The video shows a group of people gathered in a backyard, enjoying a picnic. The scene starts with a woman sitting on a chair, **holding a plate of food and a drink.** She is surrounded by other people who are also enjoying their meals. The camera then pans to a table filled with food, including hot dogs, hamburgers, and other picnic favorites. The people are seen laughing and having a good time. **The video then shows a group of people dancing and having fun in the backyard. The scene is lively and full of energy, with everyone enjoying the outdoor party. Overall, the video captures the essence of a fun and enjoyable picnic party.**

Figure A22: The performance of the various models was relatively poor in describing this video. In this video, the performance of the models was unsatisfactory due to the depiction of a woman immersed in her fantasies. The content of the fantasies and the environment around the woman contain many details, such as camera transitions and temporal discontinuities. These complex elements make it difficult for the models to accurately interpret and describe the video, resulting in an overall description that is not clear or easy to understand.
So all LVLMs have varying degrees of content omissions, and most exhibit hallucination and repetitive description phenomena.

**Human1:** man in white falls down to the ground and keeps speaking. His right leg is under the crotch of the man in black who is kneeling down. the right foot of the man in black is on the ground. man in white holds trousers of the man in black. the man wears black dress,the man 's right leg restores the original action. Man in white is on the right side of his body, he puts his left foot on his right foot, left hand holds the left shoulder of the man in black , then he uses his left leg to draw a circle and pulls the man in black to the left rear. His left hand seizes the left arm of the man in black,he raises his right beg to turn the man in black over. His right hand presses the left arm of the man in black to his back, conversation is over.

**Human2:** In a judo field, a man in black stands between the legs of the man in white and raises his arms,the man in white lies on the ground, the white man lies on the ground and speaks. He touches the shanks of the man in black and puts him on the ground. His legs clamps the thigh of the man in black, his left hand is on the left shoulder of the man in black, the man in black lies on him,the body of the man in white turns over, he throws the man in black down to the ground and hugs his arm.

**Human3:** a man in white lies on the ground and talks, man in black kneels down in front of him. the man in black raises his leg and crosses with one leg of the man in white. man in white pulls the pants of the man in black, and pulls his legs down to the ground. Then the man in white lifts another leg to hit the chest of the man in black, and pushes his shoulders with his hands. After the white man stretching his legs twice, he raises his legs bypass the head of the man in black, with leveraging knocks down the man in black. After then, the man in white uses the leveraging again, turns over the man in black, he takes advantage of this opportunity and gets up, locks his arms. He releases the man in black.

**Human4:** the man in white and the man in black perform to explain the action essentials. man in white lies on the ground, the man in black presses him. The man in white gives a sigh to the man in black to loosen his legs and expose legs' movements. They restore the original action, the man in white pulls down the man in black,and puts his leg across the man in black. man in white explains the action shortly, turns over and presses the man in black to the ground.

**Human5:** a man in white lies on the ground, a man in black lies on him,the man in white points,and explains where to puts hands and feet.and then demonstrates how to turn over the man in black, and man in white continues to show how presses man in black under his body, and shows how to controls the hands of his opponent. The two separate.

**Groundtruth:** In a judo field, a man in black demonstrates various techniques on a man in white. The man in white lies on the ground as the man in black manipulates his limbs and demonstrates how to control the opponent. They go through the actions of turning over, pressing down, and locking arms before separating.

**Tarsier:** The video clip shows two individuals practicing martial arts on a blue and black mat in a gym. One person is wearing a black gi with a purple belt, while the other is in a white gi with a green and yellow patch. The sequence begins with the person in the white gi on the ground, being controlled by the person in the black gi. The person in the white gi attempts to escape, but the person in the black gi maintains control. The person in the white gi then transitions to a dominant position, applying pressure on the black gi. The clip concludes with the person in the black gi on the ground, while the person in the white gi holds control.

**VideoLLaMA2:** The video shows two people practicing Brazilian Jiu-Jitsu (BJJ) in a gym. One person is on top of the other, and they are both wearing black and white uniforms. The person on top is applying a submission hold on the other person, who is trying to escape. The video also shows the two people discussing techniques and strategies for BJJ.The video is filmed in a well-lit gym with blue mats on the floor. The camera angles provide a clear view of the action, **and the sound quality is good**. The video is well-edited and easy to follow. **Overall, the video is an excellent resource for anyone interested in learning BJJ or improving their skills. It demonstrates the importance of proper technique and strategy in the sport, and it provides valuable insights into the mindset and approach of experienced practitioners.However, it is important to note that BJJ involves physical contact and carries some risks, so it is essential to practice safely and under the supervision of a qualified instructor. Additionally, the video does not provide any information about the specific techniques or strategies being demonstrated, so it may not be suitable for beginners or those without prior experience in BJJ.**

**LLaVA-Next-Video:** The video depicts a Brazilian Jiu-Jitsu (BJJ) training session in progress. The main subjects are two individuals, one wearing a black gi and the other in a white gi, both engaged in a grappling match. The black-gi individual is positioned on top, demonstrating a dominant stance, with their hands on the shoulders of the white-gi individual, who is on the bottom. This position is known as a "mount" in BJJJ, where the top person has complete control over the bottom person. The black-gi individual appears to be executing a technique, possibly a sweep or a submission, as they are adjusting their grip on the shoulders and preparing to move the white-gi individual to the ground.The training session takes place in a gym with a blue mat floor, **which is a common surface for BJJJ practice due to its non-slip properties and its ability to absorb impact.** The gym is well-lit, with natural light coming in from the windows, suggesting it's daytime. The walls are painted in a neutral color, and there are no visible decorations or distinctive features that provide additional context about the location. **The focus of the video is on the grappling match, with the background being a simple, uncluttered setting that allows the viewer to concentrate on the techniques being demonstrated.**The white-gi individual is wearing a white gi, which is the traditional uniform for Brazilian Jiu-Jitsu, and **they are wearing a red belt**, indicating they are likely a student or a beginner. The black-gi individual is wearing a black gi, which is also traditional for BJJJ, **and they are wearing a black belt**, indicating they are likely an instructor or a more advanced practitioner. The video captures the intensity and precision of the grappling techniques being taught, with the black-gi individual's hands on the shoulders of the white-gi individual indicating a high level of control and the white-gi individual's posture suggesting they are trying to resist or escape the move. The overall atmosphere is one of focused training and discipline, with the participants committed to improving their skills in Brazilian Jiu-Jitsu.

**Video-LLaVA:** The video features a man wearing a white shirt and black pants who is kneeling on the floor. He is being taught how to perform a move by a man wearing a black shirt and black pants who is kneeling on the floor next to him. The man in the black shirt is demonstrating the move by placing his hand on the other man's knee and pushing him forward. The man in the white shirt is then shown performing the move by pushing the man in the black shirt forward. The man in the white shirt then demonstrates the move again by pushing the man in the black shirt forward. **The video seems to be focused on teaching the man in the white shirt how to perform a specific move in martial arts. Overall, the video is a demonstration of a martial arts move being taught to a student by a master.**

**ShareGPT4Video:** The video showcases a martial arts training session between two individuals, one in a white gi and the other in a black gi, within a blue mat-covered training area. Initially, the person in the white gi is seen in a defensive posture, with the person in the black gi attempting to engage them. As the session progresses, the individual in the white gi transitions from a defensive stance to a more offensive position, attempting to execute a technique on the person in the black gi, who is now on their back. The person in the white gi maintains control, applying pressure and attempting to secure a dominant position over the person in the black gi, who is in a vulnerable position throughout the session.The person in the white gi continues to apply pressure, maintaining control over the person in the black gi, who is in a submissive position. The individual in the white gi adjusts their grip and positioning, indicating a continuous effort to secure a dominant position. The person in the black gi attempts to defend themselves, but the individual in the white gi maintains control, applying pressure and attempting to secure a dominant position. As the session progresses, the person in the white gi continues to apply pressure, maintaining control over the person in the black gi, who is in a submissive position. **The individual in the white gi adjusts their grip and positioning, indicating a continuous effort to secure a dominant position. The person in the black gi attempts to defend themselves, but the individual in the white gi maintains control, applying pressure and attempting to secure a dominant position.Towards the end of the session, the person in the white gi continues to apply pressure, maintaining control over the person in the black gi, who is in a submissive position. The individual in the white gi adjusts their grip and positioning, indicating a continuous effort to secure a dominant position. The person in the black gi attempts to defend themselves, but the individual in the white gi maintains control, applying pressure and attempting to secure a dominant position.** Throughout the session, the camera remains stationary, focusing on the interaction between the two individuals without any movement or change in perspective. The blue mat-covered training area serves as a consistent backdrop throughout the session.

**VideoChat2:** The video shows two men practicing martial arts on a blue mat. One of the men is wearing a black uniform, while the other is wearing a white uniform. They are seen practicing various moves and techniques, including grappling and striking. The video captures the men's movements in slow motion, allowing viewers to see the intricacies of their techniques. **The men appear to be highly skilled and are able to execute their moves with precision and control. Overall, the video provides a detailed look at the practice of martial arts and the dedication and discipline required to master the art.**

Figure A23: The models all demonstrated strong descriptive abilities for this video. Typically, models excel in describing videos with simple scenes, such as this one showcasing Brazilian Jiu-Jitsu practice, featuring stable camera work and clear temporal relationships. When dealing with clear and structured video content, the models are better able to accurately recognize and describe the activities and actions within the scene.

The content of this video is relatively simple, so the models perform quite well. The main issues are repetitive descriptions and redundancy, with hallucinations being relatively rare. Notably, the other LVLMs identified the martial arts clothing worn by the characters, while Video-LLaVA only recognized the color but did not distinguish the category.

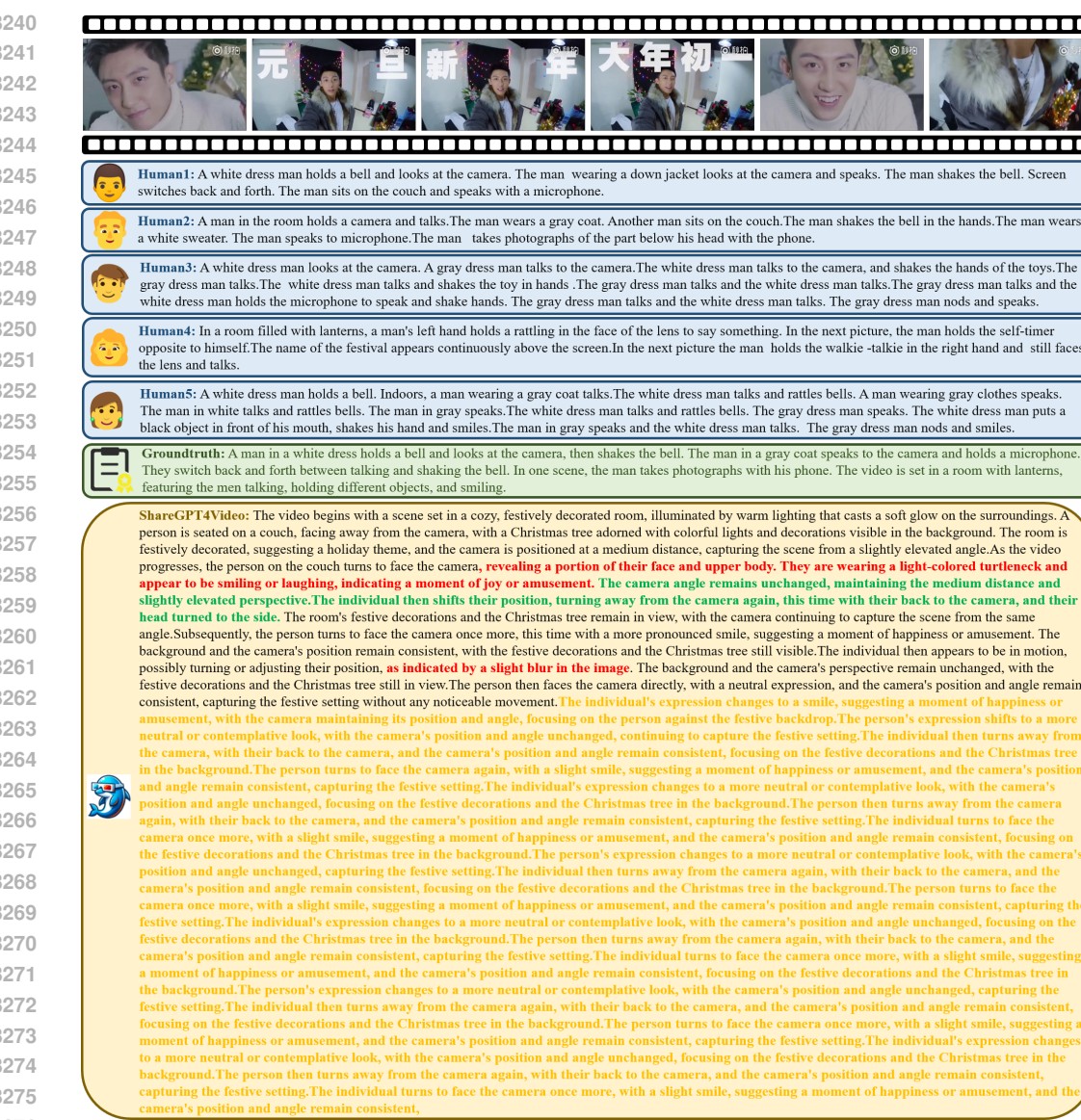

Figure A24: In the six LVLMs, ShareGPT4Video exhibits more severe hallucination issues, as shown in the image with its generation of a large amount of redundant and irrelevant descriptions.

### E.4.2 POTENTIAL CAUSE ANALYSIS

- **Architectural Limitations.**
  - **Cross-modal alignment issues:** Current LVLMs face significant challenges in effectively aligning video-text data. For instance, Tarsier processes each frame using separate visual encoders, while VideoLLaMA2 adopts a shared visual encoder for all frames. These varying alignment strategies directly impact the models' ability to interpret and understand video content comprehensively.
  - **Insufficient long-sequence modeling:** Handling long videos with multiple events requires robust attention mechanisms to ensure coherence and completeness. However, many LVLMs struggle in this aspect. For example, Video-LLaVA's descriptions often prioritize initial scenes while neglecting subsequent parts of the video.
- **Training Data Bias.**
  - **Inconsistent or insufficient data diversity:** Training data with limited diversity can lead to biased outputs. For example, Video-LLaVA shows significant difficulty in recognizing martial arts scenes (Fig. A23) compared to other LVLMs, suggesting gaps in its training dataset.
  - **Hallucination issues:** Noisy or incomplete training data may propagate hallucinated content. In Fig. A20, VideoChat2 misidentifies players and spectators in a baseball stadium as children, illustrating a severe misalignment between the output and actual video content.
- **Generation Strategy Issues.**
  - **Simplistic generation strategies:** Using basic generation techniques, such as beam search, often results in repetitive or incoherent descriptions. For instance, ShareGPT4Video, while utilizing high-quality training data, demonstrates repetitive descriptions due to inadequate constraints during generation.
  - **Weak constraints during generation:** Insufficient semantic constraints in generation processes can lead to hallucinated content or semantic errors.

### E.4.3 Suggestions for Improvement and Optimization

- **Model Optimization.**
  - **Enhancing detail capture:** Refining attention mechanisms to focus on key events and details can significantly improve the comprehensiveness of video descriptions. Hierarchical attention mechanisms for long-sequence modeling, as demonstrated by VideoLLaMA2's STC Connector, offer a promising direction for enhancing spatiotemporal continuity in descriptions.
  - **Improving semantic alignment:** Incorporating cross-modal alignment constraints, such as visual-language consistency checks, can reduce semantic discrepancies and hallucination issues. Models like LLaVA-NeXT-Video emphasize the importance of maintaining alignment consistency throughout the comprehension process.
  - **Implementing deduplication strategies:** Introducing mechanisms to detect and eliminate repetitive content during generation can improve description coherence and reduce redundancy.

- **Training Data Optimization.**
  - **Enhancing data diversity:** Expanding training datasets to include diverse scenarios, particularly complex events in long videos, can mitigate bias and improve generalization.
  - **Data cleaning:** Removing hallucinated or erroneous examples from training corpora enhances data quality. For instance, ShareGPT4Video demonstrates notable improvements through high-quality video-text data, though further refinements remain necessary.

- **Evaluation Method Enhancement.**
  - **Fine-grained error categorization:** Incorporating detailed error categorization mechanisms within the FIOVA framework can help identify model weaknesses more precisely. For example, when calculating FIOVA-DQ, event similarity between annotators' descriptions and LVLM outputs could aid in detecting specific error types.

