# OpenReview forum: "Can LVLMs Describe Videos like Humans? A Five-in-One Video Annotations Benchmark for Better Human-Machine Comparison"
_ICLR.cc/2025/Conference — Submitted to ICLR 2025_

### Official Review · Reviewer_pxyN · 2024-11-01

**Soundness:** 3
**Presentation:** 3
**Contribution:** 4
**Rating:** 6
**Confidence:** 4

**Summary:**

The authors of the paper investigate whether large vision-language models (LVLMs) can describe videos as comprehensively as humans. To facilitate the investigation, the authors propose a benchmark, FIOVA, designed to evaluate the differences between captions from LVLMs and humans, with 3,002 videos (averaging 33.6 seconds) that cover diverse scenarios.

Using the FIOVA benchmark, the authors conducted an in-depth evaluation of six state-of-the-art LVLMs (VideoLLaMA2, LLaVANEXT-Video, Video-LLaVA, VideoChat2, Tarsier, and ShareGPT4Video), comparing their performance with humans.

**Strengths:**

1. FIOVA is unique and important for advancing video-language models further. Each video in FIOVA is annotated by five distinct annotators, establishing a robust baseline that, for the first time, comprehensively represents human understanding in video description tasks.

2. In-depth evaluations of six open-source SOTA models were performed, and details of the algorithms and implementations were provided.

3. Many useful insights are provided through fine-grained evaluation and analysis. For example:
- Current LVLMs still struggle with information omission and descriptive depth.
- Different LVLMs employ varying strategies for video captioning—some prioritize completeness, while others focus on accuracy.
- For videos that are relatively easy to describe, the models show significant variability in performance. In contrast, for more challenging videos, their performance becomes more consistent.
- Significant discrepancies exist between LVLMs and humans in complex videos, particularly where human annotators exhibit substantial disagreement, whereas LVLMs tend to rely on uniform strategies for challenging content.

**Weaknesses:**

LLM hallucination and detail omission issues may be present in the ground truth description of each video, as the ground truth for each video is generated by an LLM, GPT-3.5-turbo, which synthesizes the five human-provided descriptions into a single, comprehensive video description.

Additionally, using an LLM instead of a VLM to summarize the five human captions is insufficient because an LLM cannot properly handle conflicting information in the five human captions. For example, in Figure 4, Human3 notes that the little boy cries at the end, while Human5 states that the boy smiles at the end. Since an LLM cannot 'see' the video, it may simply guess that the boy smiles at the end.

**Questions:**

1. Were the same five distinct annotators used to annotate all 3,002 videos? I assume the answer is 'No.'

2. What are the sources of the videos?

3. How do you collect and process the videos in order to obtain and control the resulting varied video complexity?

4. How does the GPT-summarized caption affect the distributions shown in Figure 3?

5. Line 306 states that each model was fine-tuned for video caption generation. Do you mean you further fine-tuned these models? On which dataset was the fine-tuning performed?

6. When the SOTA models were evaluated, 8 frames per video were used. Were 8 frames sufficient for FIOVA? Did the human annotators watch the actual video files, or were they presented with sampled frames?

7. AutoCQ only provides event-level evaluation. Why were the five dimensions (e.g., consistency, context, etc.) from Sec. 2.2 not considered in the model response evaluation metrics?

8. It appears that the comparison of an LVLM against humans is conducted by comparing the LVLM’s response with the GPT-synthesized human caption summary in the main paper. However, in the image-language domain, having multiple human annotators for each image is not new (e.g., COCO captions include 5 human captions per image), and researchers often compare a model’s caption with each human caption individually, then aggregate the metric values (e.g., averaging the 5 metric values obtained by comparing with each human caption). Is there a reason for not taking this approach?

9. In Table A.3 in the Appendix, comparing the model's response with the GPT-synthesized human caption summary tends to produce higher metric values than directly comparing the model's response with a single human caption. Do the authors have any comments or insights regarding this observation?

Minor comments:
In the qualitative results shown in Figures A12–A17, it would be helpful to highlight any hallucinated content in the model responses.

---

> ### Author Response · Authors · 2024-11-25
> **Response to Reviewer pxyN's concern about the generated groundtruth**
>
> ***Weaknesses. LLM hallucination and detail omission issues may be present in the ground truth description of each video. Additionally, using an LLM instead of a VLM to summarize the five human captions is insufficient.***
>
> **Response:** Thanks for raising concerns about hallucinations, detail omissions, and using LLMs over VLMs in the groundtruth generation process. **Below, we address these concerns in detail and clarify the reliability of FIOVA's GT methodology, supported by revisions in Appendix B.6 (see Fig. A7)**:
>
> **1. Accuracy and Diversity of the Five Annotators**
>
> - **Accuracy Assurance**: The initial annotations in FIOVA were provided by five independently trained annotators. Their descriptions, based on full video observations, ensured alignment with core events and causal logic in the videos.
> - **Diversity and Detail Coverage**: Variations in annotators’ focus on aspects such as actions, background, or narrative are natural and enhance the richness of multi-perspective annotations. These variations are integral to FIOVA’s design, creating a benchmark that better reflects human video understanding.
>
> To address concerns about the limited frame coverage in the original Fig. 4, we revised the example (**Appendix B.6, Fig. A7**) by expanding it to 35 frames sampled at 20-frame intervals. This expanded visualization confirms:
>
> - All annotators captured the core events of the video (e.g., riding a bike, pretending to fall, lying on the ground, reaching out a hand).
> - The temporal order and causal logic in the descriptions strictly align with the video content, demonstrating the accuracy of the annotations.
>
> These results affirm the reliability of multi-perspective annotations in FIOVA, distinguishing it as a benchmark with robust human-understanding baselines.
>
> **2. Rationale for Choosing LLMs over VLMs**
>
> We opted to use GPT-3.5-turbo instead of vision-language models for text integration, to minimize the risk of hallucinations and preserve annotation authenticity.
>
> - **Reducing Multimodal Hallucination Problems**: Multimodal models may exhibit alignment instability, potentially generating hallucinated content when interpreting videos. In contrast, LLM strictly processes text, focusing on semantic integration without introducing new content or subjective interpretations. This approach preserves the authenticity and diversity of human annotations while avoiding interference from multimodal alignment issues.
> - **Preserving the Diversity of Human Annotations**: LLM consolidates annotations without altering critical information or suppressing annotator-specific features. For example, in resolving conflicting descriptions (“crying” vs. “smiling”), GPT-3.5-turbo integrates contextual cues, ensuring the final groundtruth aligns with the most plausible narrative logic.
> - **Efficiency in Text Integration**: LLM effectively merges multi-annotator descriptions into coherent, semantically consistent groundtruth, validated by its application in related studies (e.g., Tarsier [1], MM-Ego [2]).
>
> **3. Analysis and Handling of Conflicts in Figure 4 (e.g., “Smiling” vs. “Crying”)**
>
> In Fig. 4, discrepancies in the boy’s expression (“crying” by Human3 vs. “smiling” by Human5) were resolved through GPT-3.5-turbo’s integration process:
>
> - **Contextual Analysis**: By analyzing key actions (e.g., “getting off the bike,” “lying down,” “pretending to fall”), combined with Human1’s mention of “pretend” and Human5’s description of “smiling,” GPT-3.5-turbo identified “smiling” as more consistent with the video’s overall narrative.
> - **Conflict Resolution**: Human3’s inference of “crying” was speculative, while Human5’s interpretation aligned better with the prank context and the video sequence, resulting in a semantically coherent GT.
>
> This integration process illustrates how LLMs effectively handle annotation conflicts, producing groundtruths that maintain consistency and completeness.
>
> **4. Measures to Ensure GT Reliability**
> - **Avoiding Hallucinations**: Strictly limiting LLMs to text integration tasks and using carefully designed prompts minimized risks of hallucinations.
> - **Enhancing Detail Retention**: Five independent annotators ensured comprehensive coverage of video content, minimizing omissions from any single perspective. Besides, manual review of all 3,002 videos identified and corrected minor redundancies, ensuring the groundtruth accurately reflects human annotations.
>
> By combining the diversity of human annotations with LLM’s text consolidation capabilities, FIOVA delivers high-quality, semantically rich groundtruths. This approach ensures credibility and distinguishes FIOVA as a benchmark for long-video description tasks.
>
> [1] Wang J, Yuan L, Zhang Y, et al. Tarsier: Recipes for training and evaluating large video description models[J]. arXiv preprint arXiv:2407.00634, 2024.
>
> [2] Ye H, Zhang H, Daxberger E, et al. MM-Ego: Towards Building Egocentric Multimodal LLMs[J]. arXiv preprint arXiv:2410.07177, 2024.

---

> ### Author Response · Authors · 2024-11-25
> **Response to Reviewer pxyN's concern about the annotators**
>
> ***Q1. Were the same five distinct annotators used to annotate all 3,002 videos? I assume the answer is 'No.'***
>
> **Response:** We appreciate the reviewer’s interest in the composition of annotators. As noted in the review, **the entire dataset was not annotated by a fixed group of five annotators**. Below, we provide a detailed explanation regarding this issue, and have added this information in **Appendix B.2**:
>
> **1. Annotator Assignment**
>
> Not all 3,002 videos were annotated by the same five annotators. To ensure diversity and operational feasibility, we adopted the following strategies:
> - **Multiple Groups of Annotators**: The FIOVA dataset was annotated by multiple groups of annotators. Each video was annotated by five independent annotators, but these annotators were not fixed across all videos.
> - **Annotator Selection Criteria**: All participating annotators underwent detailed training to ensure they understood the annotation guidelines and could deliver high-quality annotations.
>
> **2. Impact of Annotator Diversity on GT**
>
> We chose to use multiple groups of annotators rather than a fixed group of five to enhance the diversity and representativeness of the groundtruth (GT):
> - **Diverse Descriptive Styles**: The involvement of different annotators helps incorporate a wider range of perspectives and descriptive styles, reducing biases introduced by a fixed group of annotators.
> - **Broader Semantic Coverage**: The participation of multiple groups of annotators effectively improves the GT's coverage of video content details and enhances the diversity of descriptions.
> - **Improved Robustness of GT**: The diverse sources of annotations make the GT more adaptable and generalizable, allowing it to better reflect model performance across a variety of scenarios.
>
> **3. Quality Control Measures**
>
> While the diversity of annotators may introduce variations in descriptive styles, we implemented a strict quality control process to ensure the reliability and consistency of the GT:
> - **Standardized Annotation Guidelines**: All annotators followed the same annotation handbook and guidelines to ensure the consistency and comparability of annotations.
> - **Annotation Review**: Each video’s annotations underwent quality checks to ensure the descriptions aligned with the video content and contained no obvious errors.
> - **Semantic Integration and Optimization**: Using GPT-3.5-turbo, the five annotators' descriptions were consolidated to enhance linguistic consistency while retaining multi-perspective characteristics.

---

> ### Author Response · Authors · 2024-11-25
> **Response to Reviewer pxyN's concern about the data source**
>
> ***Q2 & Q3. What are the sources of the videos? How do you collect and process the videos in order to obtain and control the resulting varied video complexity?***
>
> **Response:** Thank you for raising concerns about the source of our videos. **The videos in FIOVA are sourced from publicly available copyright-compliant platforms, consistent with the practices of most benchmark datasets. Furthermore, we have adopted standardized procedures to ensure the diversity of our dataset**. Below, we provide detailed clarifications, and have added this information in **Appendix B.1**:
>
> **1. Legitimacy of Video Sources**
>
> The videos in the FIOVA dataset are all obtained from legal, publicly accessible copyright resources:
>
> - **Public Copyright Resources**: All videos are selected from online platforms with clearly defined public copyright policies, such as YouTube and similar sources. We ensure that all videos comply with relevant copyright policies.
> - **Compliance Statement**: We strictly adhere to the terms of use of the video source platforms and select only videos that can be used for non-commercial research purposes.
>
> **2. Diversity in Video Selection**
>
> To ensure the diversity of the dataset and its applicability to a wide range of use cases, we employed the following strategies during video selection:
>
> - **Coverage of Various Scene Types**: The video content spans a wide range of themes, including daily activities, sports events, and natural landscapes, ensuring the evaluation capability of LVLMs across multiple scenarios.
> - **Diverse Dynamic Complexity**: We specifically focused on videos with dynamic characteristics, such as short- and long-term temporal relationships, complex actions, and multi-agent interactions, to reflect the challenging scenarios that LVLMs may encounter in real-world applications.
>
> **3. Video Screening and Quality Control**
>
> To further ensure data quality, we implemented multiple rounds of rigorous quality control during the video screening process:
>
> - **Initial Screening**: We selected videos that complied with public copyright requirements and demonstrated strong diversity.
> - **Manual Review**: Each video was manually reviewed to ensure its clarity and suitability for video understanding tasks.
>
> During video processing, we utilized pre-screening and grouping techniques to ensure diversity in video length, content, and event complexity. These methods were designed to comprehensively evaluate LVLM performance across videos of varying difficulty levels, achieving a robust assessment outcome.

---

> ### Author Response · Authors · 2024-11-25
> **Response to Reviewer pxyN's concern about the distributions relationship of Fig.3 and Fig.4**
>
> ***Q4. How does the GPT-summarized caption affect the distributions shown in Figure 3?***
>
> **Response:** Thanks for raising this insightful question. We want to clarify that during the construction of the FIOVA benchmark, GPT-3.5-turbo was used twice in our study, and these two steps are entirely independent. **There is no direct connection between the first step (illustrated in Fig. 3) and the second step (illustrated in Fig. 4)**. In other words, the distributions shown in Fig. 3 are solely based on GPT-3.5-turbo’s analysis of human annotations and are unrelated to the Groundtruth generation process.
> Below is a detailed explanation:
>
> **1. First Use of LLM: Analyzing Human Annotations for Grouping**
>
> - **Context**: This corresponds to Sec. 2.2: Caption Quality Assessment and Fig. 3.
> - **Objective**: To quantify the consistency of human annotations based on semantic differences and calculate the Coefficient of Variation (CV) to group videos into categories A-H.
> - **Method**: GPT-3.5-turbo analyzed the textual descriptions provided by five annotators for each video across five dimensions—contextual consistency, semantic coverage, linguistic fluency, as well as the length of descriptions. The CV for each dimension was calculated, and videos were grouped based on the resulting CV scores.
>
> **2. Second Use of LLM: Generating Groundtruth Descriptions**
>
> - **Context**: This corresponds to Sec. 2.3: Groundtruth Generation and Fig. 4.
> - **Objective**: To generate a unified and comprehensive Groundtruth description by semantically integrating the multi-perspective descriptions provided by five annotators.
> - **Method**: GPT-3.5-turbo combined the multi-perspective annotations from the five annotators, performing linguistic optimization and semantic integration to create a refined Groundtruth description.
>
> To summarize, these two steps are entirely independent. The distributions in Fig. 3 reflect the semantic differences in human annotations analyzed by GPT-3.5-turbo. The purpose of this analysis was to evaluate the variability among annotators' descriptions across multiple dimensions (Consistency, Context, Correctness, etc.) and calculate CV scores for each video. These CV values measure the level of consistency in the human annotations and are used to group the videos.
>
> On the other hand, Fig. 4 demonstrates how GPT-3.5-turbo generates Groundtruth descriptions through semantic integration. During this process, GPT-3.5-turbo does not introduce new modal information; it simply integrates and optimizes the language based on the textual descriptions provided by the five human annotators. This step primarily aims to create a unified Groundtruth reference to evaluate model performance on long video captioning tasks.

---

> ### Author Response · Authors · 2024-11-25
> **Response to Reviewer pxyN's concern about the fine-tuning**
>
> ***Q5. Line 306 states that each model was fine-tuned for video caption generation. Do you mean you further fine-tuned these models? On which dataset was the fine-tuning performed?***
>
> **Response:** Thanks for pointing this out. During the inference process, **we did not fine-tune the LVLMs, and to ensure fairness in evaluation, we preserved the original configurations of the models as much as possible**. We have revised this section in the updated manuscript to clarify this point and prevent any misunderstanding. We sincerely apologize for any confusion caused by the previous description.

---

> ### Author Response · Authors · 2024-11-25
> **Response to Reviewer pxyN's concern about the 8 frames**
>
> ***Q6. When the SOTA models were evaluated, 8 frames per video were used. Were 8 frames sufficient for FIOVA?***
>
> **Response:** We appreciate the reviewer’s attention to the frame selection strategy. In our study, **we chose to use 8 frames per video based on a careful consideration of balancing evaluation efficiency and information capture**, which aligns with the standard experimental paradigm in the current field of video tasks. Below are the detailed explanations:
>
> **1. Consistency with Experimental Paradigm**
>
> The core design of FIOVA is to provide an open and high-quality evaluation benchmark for analyzing and comparing the long-video description capabilities of various LVLMs and human annotators. To build a reproducible and scalable evaluation platform, we adhered to standard paradigms widely adopted in the field, such as fixed-frame sampling. This consistency not only facilitates horizontal comparisons with existing works but also provides a reference framework for future studies.
>
> **2. Methodological Generality**
>
> Long-video tasks often involve processing a substantial amount of visual information. Selecting 8 frames as input effectively reduces computational cost while supporting large-scale experiments and benchmark construction. In fact, many related works (e.g., VideoGPT+ [1] and Emu-3 [2]) also adopt 8-frame inputs, highlighting the representativeness and generality of this setting in long-video understanding tasks. Moreover, current LVLMs often have frame count limitations; too many frames may overload the input capacity or degrade performance. The 8-frame setup accommodates the capabilities of mainstream models while avoiding information redundancy.
>
> **3. Fairness and Feasibility of the Evaluation Platform**
>
> In our study, all experimental results were derived from the 8-frame setting. This choice not only validated FIOVA’s evaluation capability under this configuration but also ensured fairness and feasibility within the evaluation platform. The selection of 8 frames strikes an optimal balance among semantic capture, experimental efficiency, and model constraints, making it a reasonable setting under the current standard experimental paradigm.
>
> In the revised manuscript, we further clarify the experimental rationale behind the 8-frame setup and emphasize FIOVA’s flexibility and extensibility in **Appendix D.2**. By providing an open frame-setting option, future research can explore more sophisticated frame sampling strategies to further unlock the potential of models.
>
> [1] Maaz M, Rasheed H, Khan S, et al. VideoGPT+: Integrating Image and Video Encoders for Enhanced Video Understanding[J]. arXiv preprint arXiv:2406.09418, 2024.
>
> [2] Wang X, Zhang X, Luo Z, et al. Emu3: Next-token prediction is all you need[J]. arXiv preprint arXiv:2409.18869, 2024.

---

> ### Author Response · Authors · 2024-11-25
> **Response to Reviewer pxyN's concern about the human annotators**
>
> ***Q7. Did the human annotators watch the actual video files, or were they presented with sampled frames?***
>
> **Response:** We appreciate the reviewer’s attention to the annotation process. All videos in the FIOVA dataset were annotated by human annotators after **watching the complete videos**, ensuring both comprehensiveness and consistency in the annotations. Specifically:
>
> **1. Comprehensiveness**
>
> Annotators, by watching the full videos, could accurately capture all key events, dynamic relationships, and temporal sequences. This comprehensive annotation approach ensures that the core semantics of the video content are fully represented.
>
> **2. Consistency in Key Details**
>
> Viewing the entire video enabled annotators to understand complex spatiotemporal causalities and multi-agent interactions, resulting in semantically consistent and detail-rich descriptions. This approach effectively avoids semantic biases or missing details that might arise from fragmentary viewing.
>
> **3. Unified Training and Strict Guidelines**
>
> All annotators underwent standardized training to familiarize themselves with the annotation process and task objectives. They followed strict annotation guidelines, ensuring the quality and consistency of the descriptions and establishing the reliability of the dataset.
>
> Through these measures, the annotation process for the FIOVA dataset not only guarantees multidimensional coverage of video content but also enhances the semantic reliability and applicability of the dataset.

---

> ### Author Response · Authors · 2024-11-25
> **Response to Reviewer pxyN's concern about using 5 evaluation dimensions in future experiments**
>
> ***Q8. AutoCQ only provides event-level evaluation. Why were the five dimensions (e.g., consistency, context, etc.) from Sec. 2.2 not considered in the model response evaluation metrics?***
>
> **Response:** We sincerely thank the reviewer for raising this important question. The five dimensions introduced in Sec. 2.2 (Consistency, Context, Correctness, Detail Orientation, Temporality) were specifically designed using GPT to assess **the relative differences among human annotators' descriptions, rather than to serve as absolute metrics for model evaluation**. Therefore, we did not incorporate them as direct metrics in the subsequent evaluation of model performance. Below are detailed explanations:
>
> **1. Core Objectives and Application Scenarios of the Five Dimensions**
>
> The core goal of these five dimensions is to measure the semantic differences among human annotators, rather than to judge the absolute quality of the descriptions. For example, a video may have descriptions with differing focuses (e.g., one focusing on background details and another on actions). Such variations reflect the natural diversity of human cognition and should not be viewed as strictly better or worse. To this end, we calculate the coefficient of variation (CV) for each dimension to quantify the relative consistency of descriptions. Importantly, CV emphasizes the overall distribution of descriptions rather than the absolute accuracy of individual scores. This calculation provides the foundation for grouping videos into categories (Groups A-H). Thus, these five dimensions are more suitable for **relative evaluation** rather than **absolute performance assessment**.
>
> **2. Why Were the Five Dimensions Not Directly Used for Model Evaluation?**
>
> Although these five dimensions effectively capture the variability among human annotations, there are limitations to applying them directly to model evaluation:
> - **Relative vs. Absolute Evaluation**: The five dimensions are designed to assess the relative differences in human annotations, whereas model evaluation focuses on the **absolute semantic quality** of generated descriptions. For example, in measuring Consistency, a higher CV indicates significant variability in human annotators' styles; however, for models, we require explicit metrics to determine the quality of the generated descriptions.
> - **Suitability**: Model evaluation typically relies on established absolute metrics (e.g., BLEU, METEOR) and event-level metrics (e.g., AutoDQ and FIOVA-DQ). These metrics are better suited for directly measuring semantic consistency, event coverage, and temporal coherence in model-generated descriptions.
>
> **3. Embedding Relative Evaluation Principles into the Framework**
>
> While the five dimensions were not directly used for model evaluation, we incorporated the **relative evaluation** principle into subsequent experiments. For instance, in the Batch Ranking experiment (**Sec. 4.3 and Appendix C**), we compared the consistency of descriptions between human annotators and model-generated outputs, thereby indirectly quantifying their relative differences.
>
> In summary, the primary objective of the five dimensions is to capture the relative variability among human annotators, rather than to serve as absolute metrics for model performance. This design emphasizes the distribution and diversity of descriptions but is less suited for absolute semantic evaluation compared to traditional and event-level metrics (e.g., AutoDQ and FIOVA-DQ). By embedding the principles of relative evaluation into later experiments (e.g., Batch Ranking), we expanded their applicability. We sincerely thank the reviewer for their insightful feedback on this matter.

---

> ### Author Response · Authors · 2024-11-25
> **Response to Reviewer pxyN's concern about metrics**
>
> ***Q9. It appears that the comparison of an LVLM against humans is conducted by comparing the LVLM’s response with the GPT-synthesized human caption summary in the main paper. However, in the image-language domain, having multiple human annotators for each image is not new (e.g., COCO captions include 5 human captions per image), and researchers often compare a model’s caption with each human caption individually, then aggregate the metric values (e.g., averaging the 5 metric values obtained by comparing with each human caption). Is there a reason for not taking this approach?***
>
> **Response:** We sincerely thank the reviewer for raising this very important question. Based on the reviewer's suggestion and considering the characteristics of FIOVA (each video is annotated by five human annotators), we have optimized the AutoDQ metric and **proposed the FIOVA-DQ metric (see Common Concern 2: Propose a New Evaluation Method FIOVA-DQ). Its computational approach aligns with the key points raised by the reviewer**. Below, we elaborate on the reasons behind our prior choice of using a synthesized GT for evaluation and provide an introduction to FIOVA-DQ to better address the reviewer's suggestion.
>
> **1. Reasons for Merging Multiple Annotators’ Descriptions**
>
> - **Integration of Multi-Perspective Information**: Each annotator may focus on different aspects of the video content (e.g., action details, background information, or event logic). By merging the descriptions from five annotators, the GT captures a broader semantic representation of the video, reducing the risk of missing details inherent in single annotator descriptions.
> - **High-Quality Reference Standard**: Compared to using the description from a single annotator, the merged GT more accurately reflects the overall human understanding of video content, providing a higher-quality baseline for evaluating LVLMs.
>
> **2. Experimental Support for Individual Comparisons**
>
> As suggested by the reviewer, we conducted experiments in Appendices E.1 and E.2 that adopt the approach of individual comparisons. In these experiments, the LVLM-generated outputs are compared with each annotator’s description individually, and the results are then aggregated. This method offers additional insights and demonstrates the differences between individual comparison and merged description evaluations.
>
> **3. Optimization and Improvements with FIOVA-DQ**
>
> In the revised version of our paper, we have taken the reviewer’s suggestion into account and further optimized the AutoDQ method, proposing a new evaluation metric, FIOVA-DQ. The FIOVA-DQ metric is conceptually aligned with the idea of individual comparisons, as it incorporates the following features:
> - **Event Weighting Mechanism**: During evaluation, FIOVA-DQ assigns weights to events based on their importance, as determined by the descriptions from all five annotators, thereby integrating the perspectives of multiple annotators into the evaluation process.
> - **Closer Alignment with Human Judgment**: Compared to traditional metrics and the original AutoDQ, FIOVA-DQ better captures event importance and semantic consistency, producing evaluation results that are more aligned with human intuitions regarding description quality (**see revised paper Sec 3.2 and Sec 4.1**).
>
> In conclusion, we believe that merging the descriptions from multiple annotators to construct the GT is an effective approach. At the same time, we have explored the method of individual comparisons in our experiments and incorporated the strengths of both approaches into FIOVA-DQ. This ensures a more comprehensive and robust evaluation framework.

---

> ### Author Response · Authors · 2024-11-25
> **Response to Reviewer pxyN's concern about scores**
>
> ***Q10. In Table A.3 in the Appendix, comparing the model's response with the GPT-synthesized human caption summary tends to produce higher metric values than directly comparing the model's response with a single human caption. Do the authors have any comments or insights regarding this observation?***
>
> **Response:** We sincerely thank the reviewer for their valuable observation. Below is our detailed analysis and explanation, we have added it in **Appendix E.1**:
>
> **1. GPT-Summarized Descriptions Improve Information Coverage**
>
> - **Integration of Multi-Perspective Information:** Each video in the dataset was annotated by five independent annotators who watched the entire video before providing detailed descriptions. Due to differing observational focuses, each annotator's description may highlight different aspects, such as:
>
>   - One annotator may focus on the details of characters' actions;
>   - Another annotator may emphasize the environment or background.
>
> GPT-3.5-turbo aggregates these descriptions, integrating the multi-perspective information from the five annotators into a more comprehensive and diverse GT description.
>
> - **Broader Content Coverage:** Compared to single annotator descriptions, GPT-summarized descriptions provide a more extensive representation of video content. For instance, in the case shown in Figure 4 (**further analyzed in Appendix B.6**), some annotators may focus on the actions of the child, while others may note background or secondary details. Through aggregation, these elements are comprehensively represented in the GT.
>
> **2. Reasons for Higher Metric Scores**
>
> - **Higher Consistency Between Model Output and Aggregated GT:** When compared with the aggregated GT, model outputs are more likely to align with certain annotators' observations, leading to higher metric scores (e.g., BLEU, METEOR). This is because:
>   - Model outputs may randomly cover content described by multiple annotators;
>   - The aggregated GT includes richer content details, reducing the likelihood of missing information.
> - **Limitations of Single Annotator Descriptions:** Single annotator descriptions may only cover certain aspects of the video content. When compared with model outputs, these descriptions may suggest that the model has omitted certain details, leading to relatively lower scores.
>
> **3. Rationale for Using Aggregated GT**
>
> Using GPT-summarized descriptions as GT is both reasonable and necessary, for the following reasons:
> - **Improved Comprehensiveness**: Aggregated GT maximally retains the diverse perspectives of the five annotators, providing a more complete representation of the video content;
> - **Reflection of Multi-Perspective Features**: This aligns with FIOVA's core philosophy of evaluating models' semantic understanding through multi-perspective information;
> - **Enhanced Fairness in Evaluation**: Multi-perspective GT avoids the biases that may result from individual annotators' preferences, enabling a more comprehensive evaluation of model performance.

---

> ### Author Response · Authors · 2024-11-25
> **Response to Reviewer pxyN's concern about cases**
>
> ***Q11. In the qualitative results shown in Figures A12–A17, it would be helpful to highlight any hallucinated content in the model responses.***
>
> **Response:** We sincerely thank the reviewer for the valuable suggestion. In the revised **Appendix E.4** of the paper, we have added more analyses, **using different colors to annotate the types of errors made by the models**. Specifically, we categorized the following five common types of errors:
>
> - **Omission**: The model fails to describe key events or objects in the video. While this cannot be directly marked in the model's output, we have provided textual analyses of such omissions after the relevant examples.
> - **Misrepresentation**: The description contains information inconsistent with the video content. These errors are marked in purple in the model outputs.
> - **Redundancy**: The model repeats descriptions of the same event. These errors are marked in yellow in the outputs.
> - **Excessive Redundancy**: The model overextends or speculates excessively, introducing unnecessary content. These errors are marked in green in the outputs.
> - **Hallucination Issues**: The model includes content that is not present in the video. These errors are marked in red in the outputs.
>
> By incorporating detailed error analyses and exploring potential causes, we aim to further enhance the value of FIOVA in evaluating LVLM performance and provide additional insights for model optimization. We appreciate the reviewer’s suggestion, which helps us present our research more comprehensively and explore broader research directions.

---

> > ### Comment · Reviewer_pxyN · 2024-11-29
> > **Additional review comments (part 1)**
> >
> > I appreciate the authors' response and the significant efforts made to improve the paper. However, some concerns remain.
> >
> > 1. I understand that using five annotators would lead to more accurate and diverse video captions, which is a reasonable approach. However, the final ground truth is obtained by using an LLM to integrate the five human captions. LLMs are not perfect; they are prone to hallucinations and lack access to actual visual cues. The LLM-based integration process may introduce new hallucinations, which is concerning given that FIOVA serves as an evaluation benchmark. I am particularly worried about the quality of the final, integrated ground truth. In my opinion, this final integration step should be performed by humans rather than an LLM.
> >
> > To address this concern, could you provide quantitative human evaluation results? Specifically, if humans were the judges, how would they rate the quality of the LLM-integrated ground truth?
> >
> > 2. Section 2.2 requires better clarity.
> > Lines 204-206 state:
> > "... a consolidated human description was generated as the final groundtruth, serving as a refined summary for video captioning evaluation."
> > This led me to believe that when GPT-3.5 was used to evaluate human captions across the five key dimensions, each human caption was compared to the GPT-3.5-integrated ground truth for each dimension. However, the prompts in Appendix D.1.1 and Algorithm A1 suggest that each human caption is actually not compared against any reference caption.
> >
> > The method described in Section 2.2 appears to be adapted from the Video-ChatGPT evaluation metrics (https://github.com/mbzuai-oryx/Video-ChatGPT/tree/main/quantitative_evaluation) with slight modifications to the prompts and the removal of the reference caption. The lack of a reference caption is concerning. For instance, the definition of "Detail Orientation" is "Whether the description captures critical details," and the definition of "Temporality" is "Whether the description follows the chronological order of events without skipping or over-summarizing." Without a reference caption, how can GPT-3.5 evaluate whether the provided human caption captures critical details or follows the correct chronological order?
> >
> > If I have misunderstood this, please clarify and improve the paper’s explanation.
> >
> > 3. My question about whether 8 frames are sufficient for FIOVA primarily concerns the temporal complexity of the videos in FIOVA. Specifically, how many frames are necessary to effectively "solve" FIOVA? What is the minimum number of frames required to capture all critical details and events in FIOVA?
> >
> > FIOVA is a video dataset with an average duration of 33.6 seconds. Nevertheless, researchers have found that for some video benchmarks, high performance can be achieved with relatively few frames because those benchmarks are not temporally challenging. I am curious whether 8 frames are sufficient for FIOVA to yield high results, or whether FIOVA is more temporally complex, requiring more than 8 frames to solve effectively.
> >
> > Among the models being evaluated, some are capable of handling more frames, which typically improves performance. For example, Tarsier was trained to handle over 30 frames, and in my experience, providing more than 8 input frames (around 24, though the optimal number depends on the dataset) usually yields much better performance. Is this the case for FIOVA as well?
> >
> > 4. The authors did not address my question about why the five dimensions from Section 2.2 were not considered as the model response evaluation metrics. As mentioned earlier, these dimensions were adapted from Video-ChatGPT evaluation metrics, which have gained popularity. To compare the model response with the ground truth, this work should include the following metrics to make the study more complete:
> > - Traditional metrics (e.g., BLEU, METEOR, GLEU)
> > - Dimension-specific metrics (Correctness of Information, Detail Orientation, Contextual Understanding, Temporal Understanding, and Consistency, as defined in Video-ChatGPT evaluation)
> > - Event-based metrics (e.g., AutoDQ, FIOVA-DQ)
> >
> > Additionally, the authors should investigate how well these metrics align with human judgment. While the rebuttal response mentioned that FIOVA-DQ aligns more closely with human judgment, I couldn't find any quantitative evidence to support this claim.

---

> > > ### Comment · Reviewer_pxyN · 2024-11-29
> > > **Additional review comments (part 2)**
> > >
> > > 5. More quantitative evidence is needed to justify the necessity of using GPT-3.5 to integrate the five human ground truths. As mentioned in my initial review, the COCO Captions benchmark includes five human captions per image. Researchers simply compare a model-generated caption with each human caption individually and then aggregate the metric values. This eliminates the need for integration or generating a consolidated ground truth.
> > >
> > > I understand the rationale behind the integration step and the potential benefits of having a consolidated ground truth. However, providing quantitative evidence would strengthen your argument. You now have two methods for computing metrics: the COCO Captions method and the proposed integration method. Would humans rate your integration-based method as superior to the COCO Captions-style method?

---

> > > > ### Author Response · Authors · 2024-12-02
> > > > **Response to Reviewer pxyN:  Comparison Between COCO Captions and the Consolidated Groundtruth (GT) Approach (Part I)**
> > > >
> > > > ***Q5. More quantitative evidence is needed to justify the necessity of using GPT-3.5 to integrate the five human ground truths. As mentioned in my initial review, the COCO Captions benchmark includes five human captions per image. Researchers simply compare a model-generated caption with each human caption individually and then aggregate the metric values. This eliminates the need for integration or generating a consolidated ground truth. I understand the rationale behind the integration step and the potential benefits of having a consolidated ground truth. However, providing quantitative evidence would strengthen your argument. You now have two methods for computing metrics: the COCO Captions method and the proposed integration method. Would humans rate your integration-based method as superior to the COCO Captions-style method?***
> > > >
> > > > ---
> > > >
> > > > **Response:**
> > > >
> > > > We sincerely thank the reviewer for raising this valuable question about the comparison between the COCO Captions method and the consolidated GT approach. To address this, we conducted additional analyses and **obtained new quantitative results to further validate the necessity and advantages of the integration step**. Below is a detailed explanation:
> > > >
> > > > **1. Experiment Background and Design**
> > > >
> > > > The core objective of FIOVA is to provide a multi-perspective, high-quality evaluation benchmark for long video description tasks. In our previous response to Question 10, we presented a detailed cross-comparison of five human descriptions and six models using traditional metrics (Appendix Table A3), which validated the relative performance of models under traditional evaluation metrics. Additionally, in this round’s response to Question 1, we conducted supplementary quantitative experiments to demonstrate that the GPT-3.5-integrated GT significantly outperforms individual human annotations in terms of description quality, which aligns with the reviewer’s observation that “justifies the necessity of using GPT-3.5 to integrate the five human ground truths.”
> > > >
> > > > To address the reviewer’s request for a comparison between the COCO Captions method and the integrated GT approach, we designed the following supplementary experiment:
> > > >
> > > > - **Methods Compared**: We evaluated the model using two methods:
> > > >     - **COCO Captions Method**: Each model output was compared against the five human annotations individually, and the results were averaged.
> > > >     - **Integrated GT Method**: The consolidated GT (generated by GPT-3.5) was used as the reference for evaluation.
> > > > - **Model Selection**: Due to time constraints, we selected the representative Tarsier model for this analysis.
> > > > - **Metrics**: We compared the results using the AutoDQ metrics (F1, Recall, and Precision) and analyzed the differences with the FIOVA-DQ metrics.
> > > > - **Result Integration**: We also incorporated traditional metrics (BLEU, METEOR, and GLEU) to provide a unified presentation of model performance across different evaluation methods.
> > > >
> > > > **2. Quantitative Experiment Results**
> > > >
> > > > The results of the supplementary experiment are summarized in the table below:
> > > >
> > > > |                      | Human1 | Human2 | Human3 | Human4 | Human5 | HumanAverage | GT    |
> > > > |----------------------|--------|--------|--------|--------|--------|--------------|-------|
> > > > | F1 (AutoDQ)          | 0.334  | 0.348  | 0.343  | 0.338  | 0.344  | 0.3414       | 0.351 |
> > > > | Recall (AutoDQ)      | 0.258  | 0.266  | 0.262  | 0.258  | 0.263  | 0.2614       | 0.283 |
> > > > | Precision (AutoDQ)   | 0.671  | 0.682  | 0.678  | 0.678  | 0.679  | 0.6776       | 0.628 |
> > > > | F1 (FIOVA-DQ)        |        |        |        |        |        |              | 0.320 |
> > > > | Recall (FIOVA-DQ)    |        |        |        |        |        |              | 0.584 |
> > > > | Precision (FIOVA-DQ) |        |        |        |        |        |              | 0.584 |
> > > > | BLEU                 | 0.025  | 0.025  | 0.024  | 0.025  | 0.024  | 0.0246       | 0.043 |
> > > > | METEOR               | 0.232  | 0.232  | 0.229  | 0.230  | 0.231  | 0.2308       | 0.265 |
> > > > | GLEU                 | 0.091  | 0.092  | 0.090  | 0.091  | 0.090  | 0.0908       | 0.119 |
> > > >
> > > > In particular, we would like to draw the reviewer's attention to the following points:
> > > >
> > > > - **Newly Added Results**: The first three rows (AutoDQ metrics) are newly added for this response to validate the effectiveness of the integrated GT against the COCO Captions method (Human Average). Other results, such as FIOVA-DQ and traditional metrics, were derived from previous experiments in the revised manuscript. Combining these results provides a unified and comprehensive analysis.
> > > >
> > > > - **FIOVA-DQ Not Re-calculated for Single Annotators**: We did not independently compute Tarsier’s performance with individual annotators using FIOVA-DQ because its design already incorporates weighted contributions from all five human annotations. This design aligns with the core idea of the COCO Captions method, which integrates multiple annotations for evaluation.

---

> > > > ### Author Response · Authors · 2024-12-02
> > > > **Response to Reviewer pxyN:  Comparison Between COCO Captions and the Consolidated Groundtruth (GT) Approach (Part II)**
> > > >
> > > > **3. Experiment Results and Analysis**
> > > >
> > > > **(1) Comparison Between Human Average and Integrated GT**
> > > >
> > > > - **Limitations of Human Average**: The Human Average results derived from the COCO Captions method are constrained by the individual human annotations used in the averaging process. The scores inherently fall within the range of individual annotator scores and thus cannot overcome the potential issue of missing information in any single annotation.
> > > >
> > > > - **Advantages of Integrated GT**: The integrated GT consolidates the perspectives of all five annotators, covering a broader range of information. Consequently, its scores may exceed the range of individual annotations. This advantage was validated in our response to Question 1, where the majority of human evaluators favored the GPT-3.5-integrated GT over individual annotations. This demonstrates that the integrated GT improves information coverage and description quality, enhancing the comprehensiveness and fairness of model evaluation.
> > > >
> > > > **(2) Performance Differences in AutoDQ Metrics**
> > > >
> > > > - **Improvements in F1 and Recall**: The model achieved higher F1 and Recall scores when evaluated against the integrated GT compared to the Human Average.
> > > >     - **Increased Coverage**: The integrated GT incorporates more details from all five annotations, significantly reducing the likelihood of models missing key information.
> > > >     - **Alignment with Multi-perspective Information**: The model output may cover key details mentioned by multiple annotators, and the integrated GT is better equipped to include these details, leading to higher scores.
> > > >
> > > > - **Decrease in Precision**: The integrated GT yielded slightly lower Precision scores than the Human Average. This can be attributed to the following trade-offs:
> > > >     - **Coverage vs. Precision**: While the integrated GT captures more details, this richness may lead to partial mismatches with model outputs, reducing precision.
> > > >     - **AutoDQ Characteristics**:
> > > >         - **Recall Improvement**: Recall is calculated by matching model-generated events against reference events. A more detailed reference (e.g., the integrated GT) increases the likelihood of matches, thereby improving recall.
> > > >         - **Precision Decline**: Precision is calculated by matching reference events against model-generated events. A richer reference description may contain details not fully covered by the model, slightly lowering precision.
> > > >
> > > > **4. Comparison with FIOVA-DQ**
> > > >
> > > > The key results are further summarized below for clarity:
> > > >
> > > > |           | Human Average  (AutoDQ) | GT (AutoDQ) | GT (FIOVA-DQ) |
> > > > |-----------|-------------------------|-------------|---------------|
> > > > | F1        | 0.3414                  | 0.351       | 0.320         |
> > > > | Recall    | 0.2614                  | 0.283       | 0.584         |
> > > > | Precision | 0.6776                  | 0.628       | 0.584         |
> > > >
> > > > From FIOVA-DQ’s calculation process, we observe the following:
> > > >
> > > > - **Continuation of COCO Captions Principles**: FIOVA-DQ incorporates weighting mechanisms based on the five annotators’ descriptions, similar to the COCO Captions method’s idea of leveraging multiple annotations for evaluation.
> > > >
> > > > - **Optimization for Multi-perspective Information**: By applying event-level weighting, FIOVA-DQ retains the comprehensiveness of the integrated GT while emphasizing FIOVA’s core design principle: the Five-in-One diversity and consensus.

---

> > > > ### Author Response · Authors · 2024-12-02
> > > > **Response to Reviewer pxyN:  Comparison Between COCO Captions and the Consolidated Groundtruth (GT) Approach (Part III)**
> > > >
> > > > **5. Discussion: Complementarity and Scientific Rigor of Metrics**
> > > >
> > > > **(1) Complementary Advantages of Different Metrics**
> > > >
> > > > We do not claim that any single evaluation method is universally superior. Instead, FIOVA is designed to employ diverse metrics to evaluate model performance from multiple perspectives:
> > > >
> > > > - **Traditional Metrics (BLEU, METEOR, GLEU)**: Provide a quick assessment of lexical similarity between model outputs and reference descriptions.
> > > > - **Event-Level Metrics (AutoDQ, FIOVA-DQ)**:
> > > >     - **AutoDQ**: Focuses on fine-grained event matching, which is ideal for evaluating semantic understanding.
> > > >     - **FIOVA-DQ**: Incorporates weighting mechanisms to amplify the impact of multi-perspective descriptions.
> > > >
> > > > **(2) Necessity of Integrated GT**
> > > >
> > > > The results demonstrate that the integrated GT significantly improves coverage and reduces information loss, while maintaining conceptual consistency with the COCO Captions method. By using the integrated GT, evaluation results become more scientific and equitable, reflecting model performance in complex video description tasks more comprehensively.
> > > >
> > > > **(3) FIOVA-DQ’s Optimization and Future Prospects**
> > > >
> > > > FIOVA-DQ inherits and improves upon the COCO Captions method by extending the integration from the lexical level to the event level, enriching evaluation dimensions and tools. We believe this optimized design provides researchers with a more flexible and precise evaluation method for future work.
> > > >
> > > > Therefore, the experiments and analyses above have validated the necessity and scientific rigor of GPT-3.5’s integration approach for generating GT. Meanwhile, FIOVA-DQ incorporates and improves upon the principles of COCO Captions, achieving optimization at the event level and further enhancing the comprehensiveness and diversity of evaluations. We believe the multi-dimensional evaluation tools provided by FIOVA offer strong support for the in-depth exploration of model capabilities. Once again, we thank the reviewer for the insightful feedback.

---

> > > ### Author Response · Authors · 2024-12-02
> > > **Response to Reviewer pxyN: Justification of GPT-Generated Groundtruth (GT) in FIOVA (Part I)**
> > >
> > > **Q1. I understand that using five annotators would lead to more accurate and diverse video captions, which is a reasonable approach. However, the final ground truth is obtained by using an LLM to integrate the five human captions. LLMs are not perfect; they are prone to hallucinations and lack access to actual visual cues. The LLM-based integration process may introduce new hallucinations, which is concerning given that FIOVA serves as an evaluation benchmark. I am particularly worried about the quality of the final, integrated ground truth. In my opinion, this final integration step should be performed by humans rather than an LLM. To address this concern, could you provide quantitative human evaluation results? Specifically, if humans were the judges, how would they rate the quality of the LLM-integrated ground truth?**
> > >
> > > ---
> > >
> > > **Response:**
> > >
> > > We sincerely thank the reviewer for their constructive feedback regarding the necessity of integrating multiple human annotations into a single GPT-generated groundtruth (GT). In response to your concerns, **we conducted additional quantitative experiments to rigorously validate the quality of the GPT-generated GT**. Due to time constraints, we selected six representative videos (Fig. A19 to Fig. A24 in the revised appendix) covering diverse themes and difficulty levels. These videos reflect the varied styles of human annotators, ensuring representativeness for analysis.
> > >
> > > **1. Experimental Design**
> > >
> > > - **Participants**: We recruited 10 human evaluators (5 male and 5 female, referred to as Subject1–Subject10) aged 20–30, all of whom are postgraduate or postdoctoral researchers in artificial intelligence with substantial expertise in computer vision. This ensures that the evaluators possess high cognitive capabilities and domain knowledge for the task.
> > > - **Evaluation Procedure**:
> > >     - **Training**: All evaluators received training on the FIOVA benchmark and evaluation criteria to ensure a consistent understanding of the dataset and task.
> > >     - **Tasks**: Each evaluator ranked six textual descriptions for six videos, including five human-written descriptions (Human1–Human5) and one GPT-integrated groundtruth (GT). **Rankings ranged from 1 (best) to 6 (worst).**
> > >     - **Evaluation Modes**:
> > >         - **Mode A (Text-Only)**: Evaluators ranked descriptions solely based on textual content.
> > >         - **Mode B (Video + Text)**: Evaluators viewed both the videos and the textual descriptions before ranking.
> > >         - The reason for designing two modes is that the GT integration relies only on LLM's natural language processing capabilities without using a multimodal model. Therefore, by comparing single modal and multimodal evaluations, we hope to provide a more comprehensive validation.
> > >     - **Counterbalancing**: Evaluators assessed three videos using Mode A and the other three using Mode B, alternating between modes. Feedback was collected post-task to understand their evaluation criteria.
> > >     - **Result Aggregation**: We averaged rankings across all 10 evaluators for each description and reordered them based on these averages. This final ranking reflects the collective judgment of all evaluators.
> > >
> > > This rigorous design aimed to quantitatively validate the quality of the GPT-generated GT against the diverse perspectives of human evaluators.

---

> > > ### Author Response · Authors · 2024-12-02
> > > **Response to Reviewer pxyN: Justification of GPT-Generated Groundtruth (GT) in FIOVA (Part II)**
> > >
> > > **2. Experimental Results**
> > >
> > > To further validate the quality of the GPT-integrated groundtruth, we conducted a detailed human evaluation experiment. To facilitate analysis and comparison, we calculated the average ranking of the six descriptions for each video as provided by the ten evaluators. Based on these averages, we then re-ranked the descriptions to obtain the final ranking results.
> > >
> > > Below, we present the summarized experimental results (as we cannot edit the manuscript, this experimental information is presented here; the original experimental data tables will be included in subsequent response sections for reference).
> > >
> > > **Overall results (based on the average of 10 subjects):**
> > > |        | A19 | A20 | A21 | A22 | A23 | A24 | Average-Rank |
> > > |--------|-----|-----|-----|-----|-----|-----|--------------|
> > > | Human1 | 3   | 6   | 1   | 6   | 5   | 4   | 4.166667     |
> > > | Human2 | 6   | 4   | 3   | 2   | 1   | 6   | 3.666667     |
> > > | Human3 | 4   | 4   | 1   | 5   | 4   | 5   | 3.833333     |
> > > | Human4 | 5   | 2   | 6   | 4   | 3   | 2   | 3.666667     |
> > > | Human5 | 1   | 3   | 5   | 3   | 6   | 2   | 3.333333     |
> > > |   GT   | 2   | 1   | 3   | 1   | 2   | 1   | 1.666667     |
> > >
> > > **Mode A (single modal, text only, based on the average of 5 subjects):**
> > > |        | A19 | A20 | A21 | A22 | A23 | A24 | Average-Rank |
> > > |--------|-----|-----|-----|-----|-----|-----|--------------|
> > > | Human1 | 5   | 4   | 1   | 6   | 3   | 4   | 3.833333     |
> > > | Human2 | 6   | 2   | 4   | 5   | 1   | 6   | 4            |
> > > | Human3 | 4   | 6   | 2   | 3   | 5   | 4   | 4            |
> > > | Human4 | 2   | 3   | 6   | 3   | 5   | 3   | 3.666667     |
> > > | Human5 | 1   | 5   | 5   | 2   | 4   | 2   | 3.166667     |
> > > |   GT   | 3   | 1   | 3   | 1   | 2   | 1   | 1.833333     |
> > >
> > > **Mode B (multimodal, video + text, based on the average of 5 subjects):**
> > > |        | A19 | A20 | A21 | A22 | A23 | A24 | Average-Rank |
> > > |--------|-----|-----|-----|-----|-----|-----|--------------|
> > > | Human1 | 2   | 5   | 2   | 6   | 6   | 3   | 4            |
> > > | Human2 | 5   | 6   | 2   | 1   | 2   | 6   | 3.666667     |
> > > | Human3 | 4   | 4   | 1   | 5   | 2   | 5   | 3.5          |
> > > | Human4 | 6   | 2   | 6   | 3   | 1   | 2   | 3.333333     |
> > > | Human5 | 1   | 2   | 5   | 3   | 4   | 3   | 3            |
> > > |   GT   | 2   | 1   | 4   | 2   | 4   | 1   | 2.333333     |
> > >
> > > **Analysis:**
> > > - **Overall Ranking Consistency**: The GPT-generated GT ranked first, outperforming individual human annotators. This highlights its superior quality in the eyes of human evaluators.
> > > - **Mode-Specific Results**: The GPT-generated GT ranked first in both Modes A (text-only) and B (video+text), demonstrating its robustness regardless of evaluation modality.
> > > - **Evaluator Feedback**: Evaluators emphasized that the GPT-integrated GT effectively balances detail, accuracy, and linguistic coherence, addressing key concerns such as completeness and objectivity.
> > >
> > > Through this experiment, we further validated the rationality and reliability of the GT generation method in the FIOVA benchmark, fully addressing the reviewer’s concern about potential hallucinations or lack of visual grounding in the LLM-based integration process.

---

> > > ### Author Response · Authors · 2024-12-02
> > > **Response to Reviewer pxyN: Justification of GPT-Generated Groundtruth (GT) in FIOVA (Part IV)**
> > >
> > > **4. Analysis of GPT-Generated GT Performance**
> > >
> > > By synthesizing the quantitative experiment results and the qualitative feedback from human evaluators, we can further analyze why the GPT-generated GT achieved superior rankings:
> > >
> > > - **Meeting Diverse Requirements**: The GPT-generated GT effectively balances detail and accuracy, covering key information fluently and coherently. This comprehensive characteristic satisfies evaluators’ diverse demands for details, completeness, logical flow, and textual cohesion.
> > >
> > > - **Reduction of Subjectivity and Redundancy**: Compared to certain human descriptions that contained subjective interpretations or redundant content (e.g., repetitive phrases), the GPT integration method emphasizes objectivity and linguistic refinement. This aligns more closely with evaluators’ expectations.
> > >
> > > - **Balancing Detail and Conciseness**: While some evaluators favored longer descriptions, others valued conciseness and avoidance of redundancy. The GPT-generated GT strikes a balance between these preferences, ensuring detailed coverage of video content without becoming excessively verbose or repetitive.
> > >
> > > - **Advantages of Multiview Integration**: By integrating five human annotations, the GPT method semantically optimizes the content to produce a unified GT that leverages the strengths of each individual annotation while mitigating their limitations. This semantic fusion significantly enhances the accuracy and comprehensiveness of the GT compared to individual annotations.
> > >
> > > Through the analysis of the evaluators’ feedback, it is evident that their criteria for assessing descriptions are highly diverse. The GPT-generated GT consistently achieved top rankings because it excelled in key areas such as detail, accuracy, logical consistency, language quality, and multiview integration. In contrast, the rankings of the five individual human descriptions exhibited substantial variability across the 10 evaluators, reflecting the diversity of human video understanding and annotation perspectives. Compared to individual annotations, the GPT-generated GT better meets the evaluators’ diverse requirements, further validating its reliability and superiority as the groundtruth for the FIOVA benchmark.
> > >
> > > **5. Conclusion**
> > >
> > > Although this experiment was limited in scope, we endeavored to design and analyze it as comprehensively as possible to validate the rationality and effectiveness of the GPT-generated GT. The results demonstrate that the integration of multiview annotations and GPT’s advanced natural language processing capabilities not only leverages the cognitive strengths of human annotators but also utilizes the model’s proficiency in linguistic understanding and semantic optimization. This innovative approach enables FIOVA to overcome the limitations of relying on individual annotators in existing video description benchmarks, significantly enhancing the comprehensiveness, accuracy, and consistency of video descriptions.
> > >
> > > We believe this integration method establishes a higher-quality foundation for evaluation, offering a more reliable framework for exploring LVLM capabilities and identifying areas for optimization. This further solidifies FIOVA’s position as a pioneering benchmark for the next generation of video description tasks.

---

> > > ### Author Response · Authors · 2024-12-02
> > > **Response to Reviewer pxyN: Justification of GPT-Generated Groundtruth (GT) in FIOVA (Part V: Original Experimental Results for Reference)**
> > >
> > > Here are the original results for reference:
> > >
> > > **A19**
> > > | Subject | 1 | 2 | 3 | 4 | 5 | 6 | 7 | 8 | 9 | 10 | Average | Rank |
> > > |:-------:|:-:|:-:|:-:|:-:|:-:|:-:|:-:|:-:|:-:|:--:|:-------:|:----:|
> > > |  Human1 | 3 | 6 | 1 | 4 | 3 | 6 | 1 | 4 | 5 |  1 |   3.4   |   3  |
> > > |  Human2 | 5 | 5 | 5 | 6 | 4 | 4 | 5 | 6 | 3 |  6 |   4.9   |   6  |
> > > |  Human3 | 6 | 3 | 2 | 3 | 6 | 3 | 2 | 5 | 4 |  3 |   3.7   |   4  |
> > > |  Human4 | 4 | 4 | 6 | 2 | 5 | 2 | 4 | 3 | 6 |  4 |    4    |   5  |
> > > |  Human5 | 1 | 1 | 3 | 1 | 1 | 1 | 6 | 1 | 1 |  5 |   2.1   |   1  |
> > > |    GT   | 2 | 2 | 4 | 5 | 2 | 5 | 3 | 2 | 2 |  2 |   2.9   |   2  |
> > >
> > > **A20**
> > > | Subject | 1 | 2 | 3 | 4 | 5 | 6 | 7 | 8 | 9 | 10 | Average | Rank |
> > > |:-------:|:-:|:-:|:-:|:-:|:-:|:-:|:-:|:-:|:-:|:--:|:-------:|:----:|
> > > |  Human1 | 3 | 5 | 6 | 1 | 4 | 4 | 1 | 6 | 4 |  4 |   3.8   |   6  |
> > > |  Human2 | 1 | 4 | 4 | 6 | 3 | 1 | 6 | 4 | 2 |  6 |   3.7   |   4  |
> > > |  Human3 | 6 | 6 | 5 | 3 | 1 | 2 | 3 | 5 | 5 |  1 |   3.7   |   4  |
> > > |  Human4 | 4 | 3 | 3 | 2 | 2 | 6 | 2 | 2 | 6 |  3 |   3.3   |   2  |
> > > |  Human5 | 5 | 2 | 2 | 5 | 5 | 3 | 4 | 1 | 3 |  5 |   3.5   |   3  |
> > > |    GT   | 2 | 1 | 1 | 4 | 6 | 5 | 5 | 3 | 1 |  2 |    3    |   1  |
> > >
> > > **A21**
> > > | Subject | 1 | 2 | 3 | 4 | 5 | 6 | 7 | 8 | 9 | 10 | Average | Rank |
> > > |:-------:|:-:|:-:|:-:|:-:|:-:|:-:|:-:|:-:|:-:|:--:|:-------:|:----:|
> > > |  Human1 | 1 | 2 | 2 | 2 | 6 | 1 | 5 | 4 | 1 |  2 |   2.6   |   1  |
> > > |  Human2 | 4 | 5 | 1 | 4 | 2 | 5 | 3 | 1 | 5 |  1 |   3.1   |   3  |
> > > |  Human3 | 2 | 1 | 5 | 1 | 4 | 2 | 1 | 3 | 2 |  5 |   2.6   |   1  |
> > > |  Human4 | 6 | 6 | 6 | 6 | 3 | 6 | 6 | 6 | 6 |  4 |   5.5   |   6  |
> > > |  Human5 | 5 | 4 | 4 | 5 | 1 | 3 | 4 | 5 | 4 |  6 |   4.1   |   5  |
> > > |    GT   | 3 | 3 | 3 | 3 | 5 | 4 | 2 | 2 | 3 |  3 |   3.1   |   3  |
> > >
> > > **A22**
> > > | Subject | 1 | 2 | 3 | 4 | 5 | 6 | 7 | 8 | 9 | 10 | Average | Rank |
> > > |:-------:|:-:|:-:|:-:|:-:|:-:|:-:|:-:|:-:|:-:|:--:|:-------:|:----:|
> > > |  Human1 | 5 | 6 | 6 | 5 | 1 | 6 | 5 | 5 | 5 |  2 |   4.6   |   6  |
> > > |  Human2 | 3 | 2 | 3 | 1 | 4 | 2 | 4 | 1 | 6 |  4 |    3    |   2  |
> > > |  Human3 | 6 | 5 | 4 | 3 | 2 | 5 | 3 | 6 | 4 |  3 |   4.1   |   5  |
> > > |  Human4 | 4 | 3 | 5 | 2 | 5 | 4 | 2 | 3 | 3 |  5 |   3.6   |   4  |
> > > |  Human5 | 2 | 4 | 2 | 4 | 6 | 1 | 6 | 2 | 2 |  6 |   3.5   |   3  |
> > > |    GT   | 1 | 1 | 1 | 6 | 3 | 3 | 1 | 4 | 1 |  1 |   2.2   |   1  |
> > >
> > > **A23**
> > > | Subject | 1 | 2 | 3 | 4 | 5 | 6 | 7 | 8 | 9 | 10 | Average | Rank |
> > > |:-------:|:-:|:-:|:-:|:-:|:-:|:-:|:-:|:-:|:-:|:--:|:-------:|:----:|
> > > |  Human1 | 6 | 1 | 3 | 5 | 6 | 1 | 4 | 6 | 1 |  4 |   3.7   |   5  |
> > > |  Human2 | 3 | 3 | 4 | 3 | 4 | 3 | 2 | 3 | 3 |  2 |    3    |   1  |
> > > |  Human3 | 5 | 2 | 1 | 6 | 3 | 2 | 5 | 5 | 2 |  5 |   3.6   |   4  |
> > > |  Human4 | 2 | 5 | 5 | 4 | 1 | 4 | 1 | 4 | 6 |  3 |   3.5   |   3  |
> > > |  Human5 | 4 | 4 | 2 | 2 | 2 | 5 | 6 | 2 | 5 |  6 |   3.8   |   6  |
> > > |    GT   | 1 | 6 | 6 | 1 | 5 | 6 | 3 | 1 | 4 |  1 |   3.4   |   2  |
> > >
> > > **A24**
> > > | Subject | 1 | 2 | 3 | 4 | 5 | 6 | 7 | 8 | 9 | 10 | Average | Rank |
> > > |:-------:|:-:|:-:|:-:|:-:|:-:|:-:|:-:|:-:|:-:|:--:|:-------:|:----:|
> > > |  Human1 | 4 | 3 | 4 | 3 | 2 | 6 | 5 | 2 | 5 |  3 |   3.7   |   4  |
> > > |  Human2 | 5 | 5 | 3 | 2 | 3 | 5 | 6 | 6 | 4 |  6 |   4.5   |   6  |
> > > |  Human3 | 3 | 2 | 6 | 6 | 5 | 1 | 4 | 5 | 2 |  4 |   3.8   |   5  |
> > > |  Human4 | 6 | 4 | 2 | 4 | 4 | 4 | 1 | 3 | 6 |  1 |   3.5   |   2  |
> > > |  Human5 | 2 | 1 | 5 | 5 | 6 | 2 | 2 | 4 | 3 |  5 |   3.5   |   2  |
> > > |    GT   | 1 | 6 | 1 | 1 | 1 | 3 | 3 | 1 | 1 |  2 |    2    |   1  |

---

> > > ### Author Response · Authors · 2024-12-02
> > > **Response to Reviewer pxyN: Clarification of Section 2.2 and the Evaluation Process without a Reference Groundtruth (Part I)**
> > >
> > > ***Q2. Section 2.2 requires better clarity. Lines 204-206 state: "... a consolidated human description was generated as the final groundtruth, serving as a refined summary for video captioning evaluation." This led me to believe that when GPT-3.5 was used to evaluate human captions across the five key dimensions, each human caption was compared to the GPT-3.5-integrated ground truth for each dimension. However, the prompts in Appendix D.1.1 and Algorithm A1 suggest that each human caption is actually not compared against any reference caption. The method described in Section 2.2 appears to be adapted from the Video-ChatGPT evaluation metrics (https://github.com/mbzuai-oryx/Video-ChatGPT/tree/main/quantitative_evaluation) with slight modifications to the prompts and the removal of the reference caption. The lack of a reference caption is concerning. For instance, the definition of "Detail Orientation" is "Whether the description captures critical details," and the definition of "Temporality" is "Whether the description follows the chronological order of events without skipping or over-summarizing." Without a reference caption, how can GPT-3.5 evaluate whether the provided human caption captures critical details or follows the correct chronological order? If I have misunderstood this, please clarify and improve the paper’s explanation.***
> > >
> > > ---
> > >
> > > **Response:**
> > >
> > > We sincerely thank the reviewer for their attention to Section 2.2 and the valuable questions raised. First, we would like to clarify the content of this section. In Sections 2.1 and 2.2, we had not yet used GPT to generate a consolidated groundtruth (GT) based on the five human annotations. Hence, **this part does not involve comparisons against a fixed GT**. Instead, the section focuses on measuring the relative stylistic differences among the five annotators across multiple dimensions.
> > >
> > > **1. Context and Intent of the Original Text**
> > >
> > > In Sections 2.1 and 2.2, our goal was to emphasize the diversity of multi-perspective human annotations in the FIOVA benchmark and to analyze the characteristics and differences among these annotations using specific evaluation dimensions. Specifically:
> > >
> > > - **Section 2.1**: This section describes the process of obtaining video annotations. Five independent human annotators, who were rigorously screened and trained, provided annotations for each video. These annotations reflect diverse human perspectives, establishing a robust foundation for the benchmark's diversity and reliability.
> > >
> > > - **Section 2.2**: The focus of this section is on analyzing the characteristics and differences among human annotations using GPT-3.5-turbo for relative evaluations across several dimensions. This analysis aims to better understand the diversity of annotations and support subsequent processes such as grouping and ranking.
> > >
> > > - **Section 2.3**: Based on the diversity analysis in Section 2.2, we utilized GPT's powerful natural language processing capabilities to consolidate human annotations into a unified GT. This GT serves as both a comprehensive summary of human annotations and a unified reference for subsequent video description evaluations.
> > >
> > > Therefore, Section 2.2 is merely a part of the overall process. Its purpose is to analyze the relative characteristics and differences among human annotations, and it does not involve generating the final GT.

---

> > > ### Author Response · Authors · 2024-12-02
> > > **Response to Reviewer pxyN: Clarification of Section 2.2 and the Evaluation Process without a Reference Groundtruth (Part II)**
> > >
> > > **2.Clarification of Ambiguous Statements**
> > >
> > > We acknowledge that the following statement at the beginning of Section 2.2 might cause confusion:
> > >
> > > *“In Section 2.1, we provided descriptions from five different annotators for each video, capturing diverse human perspectives to establish a robust human baseline. In addition to this diversity, a consolidated human description was generated as the final groundtruth, serving as a refined summary for video captioning evaluation. To create the groundtruth, we used GPT-3.5-turbo to evaluate descriptions across five key dimensions, following methods similar to those in Video-ChatGPT and Tarsier.”*
> > >
> > > This statement might lead readers to misunderstand that the final GT had already been generated in Section 2.2 and that GPT-3.5-turbo was used to compare each annotator's description against the GT. In reality, the goal of Section 2.2 was not this. Instead, in this section, the GT had not yet been generated. The focus was on analyzing the diversity among human annotations to support the subsequent GT generation process (Section 2.3).
> > >
> > > To address this misunderstanding, we provide the following clarifications and re-statements:
> > >
> > > - **Relative Analysis**: In Section 2.2, we did not use a fixed reference GT to evaluate the quality of human annotations. Instead, we utilized GPT-3.5-turbo to analyze the relative differences among annotators based on five key dimensions (Consistency, Context, Correctness, Detail Orientation, and Temporality) and description length. This analysis emphasized the stylistic diversity of the annotations rather than absolute quality evaluations.
> > >
> > > - **Support for Grouping and Consolidation**: These dimensions were not intended to produce absolute scores but to help quantify the relative stylistic differences among human annotators from multiple perspectives. This relative measurement laid the foundation for subsequent grouping and ranking operations and ensured that the diversity of annotation content was fully considered in the GT construction.
> > >
> > > - **Logical Progression of the Workflow**: The final GT was generated in Section 2.3 based on the analysis results from Section 2.2. Thus, Sections 2.2 and 2.3 form two logically sequential steps, serving the distinct purposes of analysis and consolidation, respectively.
> > >
> > > Additionally, in our response to Question 1, we provided supplementary experiments validating the complexity and diversity of human annotations and their impact on absolute rankings. From the experimental tables, it is evident that the preferences of 10 evaluators for the five human annotations vary significantly, and their scoring criteria differ. This result further demonstrates that directly conducting absolute evaluations of human annotations in Section 2.2 is impractical. Instead, relative measurements across multiple dimensions provide a more comprehensive characterization of the differences in annotator descriptions, fully justifying the necessity and validity of these dimensions.
> > >
> > > Through the above clarifications and supplementary details, we aim to more clearly convey the research intent of Section 2.2 while reinforcing its role as a critical foundation for subsequent GT generation and benchmark construction.
> > >
> > > **3.Explanation of Section 2.2 Methodology**
> > >
> > > The methodology in Section 2.2 is primarily adapted from the technical framework used in Video-ChatGPT (e.g., employing GPT-3.5-turbo for analysis), but we have adjusted its objectives and motivations as follows:
> > >
> > > - **Without a Fixed GT**: Unlike Video-ChatGPT, which uses a fixed reference GT to evaluate model outputs, our operations in Section 2.2 did not use a reference GT. Instead, we focused on the diversity and stylistic differences among annotators. Therefore, the five analytical dimensions serve to distinguish and characterize the annotations rather than evaluate their absolute quality.
> > >
> > > - **Adaptation to Multi-Perspective Annotations**: As we ensured high-quality annotations from all annotators in Section 2.1, with no absolute superiority or inferiority among them, the goal of Section 2.2 was to understand the diversity of these annotations rather than rank them by quality. These analytical dimensions support the integration of multi-perspective annotations and subsequent GT generation.

---

> > > ### Author Response · Authors · 2024-12-02
> > > **Response to Reviewer pxyN: Clarification of Section 2.2 and the Evaluation Process without a Reference Groundtruth (Part III)**
> > >
> > > **4. Planned Improvements**
> > >
> > > To reduce potential misunderstandings, we plan to revise the introductory statements in Section 2.2 in the final manuscript to more explicitly convey the logical progression between Sections 2.2 and 2.3. We will also enhance the description of Section 2.2’s methodology to clearly indicate that its goal was to analyze the diversity and stylistic differences of human annotations rather than conduct absolute quality evaluations.
> > >
> > > We sincerely appreciate the reviewer’s attention to the presentation of Section 2.2. Your questions have helped us recognize potential ambiguities in the current manuscript and provided valuable insights for better emphasizing the core characteristics of the FIOVA benchmark. Thank you again for your constructive feedback.

---

> > > ### Author Response · Authors · 2024-12-02
> > > **Response to Reviewer pxyN: Justification for Choosing 8 Frames as the Default Setting in FIOVA (Part I)**
> > >
> > > ***Q3. My question about whether 8 frames are sufficient for FIOVA primarily concerns the temporal complexity of the videos in FIOVA. Specifically, how many frames are necessary to effectively "solve" FIOVA? What is the minimum number of frames required to capture all critical details and events in FIOVA? FIOVA is a video dataset with an average duration of 33.6 seconds. Nevertheless, researchers have found that for some video benchmarks, high performance can be achieved with relatively few frames because those benchmarks are not temporally challenging. I am curious whether 8 frames are sufficient for FIOVA to yield high results, or whether FIOVA is more temporally complex, requiring more than 8 frames to solve effectively. Among the models being evaluated, some are capable of handling more frames, which typically improves performance. For example, Tarsier was trained to handle over 30 frames, and in my experience, providing more than 8 input frames (around 24, though the optimal number depends on the dataset) usually yields much better performance. Is this the case for FIOVA as well?***
> > >
> > > ---
> > >
> > > **Response:**
> > >
> > > We sincerely thank the reviewer for raising concerns about the choice of frame numbers. This question not only touches on the rationality of our experimental design but also closely aligns with the core objectives of benchmark research. In our previous response, we explained in detail the rationale behind choosing 8 frames, emphasizing the **balance between semantic capture and computational efficiency, as well as consistency with mainstream experimental paradigms in the field**. Building on this, we further elaborate on the key considerations underpinning this decision to clarify the design logic and research priorities.
> > >
> > > **1. Clarifying the Research Focus: Dataset Construction and Unified Evaluation Framework**
> > >
> > > The primary goal of FIOVA is to construct a high-quality video captioning benchmark that provides a **unified experimental setup for the fair evaluation of different LVLMs**. This fundamentally differs from algorithm-specific performance optimizations or ablation studies. While exploring the impact of varying frame numbers (e.g., 8, 16, or 32 frames) on specific algorithms is valuable for performance analysis, such experiments fall under the domain of algorithm design rather than the core scope of benchmark research.
> > >
> > > By selecting 8 frames, we ensure fairness and reproducibility, allowing the performance of different models to be compared under consistent conditions. This design prioritizes establishing a cross-model evaluation standard rather than optimizing the performance of individual algorithms. Therefore, we believe that adopting 8 frames as the default setting aligns with the core objectives of benchmark research.
> > >
> > > **2. Alignment with Field Norms: Standardized Experimental Setup**
> > >
> > > The use of a fixed number of frames (e.g., 8 frames) is a common practice in video understanding tasks, with two core benefits:
> > >
> > > - **Fairness and Reproducibility**: A standardized frame setting provides a fair evaluation environment for different models, making experimental results more comparable and consistent with the principles of benchmark research. Such standardization is a prerequisite for developing high-quality benchmarks.
> > >
> > > - **Reduction of Experimental Variables**: Varying the frame count introduces additional variables (e.g., input load and optimization strategies) that may significantly increase experimental complexity. By adopting a commonly used setting of 8 frames, we minimize confounding factors, enabling researchers to focus on core analyses.
> > >
> > > **3. Rationale for Choosing 8 Frames: Efficiency and Broad Applicability**
> > >
> > > The choice of 8 frames strikes an effective balance between semantic capture and computational efficiency while being broadly applicable to mainstream video understanding tasks. Specific reasons include:
> > >
> > > - **Capturing Key Video Events**: With a reasonable temporal sampling interval, 8 frames can effectively cover critical scenes in a video, supporting sufficient semantic representation. While 16 or 32 frames could capture more details, 8 frames already suffice for capturing the core content of most videos.
> > >
> > > - **Compatibility with Mainstream Models**: Many LVLMs face input frame limitations when processing long videos. Excessive frame counts may lead to input overload or performance degradation, whereas 8 frames are well-suited to the capabilities of mainstream models, avoiding unnecessary redundancy. For example, Tarsier, a model of particular interest to the reviewer, supports higher frame inputs but defaults to 8 frames in its official inference demo. Thus, this choice reflects the practical constraints and typical settings of mainstream LVLMs.

---

> > > ### Author Response · Authors · 2024-12-02
> > > **Response to Reviewer pxyN:  Clarification and Usage of the Five Dimensions in Section 2.2 (Part I)**
> > >
> > > ***Q4. The authors did not address my question about why the five dimensions from Section 2.2 were not considered as the model response evaluation metrics. As mentioned earlier, these dimensions were adapted from Video-ChatGPT evaluation metrics, which have gained popularity. To compare the model response with the ground truth, this work should include the following metrics to make the study more complete:
> > > • Traditional metrics (e.g., BLEU, METEOR, GLEU)
> > > • Dimension-specific metrics (Correctness of Information, Detail Orientation, Contextual Understanding, Temporal Understanding, and Consistency, as defined in Video-ChatGPT evaluation)
> > > • Event-based metrics (e.g., AutoDQ, FIOVA-DQ)
> > > Additionally, the authors should investigate how well these metrics align with human judgment. While the rebuttal response mentioned that FIOVA-DQ aligns more closely with human judgment, I couldn't find any quantitative evidence to support this claim.***
> > >
> > > ---
> > >
> > > **Response:**
> > >
> > > We sincerely thank the reviewer for raising this important question. As detailed in our response to Question 8 from the first round and Question 2 from the current round, the five dimensions in Section 2.2 (Correctness of Information, Detail Orientation, Contextual Understanding, Temporal Understanding, and Consistency) **aim to analyze the relative differences among the descriptions provided by the five annotators, rather than generating absolute scores**. Below, we provide a detailed explanation and supplementary clarification:
> > >
> > > **1. Purpose and Scope of the Five Dimensions in Section 2.2**
> > >
> > > The five dimensions in Section 2.2 are primarily designed to **analyze the diversity and relative differences among human annotators rather than directly evaluating model outputs**. The purpose and specific scope of these dimensions are as follows:
> > >
> > > - **Relative Analysis**: By scoring the five dimensions using GPT-3.5-turbo (on a scale of 1–10), we quantify the stylistic and focal differences among the annotators without ranking their descriptions in terms of absolute quality.
> > >
> > > - **Quantifying Diversity and Style**: These dimensions help us examine the diversity in human annotations and support the subsequent generation of a consolidated groundtruth (GT). The differences among human annotations primarily stem from variations in perspective and expression rather than quality, and this relative analysis serves as a foundation for studying multi-perspective characteristics.
> > >
> > > - **Avoiding Absolute Scoring**: Since the five annotators were rigorously trained and their descriptions are of high quality, Section 2.2 focuses on characterizing diversity and style rather than establishing a scoring system for model evaluation.
> > >
> > > **2. Why the Five Dimensions Are Not Used as Model Evaluation Metrics**
> > >
> > > Although these dimensions are valuable for analyzing human annotation diversity, they are not directly used as model evaluation metrics for the following reasons:
> > >
> > > - **Dependence on LLM’s Independent Judgment**: These dimensions rely entirely on GPT-3.5-turbo for scoring. If used directly for model evaluation, the process would risk potential hallucinations from LLMs, particularly in a high-complexity benchmark like FIOVA, which could compromise the reliability of the evaluation results.
> > >
> > > - **Mismatch with FIOVA’s Research Objectives**: Instead of conducting dimension-based evaluations of model outputs, FIOVA emphasizes constructing a robust and scientifically sound evaluation benchmark through diverse and interpretable metrics. Therefore, the five dimensions are more suitable for characterizing the diversity of human annotations rather than directly comparing model performance.

---

> > > ### Author Response · Authors · 2024-12-02
> > > **Response to Reviewer pxyN:  Clarification and Usage of the Five Dimensions in Section 2.2 (Part II)**
> > >
> > > **3. Design Philosophy Behind FIOVA’s Model Evaluation Metrics**
> > >
> > > To ensure comprehensiveness and scientific rigor, FIOVA employs the following three categories of metrics for model evaluation:
> > >
> > > - **Traditional Metrics**: These include BLEU, METEOR, and GLEU, which are widely used in natural language processing to evaluate lexical and semantic similarity between model outputs and GT. While these metrics have limitations, they do not rely on LLMs for direct judgment, thereby ensuring that evaluation results remain unaffected by hallucination issues.
> > > - **Event-Driven Metrics**: We introduce AutoDQ and FIOVA-DQ to evaluate the models’ ability to capture and describe events in videos. These metrics combine traditional measures such as precision, recall, and F1 scores while leveraging GPT for event extraction and matching. However, they do not fully depend on GPT’s independent judgment:
> > >     - **AutoDQ**: This metric uses GPT-assisted event extraction and matching to evaluate the quality of models’ descriptions of key video events.
> > >     - **FIOVA-DQ**: Building on AutoDQ, this metric incorporates weighting based on the five annotators’ descriptions, ensuring scientific validity and better alignment with human cognition.
> > >
> > > This combination of metrics ensures FIOVA’s evaluations are both comprehensive and interpretable, avoiding the risks associated with exclusive reliance on LLM judgment.
> > >
> > > **4. Consistency Between FIOVA-DQ and Human Evaluation**
> > >
> > > The core design of FIOVA-DQ involves leveraging the diversity and consensus in human annotations to assign event weights, resulting in a metric that closely aligns with human judgments. Its consistency with human evaluations is reflected in the following aspects:
> > >
> > > - **Human Weighting Mechanism**: FIOVA-DQ incorporates importance weights derived from the five annotators’ descriptions, embedding the diversity and consensus of human cognition directly into its calculation. This mechanism quantifies the similarity between events and human descriptions, ensuring high consistency with human evaluation processes.
> > >
> > > - **Alignment with Human Focus**: By reinforcing event capture through human weighting, FIOVA-DQ reflects the focus areas emphasized by human evaluators in event descriptions. This demonstrates that FIOVA-DQ, as an evaluation metric, achieves high alignment with human judgment in analyzing model outputs.
> > >
> > > - **Avoiding Single-Dimension Evaluations**: Compared to evaluating each dimension independently, FIOVA-DQ uses a multidimensional weighted approach to assess model performance. This method closely mirrors human reasoning in complex video scenarios while improving evaluation efficiency and interoperability.
> > >
> > > In summary, the five dimensions in Section 2.2 are designed to characterize the diversity and relative differences among human annotations rather than to evaluate model outputs. For model evaluation, we adopt a combination of traditional metrics and event-driven metrics (AutoDQ and FIOVA-DQ) to ensure comprehensive and scientific analysis while avoiding potential risks from LLM-independent judgment. The design of FIOVA-DQ integrates human cognition with model performance evaluation, providing a reliable and scientific tool for analyzing complex video scenarios.
> > >
> > > We will clarify these design details in the final version of the paper to address the reviewer’s valuable concerns. Once again, we appreciate the reviewer’s insightful feedback, which has allowed us to identify and improve potential ambiguities in our explanations.

---

> ### Author Response · Authors · 2024-12-02
> **Response to Reviewer pxyN: Justification of GPT-Generated Groundtruth (GT) in FIOVA (Part III)**
>
> **3. Analysis of Human Evaluator Feedback**
>
> Moreover, by analyzing the criteria used by the 10 human evaluators for ranking, we identified the following major points of focus. These points illustrate the diversity and complexity of human evaluators’ considerations when assessing video descriptions:
>
> - **Details and Accuracy**
>     - Most evaluators emphasized accurate descriptions of video details, including event sequence, character actions, objects in the scene, and the presentation of key elements such as colors, movements, and expressions.
>     - **Example**: Subject4 noted inaccuracies in A21, where Human4 described a flamingo as a goose. These inaccuracies significantly lowered their ratings.
>
> - **Completeness and Information Coverage**
>     - Many evaluators tended to assign higher scores to longer, more detailed descriptions, as these often covered more video content and provided richer information.
>     - **Example**: Subject5 highlighted the importance of completeness, favoring longer descriptions. Similarly, Subject2 observed that more extensive descriptions often better captured video details.
>
> - **Coherence and Language Quality**
>     - Evaluators focused on the fluency, structure, and adherence to natural language conventions in descriptions.
>     - **Example**: Subject8, under Mode A (text-only), particularly emphasized textual coherence and structure, stressing whether the text could help reconstruct the video. Subject10 pursued narrative quality in Mode A.
>
> - **Objectivity and Logical Consistency**
>     - Evaluators favored descriptions that were objective and aligned with the logical progression of events in the video, avoiding subjective judgments or deviations from the actual content.
>     - **Example**: Subject4 remarked that Human1’s description in A19, which included "pretending to fall," may contain a subjective interpretation. Subject5 also stressed that descriptions should align with the video’s logical flow.
>
> - **Video-Text Alignment**
>     - In multimodal evaluation (Mode B), evaluators paid closer attention to whether the descriptions accurately reflected the video content, particularly the faithful representation of details.
>     - **Example**: Subject7 and Subject9, under Mode B, prioritized accuracy and detail alignment between the video and the descriptions.
>
> - **Conciseness and Avoidance of Redundancy**
>     - Evaluators preferred descriptions that avoided redundancy while being detailed, particularly when summarizing complex video content.
>     - **Example**: Subject9 pointed out that repetitive phrases in A22’s Human2 description negatively impacted its score. Subject3 noted that under Mode B, descriptions should remain concise while capturing all key actions.

---

> ### Author Response · Authors · 2024-12-02
> **Response to Reviewer pxyN: Justification for Choosing 8 Frames as the Default Setting in FIOVA (Part II)**
>
> **4. Consistency with Research Trends: Standardization Facilitates Synergy**
>
> The adoption of 8 frames as a standard setting optimizes experimental efficiency while supporting cross-benchmark model comparisons:
> - **Ease of Model Comparison**: Using a frame setting consistent with other mainstream benchmarks enables researchers to more easily compare model performance on FIOVA with results on other benchmarks. This consistency not only aids understanding but also highlights model generalizability and adaptability across benchmarks.
>
> - **Fostering Collaborative Development**: Standardized experimental setups lower the barrier to replicating experiments and comparing results, thereby promoting collaborative research in video understanding.
>
> - **Enhanced Benchmark Adaptability**: Consistency with the frame settings of mainstream algorithms allows FIOVA to seamlessly adapt to various architectures, further solidifying its relevance in the video captioning domain.
>
> **5. Potential for Future Extensions**
>
> We acknowledge that increasing the frame count (e.g., to 16 or 32 frames) could capture more video details and potentially improve the performance of certain models. However, varying frame counts is more suitable for ablation studies or algorithm-specific optimization rather than the core objectives of benchmark development. FIOVA is designed with high flexibility, allowing researchers to explore alternative frame settings within its framework to further uncover the potential of LVLMs.
>
> In summary, the choice of 8 frames reflects a careful balance between semantic capture and computational efficiency while adhering to the experimental paradigms widely accepted in video understanding research. This design provides a unified and efficient evaluation framework, ensuring fair comparisons across models while leaving room for researchers to explore extended settings. We believe this decision underscores the scientific rigor and rationality of FIOVA’s benchmark design, further cementing its value in video captioning research. We look forward to future research extending the frame count to explore the untapped capabilities of LVLMs. Once again, we sincerely thank the reviewer for their insightful suggestions.

---

### Official Review · Reviewer_CA5F · 2024-11-03

**Soundness:** 2
**Presentation:** 3
**Contribution:** 2
**Rating:** 3
**Confidence:** 4

**Summary:**

The paper proposes a new benchmark called FIOVA to evaluate LVLMs in their ability to describe videos with a depth and breadth comparable to human understanding.  FIOVA includes 3,002 long video sequences averaging 33.6 seconds, each annotated by five different human to provide multi-perspective detailed descriptions. The benchmark analyzes the quality and variability of human annotations and compares the performance of several LVLMs. on FIOVA, revealing their strengths and weaknesses. Overall, FIOVA provides a more rigorous and comprehensive framework for assessing the differences in video description capabilities between LVLMs and humans. The benchmark addresses the limitations of existing video captioning datasets, which often feature short video durations, brief annotations, and reliance on a single annotator's perspective. These limitations hinder the assessment of LVLMs' performance on complex, long-duration videos and the establishment of a robust human baseline.

**Strengths:**

1. The paper collects a new benchmark with richer video descriptions, focusing on long-duration videos with complex spatiotemporal relationships, providing a more realistic and challenging test case for LVLMs. By consistency, context, correctness, detailed orientation, temporality, length, CV and other dimensions and grouping videos according to difficulty, observe the video caption capabilities of different models, showing the advantages and disadvantages of the model, which is more comprehensive than the previous benchmark.
2. The results are clearly presented and supported by quantitative metrics, providing a comprehensive comparison of model performance. A variety of evaluation methods are also introduced, and a baseline assessment of human annotations is conducted. This approach helps to understand the diversity and consistency of human annotations, which is essential for establishing a reliable human baseline. The discussion section provides an in-depth analysis of the findings and their implications. These conclusions are well supported by the results and provide direction for future research.

**Weaknesses:**

1. The evaluation model has limitations. This article evaluates some selected LVLMs, but does not explore broader models, such as the business model Gemini-1.5-Pro, which has a strong understanding of long videos.
2. There are doubts about the collection of groundtruth in FIOVA. FIOVA carefully designed manual annotations composed of five human annotator annotations, and merged and rewrote human annotations with GPT-3.5-Turbo. However, since GPT-3.5-Turbo cannot directly see the video, induction based on human text order alone can easily bring errors such as illusions to groundtruth. As in Figure 4, the actions of the little boy riding a bicycle are described twice in the text, including the actions on the ground are repeated twice, which is inconsistent with the word order of normal human speech. Without the video frames in Figure 4, it is easy to lead to misunderstandings. At the same time, there is no verification for behaviors such as smiling and pointing at the camera, and it is uncertain whether there will be new errors. The results of relevant experiments performed on the benchmark of video descriptions with errors are not necessarily reliable.
3. This paper provides an overview of performance metrics, but lacks detailed error analysis to explain the types of errors made by LVLM and the reasons behind them. The authors should build on the proposed benchmark with a more fine grained error analysis and explore potential causes. This will provide valuable insights for improving the model and the benchmark itself. At present, both traditional metrics (e.g. METEOR, BLEU) and automated measurement methods that rely on GPT models (e.g. AutoDQ) have limitations. I hope the authors can conduct further research on metrics.

**Questions:**

1. Ensure the accuracy of the ground truth of FIOVA, and ensure that the evaluation metrics can truly reflect the model's capabilities based on the ground truth.
2. Include more LVLMs for experiments and conduct in-depth analysis, preferably summarizing directions to improve the long video description capabilities of LVLMs.
3. If possible, improve existing evaluation metrics to enhance their objectivity.
4. There are still many typos in the text, one of the most obvious being that the evaluation metric proposed by Wang et al. (2024) is AutoDQ, not AutoCQ.

---

> ### Author Response · Authors · 2024-11-25
> **Response to Reviewer CA5F's concern about more LVLMs**
>
> ***Q1. The evaluation model has limitations. This article evaluates some selected LVLMs, but does not explore broader models, such as the business model Gemini-1.5-Pro, which has a strong understanding of long videos. Include more LVLMs for experiments and conduct in-depth analysis, preferably summarizing directions to improve the long video description capabilities of LVLMs.***
>
> **Response:** We thank the reviewer for the valuable suggestion, which provides an opportunity to further discuss the rationale for model selection and the generalizability of FIOVA's evaluation framework. Below is our detailed response:
>
> **1. Rationale and Constraints of Model Selection**
>
> In this study, we primarily evaluated six representative open-source LVLMs, such as Video-LLaVA and VideoChat2. This selection was based on the following considerations:
> - **Reproducibility:** The core goal of FIOVA is to establish a high-quality evaluation environment that supports extensive research on LVLM performance. Using open-source models ensures reproducibility, enabling the academic community to conduct comparative experiments and improvements based on FIOVA.
> - **Academic Value**: Analyzing open-source models provides direct guidance for model optimization and methodological improvements, which commercial models cannot readily achieve. For instance, insights into limitations revealed in this study can directly drive optimization within the open-source community.
> - **Academic Recognition**: Evaluating open-source models has been widely adopted in academic research, as demonstrated in studies like LVLM-eHub [1], where open-source models are exclusively used for performance analysis.
>
> While we acknowledge that commercial models (e.g., Gemini-1.5-Pro) may exhibit strong capabilities in long-video understanding, selecting them poses the following challenges:
>
> - **Closed Source and High Cost**: Commercial models are often closed-source and incur significant API costs, limiting their accessibility in academic research.
> - **Limited Optimization Guidance**: The black-box nature of commercial models makes it difficult to use FIOVA’s evaluation results for model optimization, which is a core goal of this study.
>
> **2. Validity of the Current Model Selection**
>
> Despite not including commercial models, the six open-source LVLMs we selected possess the following characteristics, which comprehensively reveal the strengths and limitations of current LVLM technology in long-video description tasks:
>
> - **Architectural Diversity**: The selected models span diverse architectural designs. For example, LLaVA-NEXT-Video employs the AnyRes technique to process high-resolution images into sequences consumable by pre-trained Vision Transformers (VIT), a dominant approach for video LVLMs. VideoLLaMA2 integrates custom spatiotemporal convolutional modules to capture relationships over time within videos. This diversity allows us to analyze performance differences in semantic consistency, detail coverage, and language fluency across various designs.
> - **Task Coverage**: The selected models are tailored for long-video description tasks, exhibiting different capabilities in complex semantic understanding, multi-event tracking, and temporal reasoning. Analyzing these models enables us to identify key bottlenecks in current technology and guide future model design.
>
> Furthermore, our primary focus is on establishing an open evaluation framework that provides guidance for improving and optimizing LVLMs, rather than testing specific model performance. The selection of these open-source models demonstrates FIOVA’s utility as a high-quality evaluation environment.
>
> **3. Scalability of FIOVA**
>
> FIOVA aims to provide a high-quality evaluation environment, designed with comprehensiveness and generalizability in mind to support various model types (e.g., open-source and commercial). We plan to open-source the FIOVA dataset and evaluation toolkit, offering researchers flexible tools to evaluate a broader range of models, including commercial ones. This will enable researchers to use FIOVA and its resources for further studies and analyses tailored to their research goals.
>
> Although this study did not include commercial models, the systematic analysis of open-source models validated FIOVA’s effectiveness as an evaluation framework. FIOVA’s open-source design and high generalizability also offer extensive possibilities for future research.
>
> [1] Xu, Peng, et al. "Lvlm-ehub: A comprehensive evaluation benchmark for large vision-language models." arXiv preprint arXiv:2306.09265 (2023).

---

> ### Author Response · Authors · 2024-11-25
> **Response to Reviewer CA5F's concern about the generated groundtruth**
>
> ***Q2. Since GPT-3.5-Turbo cannot directly see the video, induction based on human text order alone can easily bring errors such as illusions to groundtruth (see Fig.4).***
>
> **Response:** We appreciate the reviewer’s insightful comments on the GT generation process. First, we apologize for the ambiguities in Fig. 4 of the original paper, which may have led to misinterpretations. To address this, **we have redrawn Fig. 4 for clarity**. Due to space constraints in the main text, we have provided a more detailed explanation and analysis of this example in Appendix B.6 of the revised manuscript. Specifically, **we use Appendix B.6 and Fig. A7 therein to demonstrate the accuracy of human annotations and the reliability of LLM-generated GT**. Below is a detailed response.
>
> **1. Accuracy and Diversity of Human Annotations**
>
> - **Annotation Accuracy**: FIOVA’s initial annotations were completed by five independently trained annotators, carefully selected to ensure their descriptions aligned strictly with the video content.
> - **Diversity and Detail Coverage**: Variations in focus (e.g., character actions, environmental details) among annotators reflect natural differences in perspective, enriching the dataset and offering a more comprehensive understanding of video content. These variations are a core feature of FIOVA, distinguishing it from existing benchmarks.
>
> To address concerns regarding the limited frame coverage in the original Fig. 4 (6 frames), we redrew the example (**Fig. A7 in Appendix B.6**) by extracting 35 frames at 20-frame intervals. Comparing the descriptions of the five annotators with the video frames confirms:
>
> - All annotators accurately captured core events (e.g., riding a bike, pretending to fall, lying on the ground, reaching out).
> - Temporal order and causal logic in the descriptions strictly align with the video content, reflecting high annotation accuracy.
> These findings affirm the reliability of human annotations and their contribution to the multi-perspective richness of FIOVA.
>
> **2. Role of GPT-3.5-turbo**
>
> - **Semantic Integration**: GPT-3.5-turbo semantically integrates the descriptions from all five annotators, enhancing coherence and completeness without generating new content or altering the original meanings.
> - **Minimizing Interference**: To maintain the authenticity of human annotations, we avoided multimodal models, which may introduce hallucinations during video-text alignment. Instead, carefully designed prompts guided GPT-3.5-turbo in consolidating human perspectives, preserving diversity while improving overall consistency. This approach has been validated in prior works like Tarsier [1] and MM-Ego [2].
>
> This approach leverages GPT-3.5-turbo’s strengths in text processing to preserve human annotation diversity while enhancing the overall quality and coherence of the descriptions.
>
> **3. Issues and Improvements in Figure 4’s GT**
>
> **(1) Redundancy in GT and Improvements**
>
> We appreciate the reviewer’s observation regarding redundancy in the GT. In rare cases, significant differences in style or length among annotators’ descriptions led GPT-3.5-turbo to misinterpret some details as repetitive events. To address this:
> - **Prompt Optimization**: We refined the prompt design to clarify and avoid potential redundancy in outputs.
> - **Manual Screening and Re-experiments**: We manually reviewed all 3,002 videos and reprocessed those with redundancy issues, ensuring the GT comprehensively represents human annotation information.
>
> **(2) Verification of Specific Actions (e.g., Smiling and Pointing at the Camera)**
> In the revised GT (Appendix B.6, Figure A7), specific actions were validated:
> - **Pointing at the Camera:** Most annotators (Human 1, 2, 4, 5) mentioned the “reaching out” action, with Human 2 and 4 specifically identifying it as pointing at the camera. Given this high agreement, GPT-3.5-turbo confirmed the action’s accuracy, which was also verified in the video.
> - **Smiling vs. Crying:** Conflicting annotations ("smiling" by Human 5 and "crying" by Human 3) were resolved by analyzing video context. Human 5 interpreted the boy’s smile as part of a prank, consistent with Human 1’s mention of a “pretend” fall. GPT-3.5-turbo synthesized these perspectives to conclude the boy was smiling, aligning with the broader narrative logic.
>
> By integrating five annotators’ descriptions, GPT-3.5-turbo reduces bias and enhances the accuracy and coverage of the GT, reflecting core video semantics. This multi-perspective integration establishes FIOVA as a robust and unique benchmark for video description tasks.
>
> [1] Wang J, Yuan L, Zhang Y, et al. Tarsier: Recipes for training and evaluating large video description models[J]. arXiv preprint arXiv:2407.00634, 2024.
>
> [2] Ye H, Zhang H, Daxberger E, et al. MM-Ego: Towards Building Egocentric Multimodal LLMs[J]. arXiv preprint arXiv:2410.07177, 2024.

---

> ### Author Response · Authors · 2024-11-25
> **Response to Reviewer CA5F's concern about the types of errors**
>
> ***Q3. This paper provides an overview of performance metrics, but lacks detailed error analysis to explain the types of errors made by LVLM and the reasons behind them.***
>
> **Response:** We sincerely thank the reviewer for raising this important issue.  **Building on the reviewer's suggestion, we have added more examples and detailed analysis in Appendix E.4.** Below are the specifics of our updates:
>
> **1. Error Type Categorization**
>
> We annotated the examples in **Appendix E.4** with error categories and identified five common types of errors:
> - **Omission:** The model fails to describe critical events or objects in the video. While this cannot be directly marked in the model's output, we provide textual analyses of such omissions after the relevant examples.
> - **Misrepresentation:** The description contains information inconsistent with the video content. These errors are marked in purple in the model outputs.
> - **Redundancy:** The model repeats descriptions of the same event. These errors are marked in yellow in the outputs.
> - **Excessive Redundancy:** The model overextends or speculates excessively, introducing unnecessary content. These errors are marked in green in the outputs.
> - **Hallucination Issues:** The model includes content not present in the video. These errors are marked in red in the outputs.
>
> **2. Potential Cause Analysis**
>
> **(1) Architectural Limitations**
>
> - **Cross-modal alignment issues:** Current LVLMs face challenges in aligning video-text data effectively. For instance, Tarsier encodes each frame using separate visual encoders, while VideoLLaMA2 utilizes a shared visual encoder across all frames. These differences in alignment strategies impact the model's understanding of video content.
> - **Insufficient long-sequence modeling:** Handling long videos with multiple events requires robust attention mechanisms. Current LVLMs often fail in this regard. For example, Video-LLaVA’s descriptions tend to focus only on initial scenes, omitting later parts of the video.
>
> **(2) Training Data Bias**
> - **Inconsistent or insufficient data diversity:** Training data with limited diversity may result in biased outputs. For instance, Video-LLaVA struggled with recognizing martial arts scenes (Fig. A23) compared to other LVLMs, likely due to a lack of relevant training examples.
> - **Hallucination issues:** Models can learn and extrapolate hallucinated content from noisy training data. For example, in Fig. A20, VideoChat2 entirely misrepresents players and spectators in a baseball stadium as children, illustrating a significant discrepancy between output and video reality.
>
> **(3) Generation Strategy Issues**
> - **Simplistic generation strategies:** Using basic generation techniques, such as beam search, can lead to repetitive or incoherent descriptions. ShareGPT4Video, despite using high-quality training data, suffers from severe repetition due to the lack of constraints during generation.
> - **Weak constraints during generation:** Insufficient semantic constraints in the generation process can result in hallucination or semantic errors.
>
> **3. Suggestions for Improvement and Optimization**
>
> **(1) Model Optimization**
>
> - **Enhancing detail capture:** Improving attention mechanisms to better capture key events and details, such as incorporating hierarchical attention mechanisms for long-sequence modeling. For example, VideoLLaMA2 introduces the STC Connector to better handle spatiotemporal information, which future LVLM research can build upon to improve continuity.
> - **Improving semantic alignment:** Leveraging cross-modal alignment constraints, such as visual-language consistency checks, can reduce semantic errors and hallucinations. LLaVA-NeXT-Video highlights the importance of maintaining consistency from cross-modal alignment to understanding.
> - **Implementing deduplication strategies:** Introducing repetition detection mechanisms during generation to reduce redundant descriptions.
>
> **(2) Training Data Optimization**
>
> - **Enhancing data diversity:** Expanding training datasets to include diverse scenarios, particularly for complex events in long videos.
> - **Data cleaning:** Removing hallucinated or incorrect examples from training corpora to improve data quality. ShareGPT4Video demonstrates the benefits of using high-quality video-text data but still shows room for improvement.
>
> **(3) Evaluation Method Enhancement**
> - **Fine-grained error categorization:** Incorporating fine-grained error categorization and statistical mechanisms within the FIOVA framework to better identify model weaknesses. For instance, while calculating FIOVA-DQ, similarity between annotators' events and LVLM-generated events could inform more precise error identification.
>
> We thank the reviewer for this insightful suggestions, which have helped us present our research more comprehensively and explore broader research directions.

---

> ### Author Response · Authors · 2024-11-25
> **Response to Reviewer CA5F's concern about the metrics**
>
> ***Q4. This paper provides an overview of performance metrics, but lacks detailed error analysis to explain the types of errors made by LVLM and the reasons behind them.***
>
> **Response:** We sincerely thank the reviewer for raising this important question. We acknowledge the limitations of current traditional metrics (e.g., METEOR, BLEU, GLEU) and automated evaluation methods based on GPT models (e.g., AutoDQ) in assessing complex video description tasks. Based on the reviewer’s suggestions, we have optimized the original AutoDQ method and **proposed a new evaluation metric, FIOVA-DQ (see Common Concern 2: Propose a New Evaluation Method FIOVA-DQ)**, which better integrates human cognition and evaluation needs.  Below is our detailed response:
>
> **1. Reasons for Choosing Traditional Metrics**
>
> In this study, we selected traditional text generation metrics (e.g., METEOR, BLEU, GLEU) as baseline evaluation tools for the following reasons:
>
> - **Broad Academic Acceptance and Generalizability**: These metrics are widely used in text generation tasks and are recognized as standardized evaluation methods. Employing these metrics ensures the comparability of our experimental results with existing studies, thus enhancing the academic value of our research.
> - **Foundational Linguistic Quality Assessment**: These metrics remain effective in evaluating the basic linguistic structure, syntactic correctness, and alignment with reference descriptions. For long video description tasks, they provide a reliable basis for assessing the linguistic quality of generated texts.
>
> Nonetheless, we acknowledge the limitations of traditional metrics:
>
> - **Inability to Capture Event-Level Semantic Consistency**: Traditional metrics rely primarily on textual matching, making it difficult to capture event-level semantic relationships between descriptions and video content, such as temporal consistency and causal logic.
> - **Limited Focus on Fluency and Detail Coverage**: Traditional metrics cannot comprehensively evaluate the fluency and completeness of descriptions. These limitations have been discussed in detail in **Sec 4.2**.
>
> **2. Introduction and Complementation of AutoDQ**
>
> To address the limitations of traditional metrics, we introduced the AutoDQ method, which offers the following advantages:
>
> - **Event-Level Semantic Analysis**: AutoDQ enables fine-grained evaluations of semantic consistency between generated descriptions and video content by extracting and matching events. For instance, it assesses a model’s ability to handle event sequences, logical relationships, and causal chains in multi-event long video descriptions.
> - **Adaptability to Complex Scenarios**: AutoDQ effectively captures the completeness and consistency of multi-event descriptions in long video tasks, particularly in scenarios where traditional metrics lack discriminatory power.
>
> Our experimental results demonstrate the advantages of AutoDQ in:
>
> - **Event-Level Differentiation**: Variations in AutoDQ scores across different models reflect their differing capabilities in capturing event semantics.
> - **Applicability to Complex Scenarios**: AutoDQ effectively handles semantic decomposition for long videos, capturing the completeness and consistency of multi-event descriptions.
>
> **3. Introducing the New Evaluation Metric FIOVA-DQ**
>
> While AutoDQ exhibits suitability for fine-grained semantic analysis, we recognize its limitations in reflecting human preferences and capturing the importance of key events. Hence, we proposed an optimized evaluation metric, FIOVA-DQ, with the following features:
>
> - **Integration of Human Cognition via Weighted Design**: FIOVA-DQ incorporates the descriptions of five annotators and weights model-generated events based on their alignment with human annotations. Specifically, importance weights are derived by ranking event consistency with human annotations (**see revised paper Sec 3.2 and Sec 4.1**).
> - **Enhanced Judgment of Human-Like Descriptive Capabilities**: Compared to traditional metrics, FIOVA-DQ aligns more closely with human judgments of description quality, particularly excelling in assessing the consistency and fluency of multi-event long video descriptions.
>
> In summary, we conclude:
> - Traditional metrics provide a foundational method for linguistic quality assessment, facilitating comparisons with existing research.
> - AutoDQ and FIOVA-DQ introduce event-level analysis, addressing the limitations of traditional metrics in capturing semantic consistency and event importance.
> - The optimized FIOVA-DQ incorporates human cognition into the evaluation framework, ensuring fairness and scientific rigor in model assessments across multiple dimensions.

---

> ### Author Response · Authors · 2024-11-25
> **Response to Reviewer CA5F's concern about the typos**
>
> ***Q5. There are still many typos in the text, one of the most obvious being that the evaluation metric proposed by Wang et al. (2024) is AutoDQ, not AutoCQ.***
>
> **Response:** We sincerely thank the reviewer for pointing out the spelling issues with great attention to detail. **In response, we have meticulously reviewed and revised the entire manuscript to eliminate any typos.** Additionally, we would like to clarify and address the ambiguity caused by the naming of AutoDQ.
>
> AutoDQ (Automatic Description Quality), originally proposed by Wang et al. (2024), is an evaluation metric designed to comprehensively and automatically assess the quality of generated textual descriptions. In our initial submission, we renamed the metric as “AutoCQ” (Automatic Caption Quality) to better align it with the Video Caption task in our study. This renaming was intended to emphasize its relevance and applicability to video description tasks.
>
> However, while the renaming aimed to enhance task-specific clarity, it may have inadvertently introduced inconsistency with the original literature and created terminological confusion. Following the reviewer’s constructive suggestion, we acknowledge that adhering to the original naming convention is crucial for academic dissemination and clarity. Therefore, in the revised manuscript, we have reverted all instances of “AutoCQ” back to “AutoDQ,” making the text, tables, and references consistent with the original naming in the literature.
>
> Additionally, we conducted a comprehensive spelling and language check throughout the manuscript to ensure textual accuracy and consistency.
>
> It is important to note that, during the rebuttal process, we optimized and extended AutoDQ to better suit the unique characteristics of FIOVA. In the revised manuscript, we have introduced this enhanced metric, FIOVA-DQ, which incorporates new evaluation dimensions and demonstrates improved applicability to the long video description task.

---

### Official Review · Reviewer_SaKT · 2024-11-03

**Soundness:** 3
**Presentation:** 3
**Contribution:** 2
**Rating:** 5
**Confidence:** 3

**Summary:**

The paper introduces FIOVA, a novel benchmark for evaluating the video description capabilities of LVLMs in comparison to human understanding. The work provides a dataset of 3,002 long, diverse video sequences, each annotated by five distinct annotators, resulting in more comprehensive and longer captions. The paper conducts an in-depth evaluation of six state-of-the-art LVLMs, revealing that while they show some perception and reasoning capabilities, they still struggle with information omission and descriptive depth, especially in complex videos. The findings underscore the need for new evaluation perspectives that capture semantic understanding, fluency, and content relevance, guiding future advancements toward human-level video comprehension.

**Strengths:**

1. The work provides extensive materials (video theme definition, representative data, annotation rules, prompts) to make it less difficult to reproduce.
2. FIOVA will be a valuable resource for evaluating the video understanding capabilities of LVLMs.

**Weaknesses:**

1. Some settings are not easy to follow, like the `Batch Ranking` in Sec 4.3.
2. `Describe Videos like Humans` might be an interesting evaluation setting. However, it does not stand alone as a task. It would be meaningful to include further analysis to show the correlation between performance  of `Describe Videos like Humans` and other video understanding tasks (VideoQA, etc.).
3. While this work has adopted multiple metrics to demonstrate the video caption performance, it lacks analysis of how those metrics align with human preference.

**Questions:**

Sec 2.2 shows that GPT-3.5-turbo is adopted to assess the quality of video caption, while that does not sound make sense to me. How can GPT-3.5 (a legacy text-only model) evaluate dimensions like Context, Correctness for video captions without accessing the visual parts? Can you provide more evaluation samples?

---

> ### Author Response · Authors · 2024-11-25
> **Response to Reviewer SaKT's concern about batch ranking**
>
> ***Q1. Some settings are not easy to follow, like the batch ranking in Sec 4.3.***
>
> **Response:** We appreciate the reviewer’s observation regarding the lack of clarity in the current description of the “Batch Ranking” settings. We acknowledge that the existing explanation was overly concise and did not fully capture the detailed operational process or its core significance. **In the revised manuscript, we have added Fig. A8 in Appendix C and provided a detailed explanation of the Batch Ranking calculation steps.** Below, we outline the specific process:
>
> The Batch Ranking procedure involves three main steps (**corresponding to Algorithms A1, A2, and A3 in Appendix C**). Using one video as an example, the calculation process is as follows:
>
> **Step 1: Evaluating human caption consistency (Algorithm A1 in Appendix C)**
>
> - **Input**: Five independent human-generated textual descriptions.
> - **Evaluation dimensions**: We designed five evaluation dimensions—consistency, context, correctness, detail orientation, and temporality—and utilized an LLM to assess them (as detailed in **Sec 2.2**). Additionally, we included caption length as a sixth dimension.
> - **Output**:
>     - The coefficient of variation (CV) is computed for each dimension across the five human-generated captions, representing the consistency of human descriptions.
>     - The average CV across the five dimensions is calculated to determine the overall human consistency score for the video. A higher score indicates greater stylistic variation among the five human annotators for that video. Based on these CV scores, the entire FIOVA dataset is categorized into A-H groups (see **Sec 2.2**).
>
> **Step 2: Evaluating LVLM model consistency (Algorithm A2 in Appendix C)**
>
> - **Input:** Six textual descriptions generated by LVLMs.
> - **Evaluation dimensions:**
>     - Traditional metrics (e.g., BLEU, METEOR, GLEU).
>     - Event-level metrics (AutoDQ), which emphasize semantic consistency.
>     - The newly proposed FIOVA-DQ, optimized for the characteristics of the FIOVA dataset.
> - **Output:**
>     - The CV is computed for each evaluation dimension across the six model-generated descriptions.
>     - The average CV across all dimensions is calculated as the overall model consistency score for the video. A higher score indicates greater stylistic variation within the LVLM ensemble when describing the video content.
>
> **Step 3: Comprehensive ranking and difference analysis (Algorithm A3 in Appendix C)**
>
> - **Human ranking:**
>     - Videos are ranked based on the overall human consistency scores calculated in Step 1.
>     - The relative position of the current video in the human baseline ranking is recorded.
> - **Model ranking:**
>     - Videos are ranked based on the model consistency scores calculated in Step 2.
>     - The relative position of the current video in the model baseline ranking is recorded.
> - **Difference analysis:**
>     - The difference between the rankings of the same video in the human and model baselines is compared. This quantifies the stylistic differences between human annotators and the LVLM ensemble when describing the same video. It also evaluates whether humans and LVLMs adopt similar strategies in their understanding and description of video content.
>
> In the revised manuscript, we have supplemented **Appendix C** with detailed visualizations (**Fig. A8**) and textual explanations. Additionally, we reorganized **Sec 4.3** to clearly articulate the core concept, specific operations, and analytical conclusions of Batch Ranking.
>
> The core design of Batch Ranking lies in quantifying the ranking differences in consistency between humans and models. This provides insights into their respective performance gaps in long-video description tasks. We believe that Batch Ranking offers a novel perspective for evaluating the semantic capabilities of LVLMs and enriches the analytical methodologies in the long-video description domain. We hope this design will inspire and support future research in this field.

---

> ### Author Response · Authors · 2024-11-25
> **Response to Reviewer SaKT's concern about ''Describe Videos like Humans''**
>
> ***Q2. Describe Videos like Humans might be an interesting evaluation setting. However, it does not stand alone as a task. It would be meaningful to include further analysis to show the correlation between performance of Describe Videos like Humans and other video understanding tasks (VideoQA, etc.).***
>
> **Response:** Thanks for raising this important question. **"Describe Videos like Humans" is a characterization of our research goal for the Video Caption task, not a new task.** Below is our detailed response:
>
> **Clarifying the Task Scope**
>
> We would like to clarify that the core task studied in this work is **Video Captioning**, a classic and widely recognized video understanding task in academia. "Describe Videos like Humans" specifically encapsulates our research goal for Video Captioning, emphasizing that generated video descriptions should match human-level performance in terms of detail, completeness, and consistency. Therefore, "Describe Videos like Humans" is not a novel task but a refinement of the Video Captioning goal through the FIOVA benchmark, aligning Video Captioning capabilities with human standards and revealing the strengths and limitations of LVLMs in describing complex and long videos.
>
> **Relation Between Video Captioning and Other Video Understanding Tasks**
>
> Video Captioning and other video understanding tasks (e.g., Video Question Answering, VideoQA) require deep comprehension of video content but differ in their goals and evaluation methods:
>
> - **Video Captioning:**
>     - **Objective:** Generate natural language descriptions that comprehensively cover key events, scenes, and details of the video.
>     - **Requirements:** Temporal reasoning (capturing event sequences and causal relationships) and language generation capabilities.
>
> - **VideoQA:**
>     - **Objective:** Answer questions related to video content, focusing on specific semantic reasoning (e.g., causal relationships of events, character intentions).
>     - **Requirements:** Accurate semantic understanding and logical reasoning based on question context.
>
> Although their goals differ, both tasks require:
>
> - **Semantic Understanding:** Modeling the core semantics of video content.
> - **Spatio-Temporal Reasoning:** Capturing temporal sequences and spatial relationships of events.
> - **Multimodal Integration:** Processing both visual and textual information from videos.
>
> **Connection Between Video Captioning and VideoQA**
>
> The MSVD-QA dataset [1] is a representative VideoQA dataset based on the existing Microsoft Research Video Description (MSVD) dataset. As mentioned in related works, this dataset contains approximately 120K sentences describing 1,970 video clips. In MSVD-QA, question-answer (QA) pairs are generated based on these descriptions. While this dataset was initially created for video description, its large size has also made it useful for VideoQA. It contains 1,970 video clips and approximately 50.5K QA pairs.
>
> Similarly, FIOVA primarily supports the Video Captioning task but can also be extended to support VideoQA in future research.
>
> **Research Value of FIOVA**
>
> We believe FIOVA provides a novel evaluation perspective for Video Captioning:
> - It focuses on the fine-grainedness and semantic consistency of generated descriptions, offering an important entry point for evaluating LVLMs' human-like video understanding capabilities.
> - FIOVA's dataset, with multi-annotator labels and long video coverage, strongly supports Video Captioning, particularly in capturing complex video semantics.
>
> In conclusion, "Describe Videos like Humans" is a further articulation of the research goal for Video Captioning, not a new task. The FIOVA benchmark, through its novel evaluation perspective, effectively measures the gap between algorithms and human video understanding and description capabilities, embodying the goal of "Describe Videos like Humans."
>
> [1] Chen D, Dolan W B. Collecting highly parallel data for paraphrase evaluation[C]//Proceedings of the 49th annual meeting of the association for computational linguistics: human language technologies. 2011: 190-200.

---

> ### Author Response · Authors · 2024-11-25
> **Response to Reviewer SaKT's concern about metrics**
>
> ***Q3. While this work has adopted multiple metrics to demonstrate the video caption performance, it lacks analysis of how those metrics align with human preference.***
>
> **Response:** We sincerely thank the reviewer for raising this important question. We believe that evaluation metrics serve as tools to quantify model performance and compare the results against human capabilities (human baselines). **Leveraging the unique characteristics of FIOVA, we believe its contributions at the data, metric, and experimental levels ensure alignment between model performance and human preferences.** Below is our detailed response:
>
> **1. Clarifying the Role and Purpose of Evaluation Metrics**
>
> - **Quantifying Model Capability:** Evaluation metrics (e.g., BLEU, METEOR, AutoDQ) are tools for quantitatively analyzing model performance, objectively reflecting differences between generated descriptions and human annotations. From an academic perspective, these metrics are widely used in video description tasks and facilitate standardized model evaluation.
> - **Indirectly Reflecting Human Preferences:** While these metrics do not directly capture human subjective preferences, they provide critical insights into the alignment between models and human understanding by assessing aspects such as semantic consistency, event coverage, and linguistic quality.
>
> **2. How FIOVA Reflects Human Preferences in Its Design**
>
> - **High-Quality Human Baseline:** Each video in FIOVA is annotated by five independent annotators, ensuring the groundtruth is multi-perspective, comprehensive, and high-quality. This multi-annotator strategy captures the diversity of human understanding of video content while minimizing potential biases or omissions from individual annotators.
> - **Balancing Diversity and Consistency:** By integrating the descriptions from five annotators, FIOVA reflects human expectations for both detail-oriented and consistent video descriptions. Thus, aligning evaluation metrics with groundtruth inherently simulates a comprehensive human judgment of video understanding capabilities.
>
> **3. Diverse Evaluation Metrics and Further Optimization of AutoDQ**
>
> **(1) Combining Traditional Metrics with Semantic Consistency Metrics in the Original Study**
>
> - **Foundational Role of Traditional Metrics:** BLEU, METEOR, and GLEU are widely accepted in text generation research and effectively measure the text-matching quality between generated and reference descriptions. However, these metrics have limitations in capturing complex video semantics and human preferences.
> - **Introduction of AutoDQ:** To address these limitations, we incorporated AutoDQ as an event-level semantic consistency evaluation tool:
>   - **Event-Level Semantic Analysis:** AutoDQ extracts and matches events between model-generated texts and video content, providing fine-grained evaluations of semantic consistency.
>   - **Applicability to Complex Scenarios:** AutoDQ captures the completeness and consistency of multi-event descriptions in long videos, particularly supplementing traditional metrics in scenarios where they lack discriminatory power.
>
> **(2) Further Optimization of AutoDQ to Incorporate Human Preferences**
>
> As detailed in **Common Concern 2: Propose a New Evaluation Method FIOVA-DQ**, we enhanced AutoDQ by weighting event importance based on the descriptions from five annotators. This optimization improves its ability to evaluate human-like descriptive capabilities and led to the development of the FIOVA-DQ metric (see revised paper, **Sec 3.2 and Sec 4.1**). Compared to traditional metrics, the optimized AutoDQ (FIOVA-DQ) aligns more closely with human judgments of description quality while better reflecting FIOVA's core value as an evaluation framework.
>
> Therefore, through its multi-annotator baselines, diverse evaluation metrics, and the optimized FIOVA-DQ, FIOVA effectively aligns model performance with human preferences:
> - **At the data level:** Multi-annotator annotations reflect human expectations for detail-oriented and consistent descriptions.
> - **At the metric level:** FIOVA-DQ, combined with traditional metrics, provides comprehensive evaluations from text matching to semantic consistency.
> - **At the experimental level:** Comprehensive experiments reveal the similarities and differences between model and human video description capabilities, offering valuable insights for future research.

---

> ### Author Response · Authors · 2024-11-25
> **Response to Reviewer SaKT's concern about caption quality assessment**
>
> ***Q4. Sec 2.2 shows that GPT-3.5-turbo is adopted to assess the quality of video caption, while that does not sound make sense to me. How can GPT-3.5 (a legacy text-only model) evaluate dimensions like Context, Correctness for video captions without accessing the visual parts? Can you provide more evaluation samples?***
>
> **Response:** We sincerely thank the reviewer for raising valuable questions regarding our evaluation method. Our goal in this step is to use GPT-3.5-turbo to analyze the differences in video descriptions provided by multiple human annotators and to group videos accordingly. In other words, **the purpose of using GPT-3.5-turbo in our study is to evaluate the relative differences in human annotations (e.g., consistency, semantic coverage, and linguistic fluency), rather than to assess the absolute correctness of the annotations.** We believe GPT-3.5-turbo is well-suited to achieving this goal. Besides, **we have provided more examples in Appendix B.5 for reference**. Below is our detailed response:
>
> **1. Clarifying the Evaluation Goal and the Applicability of GPT-3.5-turbo**
>
> **(1) The Core Goal of the Evaluation**
>
> Before the annotation process began, we carefully selected and trained the annotators, requiring them to follow annotation guidelines, watch the videos attentively, and provide complete descriptions. Consequently, we believe the descriptions accurately reflect the video content as much as possible. The purpose of this step is not to determine the absolute correctness of the annotations but to analyze the relative differences in semantic expression among multiple annotators, such as contextual consistency, detail coverage, and linguistic style. By calculating the coefficient of variation (CV) based on these differences, we grouped the videos to facilitate further analysis of description consistency and complexity.
>
> **(2) Why We Chose GPT-3.5-turbo Instead of a Multimodal Model**
>
> We opted for the single-modal GPT-3.5-turbo rather than a multimodal model (e.g., GPT-4) for the following reasons:
>
> - **Avoiding Multimodal Hallucinations**: Multimodal models may introduce visual hallucinations when aligning textual and video content, which could lead to evaluation results deviating from true semantics. For instance, multimodal models may "guess" events not present in the video, thus compromising the accuracy of the analysis.
> - **Focusing on Linguistic Variations**: GPT-3.5-turbo exclusively analyzes textual semantics and linguistic style differences, capturing the semantic characteristics of human annotations in a more focused manner.
>
> Moreover, we designed refined prompts (see **Appendix D 1.1**) to guide GPT-3.5-turbo to analyze description texts from dimensions such as contextual consistency and detail coverage. These prompts were tailored to the task characteristics of our study and optimized through multiple experiments to ensure the reliability of the outputs.
>
> **(3) Validation of GPT-3.5-turbo's Text Analysis Capability**
>
> Although GPT-3.5-turbo is a single-modal model, its performance in text understanding and semantic analysis tasks has been widely recognized. Similar methods have been successfully applied in studies such as VideoLLaMA2. This demonstrates that GPT-3.5-turbo is well-suited to our objective of capturing semantic differences in the textual modality.
>
> Based on GPT-3.5-turbo's analysis results, we grouped video descriptions into Groups A to H to study the consistency and diversity of descriptive styles. Among these, Group A (high consistency) features highly consistent descriptions with concentrated semantic distributions, while Group H (high variability) exhibits significant differences in descriptive styles and semantic distributions.
>
> **2. Specific Case Analyses in Revisied Paper**
>
> To further validate the rationale of our evaluation method, we have included evaluation samples from different groups in the revised version to better demonstrate GPT-3.5-turbo's applicability in semantic analysis (see **Appendix B.5**).
>
> The text analysis capability of GPT-3.5-turbo enables it to efficiently evaluate the semantic differences in human annotations, making it suitable for text-based grouping tasks. By presenting examples of high consistency and high variability groups, we have validated the method's rationale and effectiveness. We sincerely thank the reviewer for their feedback, which has prompted us to further refine our evaluation design and research presentation.

---

### Official Review · Reviewer_fqpC · 2024-11-04

**Soundness:** 3
**Presentation:** 3
**Contribution:** 2
**Rating:** 6
**Confidence:** 4

**Summary:**

In this paper, the authors propose FIOVA benchmark, designed to evaluate the video description capabilities of LVLMs in comparison with human understanding. The authors address the limitations of current benchmarks by introducing a long-video dataset with diverse scenarios, and annotations from multiple annotators. The paper reports an in-depth evaluation of six state-of-the-art LVLMs and compares their performance with human annotations across various metrics. The findings highlight the discrepancies between LVLMs and human annotators, particularly in complex videos.

**Strengths:**

- The paper is clearly written and presents a well-structured methodology, results, and analysis.
- The collected dataset is of high quality, featuring perspectives from multiple annotators.
- The authors provide comprehensive evaluations of LVLMs using traditional metrics and AutoCQ-based metrics.

**Weaknesses:**

The overall paper reads more like a collection of experimental analyses rather than a benchmark for the following reasons:
- There are no experiments that demonstrate why the authors' dataset is superior to other datasets. For instance, is the FIOVA dataset better than the DREAM-1K dataset used in Tarsier for evaluation? Do the models that perform better on FIOVA also perform better in human evaluations?
- No new metrics are proposed. Traditional metrics are not well-suited for current LVLMs due to the nature of long responses. Additionally, is AutoCQ sufficient for evaluating dense captioning?

**Questions:**

See Weaknesses.

---

> ### Author Response · Authors · 2024-11-25
> **Response to Reviewer fqpC's concern about FIOVA's article style**
>
> ***Q1. The overall paper reads more like a collection of experimental analyses rather than a benchmark.***
>
> **Response:** We sincerely thank the reviewer for raising an important point about balancing experimental analysis and the positioning of the FIOVA benchmark. **We firmly believe that FIOVA not only satisfies the core requirements of a high-quality benchmark but also provides deep insights through experimental analysis, which have been highly appreciated by the other three reviewers.** Specifically, Reviewer SaKT noted that FIOVA offers extensive materials, making it a valuable resource for evaluating the video comprehension capabilities of LVLMs; Reviewer CA5F recognized FIOVA as being more comprehensive than previous benchmarks, providing realistic and challenging test cases for LVLM evaluation; and Reviewer pxyN highlighted FIOVA as a unique and important benchmark that advances video-language model development, representing a comprehensive understanding of video description tasks.
>
> Below is our detailed response:
>
> **1. FIOVA fully satisfies the core elements of a high-quality benchmark.**
>
> The purpose of a benchmark is to provide a standardized framework for evaluating model performance, typically requiring the following three key elements:
> - **High-quality dataset**: A benchmark must include diverse scenarios and tasks to support comprehensive model performance evaluation.
> - **Systematic evaluation methods**: A standardized evaluation framework and metrics are essential for objectively comparing different models.
> - **Introduction of new insights**: Experimental analysis should reveal model characteristics and limitations, offering directions for future research.
>
> FIOVA has been meticulously designed to adhere to these principles:
> - **Dataset (see Section 2)**: FIOVA contains 3,002 long videos (average duration 33.6 seconds), each annotated by five independent annotators. These annotations enhance semantic diversity and detailed coverage, forming the foundation for a robust human benchmark.
> - **Evaluation methods (see Section 3)**: We conducted a systematic evaluation of six representative open-source LVLM models, all of which are state-of-the-art in the field. We combined traditional metrics (e.g., BLEU, GLEU, METEOR) with event-level semantic metrics (AutoDQ) to validate model performance from multiple dimensions. Based on reviewer feedback, we also introduced a novel evaluation metric, FIOVA-DQ, tailored to the unique characteristics of FIOVA.
> - **New insights (see Sections 4.2 and 4.3)**: Our experiments reveal the strengths and weaknesses of LVLMs when handling complex long videos, such as their trade-offs between accuracy and completeness and their significant gaps compared to human benchmarks. These insights provide valuable guidance for future model development.
>
> These contributions have been positively acknowledged by the other three reviewers, who recognized our work on dataset quality, evaluation methods, and comprehensive analysis supporting the benchmark construction.
>
> **2. The length of experimental analysis is justified by the need to validate the benchmark’s evaluative capabilities.**
>
> The reviewer noted that the experimental analysis occupies a significant portion of the paper. We argue that experimental analysis plays a crucial role in benchmark-type papers. The following points illustrate the close connection between experiments and the benchmark:
>
> - **Experiments validate the benchmark’s effectiveness:**
>     - Our experiments demonstrate that FIOVA effectively reveals model performance and limitations in complex scenarios, confirming its value as a benchmark.
>     - For instance, by analyzing the deficiencies of LVLMs in consistency and detail coverage for complex video scenarios, we validated both the dataset design of FIOVA and its relevance to model evaluation.
> - **Experimental analysis provides long-term value to the field:**
>     - The experimental results directly support the positioning of FIOVA as a benchmark. For example, our findings show that LVLMs tend to adopt uniform strategies when handling complex situations, despite differences from the diverse strategies employed by humans. By uncovering these behavioral patterns, we provide directions for model optimization and improvement.
>
> We believe these improvements and clarifications make FIOVA’s positioning as a comprehensive benchmark even more explicit. Once again, we thank the reviewer for the valuable feedback, which has helped us refine the paper and better present its academic value.

---

> ### Author Response · Authors · 2024-11-25
> **Response to Reviewer fqpC's concern about the uniqueness of FIOVA**
>
> ***Q2. There are no experiments that demonstrate why the authors' dataset is superior to other datasets. For instance, is the FIOVA dataset better than the DREAM-1K dataset used in Tarsier for evaluation?***
>
> **Response:** We sincerely thank the reviewer for raising the important question about the uniqueness of the FIOVA dataset, as this feedback has been highly valuable in refining our work. We believe that **FIOVA offers substantial advantages over existing benchmarks.** In addition to the representative benchmarks listed and compared in Table 1 (e.g., HowTo100M, ACAV, YT-Temporal-180M, HD-VILA-100M, Panda-70M, MSVD, LSMDC, MSR-VTT, DiDeMo, ActivityNet, YouCook2, VATEX), FIOVA also includes the features of the DREAM-1K dataset, which the reviewer specifically highlighted.
>
> Below is our detailed response:
>
> **1. The core objective of FIOVA distinguishes it from DREAM-1K, emphasizing its unique contributions.**
>
> FIOVA is designed as a high-quality benchmark to evaluate LVLMs' performance in complex long-video understanding and generation tasks. Compared to DREAM-1K, FIOVA offers distinct advantages in several critical aspects:
>
> - **Video length and complex scenarios:**
>     - **FIOVA:** The average video length is 33.6 seconds, significantly longer than DREAM-1K's 8.9 seconds. This design increases the difficulty for LVLMs in handling cross-frame information integration and capturing spatiotemporal causal relationships, aligning better with real-world application demands.
>     - **DREAM-1K:** The shorter videos primarily focus on single- or multi-event action detection, making it more suitable for localized dynamic understanding tasks.
> - **Multi-annotator labeling:**
>     - **FIOVA:** Each video is annotated by five independent annotators, providing multi-perspective semantic coverage and significantly reducing the bias introduced by single annotators.
>     - **DREAM-1K:** Videos are annotated by a single annotator, which, while excelling in fine-grained action labeling, lacks the ability to provide multi-perspective descriptions.
> - **Diversity of themes and scenarios:**
>     - **FIOVA:** Covers a wide range of themes (e.g., long-video narratives, complex dynamic scenarios), better aligned with real-world application needs.
>     - **DREAM-1K:** Primarily focuses on short video action events (single or multiple), suitable for validating localized dynamic analysis.
>
> **2. FIOVA focuses on evaluation capabilities that differ significantly from DREAM-1K.**
>
> The uniqueness of FIOVA extends beyond its dataset design to its ability to enhance LVLM evaluation:
>
> - **Long-video semantic understanding:**
>     - DREAM-1K’s shorter videos limit the validation of models’ capabilities in complex narratives and multi-event scenarios. In contrast, FIOVA’s long-video design allows testing models’ ability to understand complex spatiotemporal causal relationships.
> - **Comprehensive multi-annotator perspectives:**
>     - FIOVA leverages multi-annotator annotations to create high-quality human benchmarks, offering more reliable diversity and detailed coverage. This design also lays the foundation for fine-grained comparisons between models and humans.
> - **Broader domain applicability:**
>     - Through its diverse themes and multi-event design, FIOVA evaluates models’ generalization and real-world applicability, whereas DREAM-1K primarily focuses on action-event detection tasks.
>
> These multidimensional contributions have been recognized by multiple reviewers:
> - **Reviewer SaKT:** Highlighted that FIOVA provides a wealth of materials, making it a valuable resource for evaluating LVLMs’ video comprehension capabilities.
> - **Reviewer CA5F:** Acknowledged that FIOVA is more comprehensive than previous benchmarks, focusing on complex spatiotemporal relationships in long videos and providing realistic and challenging test cases for LVLM evaluation.
> - **Reviewer pxyN:** Emphasized that FIOVA uniquely and comprehensively represents human understanding in video description tasks, advancing video-language model development.
>
> To further highlight FIOVA’s uniqueness, we have added a direct comparison between FIOVA and DREAM-1K in the **revised manuscript (Table 1)**.
>
> Once again, we thank the reviewer for the valuable feedback on the uniqueness of the FIOVA dataset. We believe the clarifications and improvements outlined above effectively demonstrate FIOVA’s critical contributions to evaluating long-video understanding tasks.

---

> ### Author Response · Authors · 2024-11-25
> **Response to Reviewer fqpC's concern about the alignment of FIOVA results with human judgment**
>
> ***Q3. Do the models that perform better on FIOVA also perform better in human evaluations?***
>
> **Response:** Thanks for propose this insightful question. **We believe that the design of FIOVA ensures alignment between model performance and human evaluation.** Below is our detailed response:
>
> **Providing a High-Quality Human Baseline with Multi-Annotator Strategy**
>
> - **Multi-Annotator Strategy:** Each video in the dataset is annotated by five independent annotators. This multi-annotator strategy not only captures the diversity of semantics but also minimizes the potential biases and omissions introduced by individual annotators. By integrating the descriptions from all five annotators, FIOVA establishes a comprehensive, objective, and multi-perspective human baseline.
>
> - **Representativeness and Comprehensiveness:** The groundtruth in FIOVA aggregates multi-perspective descriptions from five annotators, fully representing human understanding of video content, including complex spatiotemporal relationships and multi-event semantic capture. Consequently, the alignment of model descriptions with the groundtruth reflects the degree to which the models align with human video comprehension capabilities.
>
> **Ensuring Consistency Between Evaluation Metrics and Human Judgments**
>
> - **Combining Traditional Metrics with Semantic Consistency Evaluation:** We employed traditional text-matching metrics such as BLEU, GLEU, and METEOR, which are widely used in video description tasks to assess the semantic consistency between generated and reference descriptions. In addition, we introduced event-level semantic evaluation metrics (AutoDQ), which focus on fine-grained analyses of the alignment between video content and semantic descriptions. This provides a more precise assessment of how well the generated descriptions align with the actual video content.
>
> - **Optimizing AutoDQ:** As detailed in **Common Concern 2: Propose a New Evaluation Method FIOVA-DQ**, we further improved the calculation of AutoDQ in the revised version by incorporating weightings based on the five annotators’ descriptions. This enhancement strengthens its ability to assess human-like descriptive capabilities and led to the proposed modified metric, FIOVA-DQ (see **Sec 3.2 and Sec 4.1 of the revised paper**). Compared to traditional metrics, this optimized AutoDQ aligns more closely with human intuitive judgments of description quality.
>
> In conclusion, through the multi-annotator strategy that provides a high-quality human baseline, diverse evaluation metrics, and the optimized AutoDQ metric, FIOVA ensures a robust alignment between model performance and human evaluation.

---

> ### Author Response · Authors · 2024-11-25
> **Response to Reviewer fqpC's concern about the metrics**
>
> ***Q4. Traditional metrics are not well-suited for current LVLMs due to the nature of long responses. Additionally, is AutoDQ sufficient for evaluating dense captioning?***
>
> **Response:** Thanks a lot for raising this important question. First, we adopted traditional metrics (e.g., BLEU, METEOR, GLEU) due to their representativeness and general applicability. Additionally, we incorporated AutoDQ, which evaluates video understanding by disentangling event-level semantics, offering a more fine-grained perspective compared to traditional metrics. **AutoDQ demonstrates the ability to evaluate complex video description tasks**, especially for dense, long video descriptions. Building on this, and in response to the reviewer’s suggestions, we optimized AutoDQ by leveraging FIOVA’s unique characteristics (i.e., descriptions from five human annotators per video) and **proposed a new metric, FIOVA-DQ (see Common Concern 2: Propose a New Evaluation Method FIOVA-DQ)**. Below is our detailed response:
>
> **Rationale for Choosing Traditional Metrics**
>
> In this work, we selected traditional text generation metrics (e.g., BLEU, METEOR, GLEU) for the following reasons:
>
> - **Widespread Use:** Traditional metrics are widely adopted in the text generation domain and have become standard methods for model evaluation. Their use ensures that our results can be directly compared with prior studies, thereby enhancing the academic value and comparability of our experiments.
> - **Language Quality Evaluation:** These metrics remain effective for evaluating fundamental aspects of textual outputs, such as linguistic structure, syntactic correctness, and alignment with reference descriptions. For long video description tasks, they provide a reliable foundation for assessing the linguistic quality of generated texts.
>
> We also acknowledge the limitations of traditional metrics in long video description tasks, which are discussed in detail in **Section 4.2**. For instance:
> - **Inability to Capture Semantic Consistency:** Traditional metrics are primarily text-matching-based and often fail to capture the event-level semantic relationships between descriptions and video content, such as temporal consistency and causal logic.
> - **Lack of Focus on Fluency and Detail Coverage:** These metrics cannot comprehensively assess the fluency of descriptions or the completeness of detailed coverage.
>
> **Introduction of AutoDQ as a Complementary Metric**
>
> To address the limitations of traditional metrics, we introduced AutoDQ, a recently proposed evaluation metric (July 2023) that provides fine-grained assessment and analysis from an event-level perspective. Compared to traditional metrics, AutoDQ offers several advantages:
>
> - **Event-Level Semantic Analysis:** AutoDQ evaluates the fine-grained semantic alignment between generated descriptions and video content by extracting and matching events. For instance, it can assess the model’s performance in capturing event sequences, logical relationships, and causal chains in multi-event long videos.
> - **Suitability for Complex Long-Video Scenarios:** AutoDQ effectively captures the completeness and consistency of multi-event descriptions in long videos. It provides targeted evaluation, especially in scenarios where traditional metrics lack discriminatory power.
>
> In our experiments, AutoDQ demonstrated the following capabilities:
>
> - **Discriminatory Power at the Event Level:** AutoDQ scores reveal differences in models' abilities to capture event-level semantics. For instance, when describing multi-event long videos, AutoDQ identifies strengths and weaknesses in models’ handling of event order, logical relationships, and semantic coherence.
> - **Applicability to Complex Scenarios:** AutoDQ effectively disentangles semantic complexities in long videos, capturing the completeness and consistency of multi-event descriptions. It complements traditional metrics in scenarios where they fall short.
>
> Although AutoDQ has shown considerable applicability, we acknowledge its room for improvement. In response to the reviewer’s suggestions, we further optimized AutoDQ during the rebuttal process by incorporating FIOVA’s characteristics.
>
> **Introduction of a New Evaluation Metric**
>
> We improved AutoDQ by weighting events based on the descriptions from five human annotators. This enhancement strengthens AutoDQ’s ability to evaluate human-like descriptive capabilities, leading to the proposed new metric, FIOVA-DQ (**see revised paper, Sec 3.2 and Sec 4.1**). Compared to traditional metrics, the optimized AutoDQ aligns more closely with human intuitive judgments of description quality.

---

> > ### Comment · Reviewer_fqpC · 2024-11-27
> > **Reply to authors' response**
> >
> > Thanks for the comprehensive rebuttal. It solves most of my problems.
> >
> > I tend to increase the score and hope that the authors can make the benchmark open-sourced to motivate in-depth research on dense caption.

---

> > > ### Author Response · Authors · 2024-11-27
> > > **Thanks for the affirmation and encouragement of the Reviewer fqpC**
> > >
> > > Dear Reviewer fqpC,
> > >
> > > We sincerely thank you for your thoughtful review and for taking the time to re-evaluate our work. Your recognition of our efforts and **the increase in score** immensely encourages our team. We are especially grateful for your acknowledgment of the comprehensive rebuttal and your confidence in the potential impact of our benchmark.
> > >
> > > As you suggested, we are fully committed to making the FIOVA benchmark open-sourced. We aim to foster in-depth research on dense video captioning and provide the community with a robust and well-maintained resource for future advancements. Your feedback has been invaluable in shaping this vision, and we are excited to contribute to developing this important research area.
> > >
> > > Thanks again for your constructive comments and encouragement!

---

### Author Response · Authors · 2024-11-25
**Rebuttal Summary**

We sincerely thank the reviewers and the Area Chair (AC) for their constructive feedback and their recognition of our work. The reviewers have highlighted the following key contributions of our paper:

1. **High-quality dataset**: The FIOVA dataset was praised for its richness and comprehensiveness, particularly the multi-perspective annotations provided by five annotators per video (**Reviewer fqpC, Reviewer CA5F**). The inclusion of long videos with complex spatiotemporal relationships was recognized as a significant advancement over existing benchmarks (**Reviewer pxyN**).

2. **Innovative evaluation framework**: The systematic and comprehensive nature of the multi-metric evaluation framework was highly appreciated (**Reviewer fqpC**).

3. **Performance analysis of LVLMs**: The fine-grained comparison of LVLM outputs and human annotations was commended for providing valuable insights into the strengths and limitations of LVLMs (**Reviewer CA5F, Reviewer pxyN**). The evaluation of six SOTA LVLMs was noted as a valuable benchmark contribution (**Reviewer pxyN**).

4. **Impact on future research**: FIOVA was considered a challenging and critical resource, especially for understanding complex real-world video scenarios (**Reviewer SaKT, Reviewer pxyN**).

5. **Paper quality and comprehensiveness**: The reviewers acknowledged the clarity of the paper's structure and its high-quality writing (**Reviewer fqpC, Reviewer CA5F**) as well as the richness of the supporting materials (**Reviewer SaKT**).

Based on the reviewers' invaluable suggestions, we have made extensive revisions to the paper, with all changes highlighted in blue. The key revisions are as follows:

1. **Introduction of the FIOVA-DQ metric:**
- Detailed explanations of the FIOVA-DQ metric were added in Sec 3.2 and Sec 4.1, with illustrative examples provided in the revised Fig. 4. This metric incorporates event importance weights based on human cognition, enhancing the fairness and accuracy of the evaluation.
- Additional experimental analyses based on FIOVA-DQ were included in Sec 4.2, with fine-grained results presented in Appendix E.2.
2. **Improved transparency and reliability of groundtruth generation:**
- Fig. 4 was revised, and its generation process was elaborated in Appendix B.6 to ensure transparency and accuracy.
- Explanations of video data sources and collection methods were added in Appendix B.1.
- Quality control measures for human annotations were detailed in Appendix B.2.
3. **Expanded experiments and evaluations:**
- Detailed justifications for selecting six evaluated models were added in Appendix D.2.
- Further clarification of batch ranking computations was provided in Sec 4.2, supplemented by a visual illustration (Fig. A8) in Appendix C.
- Fine-grained error annotations and analyses were added in Appendix E.4, exploring model limitations and proposing optimization directions.
4. **Additional refinements:**
- Annotation style-based grouping examples were included in Appendix B.5, showcasing case studies of Groups A and H.
- Further explanations for Table A.3 were added in Appendix E.1.

We sincerely thank the reviewers for their valuable suggestions and support. These insights have significantly enhanced the quality and impact of our paper. We believe the revised manuscript is more rigorous and compelling. In the following rebuttal, we first address the two common concerns raised by reviewers, followed by point-by-point responses to each reviewer's specific comments.

---

### Author Response · Authors · 2024-11-25
**Common Concern 1. Reasons and quality of using GPT-3.5-turbo to generate groundtruth**

We sincerely thank the reviewer for their concerns regarding the Groundtruth (GT) generation process and quality. The FIOVA benchmark is designed to provide high-quality, semantically consistent, and diverse GT by employing a multi-annotator strategy combined with LLM-based integration, enabling a comprehensive evaluation framework for long video description tasks. **Multiple measures have been implemented to ensure the GT's quality.** Below is our detailed response:

**1. Design Philosophy of FIOVA: Ensuring GT Quality**
- **Multi-Annotator Strategy:** Each video was annotated by five rigorously selected and trained independent annotators. After watching the full video, the annotators provided detailed descriptions based on their observations, ensuring that all descriptions strictly aligned with the video content. This multi-perspective annotation method not only captures rich semantic details but also minimizes potential bias or omissions from a single annotator.
- **LLM-Based Text Integration:** GPT-3.5-turbo was employed to semantically merge and linguistically enhance the multi-perspective descriptions provided by annotators. Its role is strictly confined to integration and optimization of text rather than generating new content or subjectively interpreting the video. This approach avoids potential interference from multimodal information while preserving the diversity and accuracy of the annotators' descriptions. This methodology has been validated and widely applied in related research (e.g., Tarsier [1], MM-Ego [2]).

**2. Addressing Specific Issues in GT Generation**
- **Hallucination Issues:** We opted to use LLMs (e.g., GPT-3.5-turbo) instead of VLMs (Vision-Language Models) for the integration task to reduce hallucination issues that could arise from multimodal alignment, such as generating content not present in the video. With carefully designed prompts, GPT-3.5-turbo focuses on ensuring semantic consistency during integration, thereby avoiding redundant or irrelevant content.
- **Resolving Annotator Conflicts:** As analyzed in Appendix B.6 and illustrated in Figure A7 of the revised paper, when annotators provided conflicting descriptions for certain events (e.g., "smiling" vs. "crying"), GPT-3.5-turbo synthesized the descriptions and the contextual logic of the video to arrive at a reasonable semantic conclusion. For instance, by combining Human 1's mention of the "pretend" action and Human 5's description of the boy smiling, the final conclusion aligns better with the video's overall semantics.
- **Addressing Omission of Details:** The multi-annotator strategy ensures that GT covers all core events comprehensively. During the generation phase, we optimized prompt designs through multiple iterations and manually reviewed and corrected the GT to minimize omissions and redundancies.

**3. Measures to Enhance GT Quality**
- **Prompt Optimization:** We refined prompt designs through multiple rounds of experiments to guide GPT-3.5-turbo in producing high-quality, semantically consistent, and linguistically fluent text.
- **Manual Review and Correction:** All 3002 GTs underwent comprehensive manual checks, with problematic descriptions regenerated and corrected. Corresponding experimental results were also updated to ensure that the GT is comprehensive, accurate, and consistent.

**4. Significance and Future Directions**
Through the construction of high-quality GT, FIOVA provides a more comprehensive and impartial evaluation framework for long video description tasks. The integrated GT:
- **Comprehensive Coverage:** Reflects the core semantics of the video content, encompassing multi-perspective and diverse descriptive details.
- **Reliable Evaluation:** Significantly enhances the consistency and reliability of LVLM evaluation in long video tasks.
We firmly believe that the design philosophy and practical methodology of FIOVA not only address the reviewer's concerns but also provide strong support for future research.

[1] Wang J, Yuan L, Zhang Y, et al. Tarsier: Recipes for training and evaluating large video description models[J]. arXiv preprint arXiv:2407.00634, 2024.

[2] Ye H, Zhang H, Daxberger E, et al. MM-Ego: Towards Building Egocentric Multimodal LLMs[J]. arXiv preprint arXiv:2410.07177, 2024.

---

### Author Response · Authors · 2024-11-25
**Common Concern 2. Propose a New Evaluation Method FIOVA-DQ**

We sincerely thank the reviewers for their insightful feedback on evaluation metrics. Based on the reviewers' suggestions, we extended the original AutoDQ method and **designed a novel weighted evaluation metric, FIOVA-DQ, tailored to the characteristics of the FIOVA benchmark.**

Below, we provide a detailed explanation of this new metric：

AutoDQ, proposed by Wang et al., is an event-based evaluation method that extracts and matches events from reference and model-generated descriptions. While AutoDQ offers fine-grained assessment, it does not incorporate human annotators' cognitive understanding of event importance. FIOVA-DQ addresses this limitation by achieving the following goals:

- **Incorporating Human Cognition:** Assign weights to events based on human annotators' perspectives, ensuring more critical events have greater influence on the evaluation.
- **Improving Evaluation Precision:** Calculate weighted precision and recall based on event alignment, offering fairer and more representative evaluation results.

---

**FIOVA-DQ Calculation Process (refer to Fig.4 in the revised manuscript)**

**1. Event Extraction**
FIOVA-DQ starts by extracting events from human annotations, groundtruth descriptions, and model-generated outputs:
- Extract events from human annotations: Create event lists $H_1, H_2, H_3, H_4, H_5$ from the five annotators' descriptions.
- Extract events from Groundtruth (GT): Use GPT-3.5-turbo to extract events from the aggregated GT description, forming list $G$.
- Extract events from model outputs: Extract events from the model-generated descriptions, forming list $V$.

**2. Event Weighting**

To reflect human cognition, weights are assigned to each event in the GT. The process is as follows:
- Ranking for individual references:
    -  For each event $g_i$ in $G$, calculate its similarity with events in $H_1$.
    -  Rank $g_i$ based on similarity, repeating for all annotators ($H_1, H_2, H_3, H_4, H_5$), and calculate the average rank for each $g_i$.
-  Weight Generation:
    -  Normalize the average rank (average rank/total rank sum) to assign weights.
    -  For example, in Fig.4, the GT events $g_1, g_2, \dots, g_7$ have weights $w_{g1}=0.114, w_{g2}=0.182, \dots, w_{g7}=0.114$.
    -  The same process is applied to the model events $v_1, v_2, \dots, v_6$, resulting in weights $w_{v1}=0.053, w_{v2}=0.093, \dots, w_{v6}=0.173$.

**3. Precision and Recall Calculation**
Based on the weighted results, precision and recall are computed:
- Precision: The sum of weights for matched events in $V$ divided by the total weight of $V$. For Fig.4, matched events $v_2, v_4, v_6$ have weights $0.093, 0.227, 0.173$, yielding Precision = $0.493$.
- Recall: The sum of weights for matched events in $G$ divided by the total weight of $G$. For Fig.4, matched events $g_1, g_4$ have weights $0.114, 0.219$, yielding Recall = $0.333$.

---

**Comparison with AutoDQ**

FIOVA-DQ significantly improves upon AutoDQ:

1. **Incorporates Human Cognition:** Reflects human understanding of event importance.

2. **Enhances Evaluation Effectiveness:** Prioritizes key events over minor ones.

3. **Aligns with FIOVA's Multi-Perspective Philosophy:** Leverages multi-annotator insights for comprehensive and fair evaluation.

We believe FIOVA-DQ provides a robust framework for evaluating LVLM performance in long video description tasks and offers valuable insights for future research.

---

### Meta-Review · Area_Chair_rqto · 2024-12-20

**Metareview:**

This paper proposed a new benchmark to evaluate the video caption of LVLMs. Different from existing benchmarks, the proposed benchmark contains longer videos (33.6 seconds on average) with five human annotations for each video. It applies GPT-3.5 evaluate the quality of human annotations and also use it to aggregate all annotations to generate ground truth captions. A novel metric FIOVA-DQ is also proposed as a more human-aligned metric for LVLMs.

Strength:
1. The collected dataset is different from all existing dataset: (a) Average duration is longer than existing dataset; (b) Have multiple human annotations.
2. This paper provide in-depth evaluation of six open-sourced SOTA methods.
3. As suggested by reviewers, a new metric FIOVA-DQ which is more human-aligned is proposed.

Weakness:
The biggest concern raised by several reviewers is the quality of the ground truth. The authors spend a lot of effort to answer questions about this, the concerns are not fully addressed.

**Additional Comments On Reviewer Discussion:**

1. One of major concerns raised by reviewers are the quality of the ground truth. For instance, using text only LLM GPT-3.5 turbo to summary human annotations as ground truth may not be a good choice compared with Multimodal model. Authors claimed that Multimodal  model may introduce hallucination, however, reviewer CA5F and pxyN also pointed out the obvious flaws of the ground truth generated by GPT-3.5 turbo in the original version. Finally, the ground truth needs to be verified by human. As a result, the argument of not using Multimodal model is not so strong.
2. Reviewer pxyN gave several good suggestions about the ground truth. Reviewer pxyN mentioned many of the concerns have been addressed, but the quality of ground truth still needs to be improved. Reviewer pxyN lowered the score from 8 to 6.
The above two concerns is the major reason I tend to reject this paper.

Some other concerns raised by reviewers are also not fulled addressed. For instance, reviewer fqpC asked the authors to include some experiments to demonstrate proposed benchmark is better than DREAM-1K. The authors only emphasized the difference between the two datasets. One improvement the author can do is to evaluate the six open sourced models on both FIOVA and DREAM-1K using the same metric, then show that  FIOVA may reflect some issues of existing models that DREAM-1K cannot diagnose.

Overall, I feel this paper has potential to benefit the community, however, it still needs some improvement to be qualified for a publication.

---

### Decision · Program_Chairs · 2025-01-22

Reject